# Deep sequence models tend to memorize geometrically; it is unclear why.

Shahriar Noroozizadeh [1][†]  Vaishnavh Nagarajan [2]  Elan Rosenfeld [2]  Sanjiv Kumar [2]

## Abstract

Deep sequence models are said to store atomic facts predominantly in the form of *associative* memory: a brute-force lookup of co-occurring entities. We identify a dramatically different form of storage of atomic facts that we term as *geometric* memory. Here, the model has synthesized embeddings encoding novel *global* relationships between all entities, including ones that do not co-occur in training. Such storage is powerful: for instance, we show how it transforms a hard reasoning task involving an $\ell$-fold composition into an easy-to-learn 1-step navigation task. From this phenomenon, we extract fundamental aspects of neural embedding geometries that are hard to explain. We argue that the rise of such a geometry, as against a lookup of local associations, cannot be straightforwardly attributed to typical supervisory, architectural, or optimizational pressures. Counterintuitively, a geometry is learned even when it is more complex than the brute-force lookup. Then, by analyzing a connection to `Node2Vec`, we demonstrate how the geometry stems from a spectral bias that—in contrast to prevailing theories—indeed arises naturally despite the lack of various pressures. This analysis also points out to practitioners a visible headroom to make Transformer memory more strongly geometric. We hope the geometric view of parametric memory encourages revisiting the default intuitions that guide researchers in areas like knowledge acquisition, capacity, discovery, and unlearning. [1]

## 1. Introduction

When succinct high-level patterns explain the data, deep sequence models produce corresponding high-level representations, as often witnessed in natural language and arithmetic tasks. When no such patterns exist (e.g., as in the capitals of countries), models are said to default to a brute-force lookup, known as *associative* memory (Radhakrishnan et al., 2020; Bietti et al., 2023). These two narratives have so far roughly guided our understanding of how neural networks fit sequential data. We highlight a third behavior overlooked in this narrative: even when memorizing such incompressible atomic co-occurrences, a deep sequence model can produce powerful representations. This implies a distinct form of parametric memory that we term as *geometric* memory—glimpses of which can be found scattered in recent observations (Khona et al., 2024; Nishi et al., 2025; Huang et al., 2025; Ye et al., 2025). In opposition to the well-studied associative memory, a geometric memory has synthesized "global" information not explicit in the local co-occurrences, enabling new forms of reasoning. From this behavior, we extract aspects of neural embedding geometries that are hard to explain, raising fundamental questions about memorization in deep sequence models. To these questions, we offer some preliminary answers.

To understand parametric memory, we study simple tasks involving graph memorization. These are tasks where the model is made to memorize the edge bigrams of an underlying graph in the model's weights. As a starting point, we consolidate a fragmented set of recent demonstrations (Khona et al., 2024; Wang et al., 2024a; Feng et al., 2024; Geerts et al., 2026; Ye et al., 2025; Huang et al., 2025) that, over such graphs, the Transformer can do some level of *implicit in-weights reasoning*, i.e., reasoning over parametric knowledge without emitting an explicit chain of thought. We sharpen these results by crafting a scenario where this ability is unexpected, plays out vividly, and can be cleanly isolated and analyzed. Specifically, we study path-finding on path-star graphs, a (symbolic) implicit reasoning task. The task was adversarially designed (Bachmann & Nagarajan, 2024) to cause failure of next-token trained deep sequence models—Transformer (Vaswani et al., 2017) and Mamba (Gu & Dao, 2023) models alike. Whereas in the original

[†] Work done during internship at Google Research. [1] Machine Learning Department & Heinz College, Carnegie Mellon University, Pittsburgh, PA, USA [2] Google Research, NY, USA. Correspondence to: Shahriar Noroozizadeh <snoroozi@cs.cmu.edu>, Vaishnavh Nagarajan <vaishnavh@google.com>.

*Proceedings of the 43rd International Conference on Machine Learning*, Seoul, South Korea. PMLR 306, 2026. Copyright 2026 by the author(s).

[1]Code for reproduction is at:
https://github.com/shahriarnz14/Geometric_Memory.

task, the model is given the graph in-context, here we make the model memorize the graph's edges in its weights. Where before the model spectacularly failed to learn path-finding even on small graphs, in our in-weights task the model succeeds even on massive graphs.

The success in the in-weights path-star task, we argue, is hard to reconcile within the *associative* view of parametric memory, a convenient and highly effective abstraction of neural network memory, popular in literature (see §2.3 for references). In this abstraction (see Def. 2a), each entity is embedded randomly/near-orthogonally via a "key" function $\Phi(\cdot)$, while associations are stored in what is effectively a (rotated) adjacency matrix, $\mathbf{W}_{\texttt{assoc}}$. To perform a "lookup", one computes the quantity $\Phi(v)^T \mathbf{W}_{\texttt{assoc}} \Phi(u)$ which is large if *and only if* entities $u$ and $v$ co-occur during training. With such a data structure, our path-finding task requires composing the lookup operation $\ell$ times (for path length $\ell$). Unless there is step-wise supervision for each composition, learning the $\ell$-fold composition should be a daunting needle-in-the-haystack task demanding $\exp(\ell)$ time. By the specific design of our task, *every possible form of step-wise guidance is eliminated*. Yet our model appears to find the needle.

This apparent paradox begins to be resolved by a preliminary observation in Jiang et al. (2024a); Khona et al. (2024); Nishi et al. (2025); Ye et al. (2025): that the node embeddings $\Phi$ reflect some useful notion of distance; and a separate finding in Huang et al. (2025); Saxe et al. (2019): that (two-hop) reasoning can emerge from "factorized" parameterizations. We flesh out a unified understanding of these observations and their implications. First, the existence of structured node embeddings implies that models can store atomic facts in an altogether distinct paradigm of parametric memory that we term as geometric memory. This memory is in dramatic contrast to the associative one. In the simplest form of the geometric view (see Def. 2b), the associations are no longer stored verbatim in a lookup table $\mathbf{W}_{\texttt{assoc}}$, but are *(low-rank) factorized* into the embeddings as $\Phi_{\texttt{geom}}(u)^T \Phi_{\texttt{geom}}(v)$. Our key insight is that these embeddings are no longer arbitrary. Rather, they are carefully arranged such that they encode *global* relationships without explicit supervision to do so: even if entities $u$ and $v$ never appeared in the same context, the dot-product $\Phi_{\texttt{geom}}(u)^T \Phi_{\texttt{geom}}(v)$ captures the model's own notion of multi-hop distance between the pair. This global geometry can open up new forms of reasoning, since what seemed to be a hard-to-learn $\ell$-fold composition of local associations is now approximated as an easy-to-learn spatial navigation task. We contrast the associative (Def. 2a) and geometric (Def. 2b) views of memory visually in Figure 1 and also in Table 1.

The observed geometry raises fundamental questions. Most importantly, there must be competition between the two parametric memories, both equally valid solutions to the training objective; why does the geometric prevail over the associative? To some readers, a geometric bias may seem familiar and intuitive at first sight, but we isolate aspects that cannot be easily explained: the global geometry is learned even when there is no "global" supervision (e.g., a path-finding objective), even when there is no rank constraint, even when the geometry is much harder to find, and even when it is just as succinct as a lookup. Thus, typical pressures from the supervision, architecture or optimization do not explain why a highly non-trivial geometry is synthesized. We term this the *memorization puzzle*. Towards understanding this, we reduce the setup to a minimal two-layer, `Node2Vec`-style architecture (Grover & Leskovec, 2016), where we find that global geometries emerge from well-known spectral biases in such architectures. But, in contrast to prevailing theories (e.g., Levy & Goldberg (2014); see §B.5), we identify how the spectral bias arises naturally from cross-entropy loss minimization, independent of typically-assumed pressures. While this gives us a preliminary insight into the natural rise of a geometric memory in a simple model, we leave open a foundational question for deep sequence models.

**Implications.** Although the evidence of implicit reasoning is so far limited to symbolic tasks, our insights open up broader directions. First, we find that the embeddings learned by more naive `Node2Vec` models are more strongly geometric than those of Transformers; this raises the question of how to make Transformer memory more geometric and less associative in practice. The geometry, when established in larger scale settings, could help reason and discover creative connections between information scattered in a large pretraining set. On the flip side, the interdependencies in geometric storage may impose limits on knowledge editing, unlearning, and accurate retrieval; it may also hallucinate associations. Our study is also support for why parametric memory may be superior to in-context memory. Broadly, we speculate that the associative view forms the default unstated set of intuitions that guide research in areas such as knowledge acquisition, discovery, unlearning, reasoning, and storage capacity; the geometric view may inspire researchers to revisit, spell out, and widen these latent intuitions.

### 1.1. Summary of contributions

1. We devise a clean instance of implicit in-weights reasoning in deep sequence models that succeeds even without any form of step-wise supervision. This isolates a behavior that is hard to explain within the predominant associative view of parametric memory. (§2)

2. We establish an alternative, global *geometric* view of memory in deep sequence models (simple neural networks, Transformers, and Mamba). (§2.4)

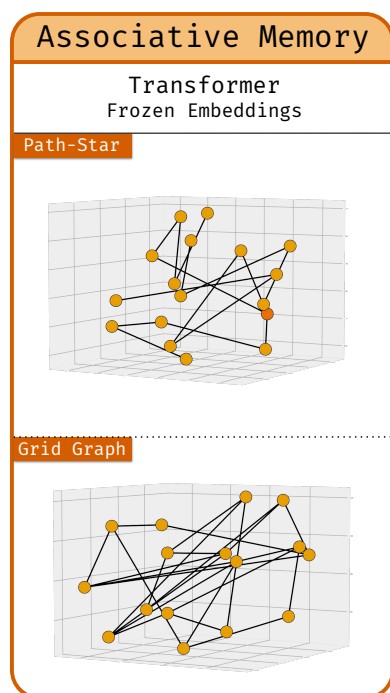
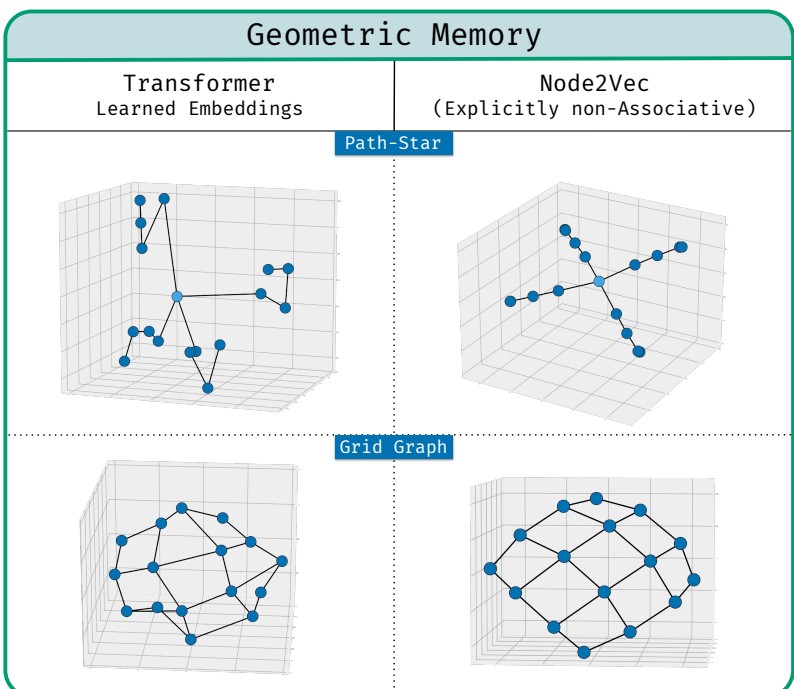

*Figure 1.* **Associative vs. geometric memory of models trained on various graphs.** There are two dramatically different ways to memorize a dataset of atomic facts. The common view is of associative memory: entities are embedded arbitrarily, and co-occurrences are stored in weight matrices. (**left**). §2.4: In practice, we find a geometric memory: the learned embeddings of a Transformer (**middle**) reflect *global* structure inferred from the *local* co-occurrences in training data. §4: When associative memory is explicitly prohibited (by removing intermediate layers), as in a 2-layer, `Node2Vec`-style model (**right**), a more elegant geometry materializes. This points to a clear headroom to improve the geometric nature of a Transformer's memory. Details of the Transformer architecture used for this visualization are provided in §D.2.2. Similar geometries for Mamba and neural networks are presented in §E.3. Also see Fig 15 for more graphs.

3. We demonstrate why the geometry is surprising: the emergence of geometric over associative memory cannot be attributed to obvious architectural, optimizational or supervisory pressures. With this, we formulate a *memorization puzzle* in sequence modeling. (§3)

4. We connect the global geometry to the spectral bias of 2-layer `Node2Vec` models. We empirically intuit how it emerges without typically-assumed pressures, when minimizing the cross-entropy loss. This makes progress towards an open question in `Node2Vec`. We also highlight significant headroom in the embedding geometry of the current architectures. (§4)

## 2. Implicit in-weights reasoning is learned

We investigate planning on a *path-star graph* (Figure 2), a graph designed to be adversarial towards next-token learning (B&N'24). The task has a clear notion of a chain-of-thought and a well-understood mechanism of failure for learning in-context reasoning. Thus, we repurpose this to cleanly analyze a different type of reasoning—in-*weights* reasoning—in a way that was not possible in earlier studies of in-weights reasoning.

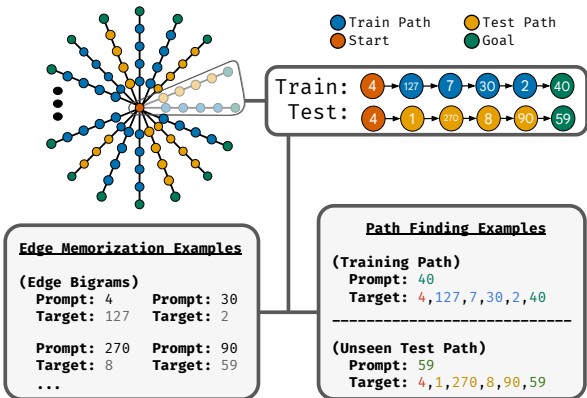

*Figure 2.* **Our in-*weights* path-star task.** All train/test examples are generated from a fixed path-star graph: (i) **edge memorization** examples (covering all edges during training) and (ii) **path-finding** examples, partitioned into a non-overlapping train and test set. Fuller version in Figure 8.

### 2.1. The in-weights path-star task

**Our task definition.** The path-star topology is a tree graph where multiple *disjoint uniform-length* paths branch out from the root. For our task, all examples are generated from a *fixed* (giant) graph $\mathcal{G}$ of degree $d$ and path length $\ell$. Part of

the training data consists of "edge-memorization" examples where the input prompt is some node $v$ and the target is an adjacent node $v'$ in $\mathcal{G}$. These training examples cover *all* the edges so that the model memorizes all of $\mathcal{G}$ in its weights. Separately, we generate path-finding examples by picking as input prompt a random leaf node $v_{\texttt{leaf}}$ and as target, the unique root-goal path (a sequence of tokens from $v_{\texttt{root}}$ to $v_{\texttt{leaf}}$). We train for path-finding on only a *subset* of such leaves, and test on the remaining "unseen" leaves, leaves whose root-to-leaf paths are never seen end-to-end—only the constituent edges are seen individually, which doesn't trivialize the path-finding task. (We extend our experiments to harder graphs in §E.2).

**Our experimental setup.** We make some notes about our setup. First, for the most stable results, we interleave the edge-memorization and path-finding examples during training as in Figure 8 and also use pause tokens (Goyal et al., 2024). We found that it is important to provide both forward and reverse edges for edge-memorization, but this may not be necessary for smaller graphs; see §E.3.1. Reverse augmentation is *not* given for the path-finding examples. We use from-scratch, decoder-only Transformer (GPT-mid) (Radford et al., 2019) and corroborate all findings on Mamba in §E.1. The formatting and the hyperparameters are in §D, followed by additional analyses in §F.

**Comparison to the original in-*context* task.** Our task is inspired by, yet fundamentally different from the original task in B&N'24. There, the model was given in *context*, a fresh example of a path-star graph as an adjacency list (with randomized node labeling and edge ordering), along with a leaf node specified as goal (see Figure 9 for this original version). We, however, generate all examples from a fixed in-weights graph, and give no adjacency list in-context as we hope to understand how well the model recalls and reasons over information stored in the weights.

**The adversarial design of the graph.** In the in-context version of B&N'24, next-token learners are known to fail in-distribution even on tiny graphs. The failure is due to the deliberate design of the graph: during training, the optimizer can fit all but the first token in the path trivially based on the preceding token; the subsequent lack of gradients from these tokens renders the first (decision-making) token (following the root) into an exponentially hard-to-learn implicit reasoning task. This hardness is particularly of interest to us and will be discussed shortly. For reference, we elaborate on this in-context task in §C and reproduce this failure in Figure 11.

## 2.2. Success of implicit in-weights reasoning

The path-star graph is adversarially constructed and involves a particularly hard-to-learn token. Yet, in our in-weights version of this task, we find that next-token-trained models

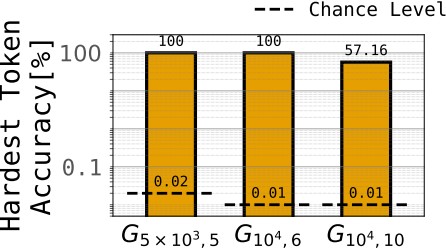

*Figure 3.* **Success of Transformer in in-weights path-star task.** **(§2)** On large path-star graphs $\mathcal{G}_{d,\ell}$, a next-token-trained Transformer achieves perfect or highly non-trivial accuracy, even when trained only on the first-token in isolation. the $Y$-axis is in log scale to compare against chance accuracy ($1/d$; dashed line). More detailed plots in Figure 10; analogous plots for Mamba in Figure 12. We invite the reader to contrast these in-weights task results with the in-context task ones in Figure 11.

succeed at path-finding, even on path lengths of 6 to 10 hops and graphs as large as $10^4$ nodes. In fact, even when we manually eliminate all supervision from the later tokens in the path, and train only on the loss of the first token loss— the key decision-making token—the model predicts this token successfully on unseen paths (Figure 3). This is striking because learning the first token in isolation, one would intuitively expect, is (theoretically and empirically) hard— an intuition we will shortly elaborate on. This success also isolates an instance of implicit in-weights reasoning sharper than what exists in the literature. Known instances are on small scales of 200 or fewer entities or on 2-hop tasks (see §B.2). We report success on some harder topologies in §E.2.

**Observation 1.** *(**Hardest, first token is learned in isolation**) On in-weights path-star graphs of as many as $5 \times 10^4$ nodes, when the model is trained on only edge-memorization examples and first-token-training examples from $75\%$ of all the paths, on unseen paths, both the Transformer and Mamba are able to predict the first token when conditioned on held-out leaves, with as much as $100\%$ accuracy (Figures 3 and 12 (right)).*

## 2.3. The contradiction behind learning the hardest token

That the hardest (first) token is learned, we argue, is difficult to square with the default abstraction of memory in neural networks. Atomic facts are often imagined to be stored in the parameters as local associations, where a matrix acts as a lookup table. We make this concrete. For any tokens $u$ and $v$, let $f(u)[v]$ denote the logit of a sequence model $f$ on the target $v$, conditioned on the input $u$. Then:

> **Definition 2a.** *(Local associative parametric memory)*
> *We say that a deep sequence model $f$ has memorized a graph $\mathcal{G} = (V, E)$ associatively iff for any vertices $u$ and $v$, the logit $f(u)[v]$ is high only on adjacent vertices, $(u, v) \in E$. In its simplest form, this model is typically abstracted as $f(u)[v] = \mathbf{\Phi}(u)^T \mathbf{W}_{\texttt{assoc}} \mathbf{\Phi}(v)$. Here, $\mathbf{W}_{\texttt{assoc}}$ encodes the adjacencies as $\sum_{(u,v) \in E} \mathbf{\Phi}(u) \otimes \mathbf{\Phi}(v)$; the embeddings $\mathbf{\Phi}$ are arbitrary "keys" that by themselves encode no associations in graph $\mathcal{G}$, e.g., they are modeled as orthogonal or random embeddings in literature (see Remark 1 for references).*

Observe that, in this abstraction, the first node in our task requires implicitly composing a local recall function $\ell$-many times: (implicitly) recall the predecessor of $v_{\texttt{leaf}}$ from the weights, namely $v_{\ell-1}$, then the predecessor of that, $v_{\ell-2}$, and so on (all without an explicit chain of thought). In short, one can write $v_1 = \texttt{Predec}^\ell(v_{\texttt{leaf}})$.

**Hardness of learning compositions.** With this structure however, the first-token $\ell$-fold composition task should intuitively require $\Omega(\exp(\ell))$ compute to learn—just like the first token was demonstrably hard-to-learn in the in-context path-star task (B&N'24). This hardness can be theoretically formulated in many ways (see §B.3); one intuition is that the optimization task is a search for a needle in the haystack: there is an $\exp(\ell)$ space of possible discrete compositions, and all but the correct one have an equally miserable loss value; with the loss terrain rendered flat and the gradients uninformative, the learner is forced to sift through the vast hypothesis space to find a needle.

**The power of step-wise aid.** This barrier could be surmounted if, rather than providing supervision from only the end output of the composition, there was supervision from each hop, say in the form of a data curriculum or chain-of-thought supervision. Indeed, prior positive results on implicit in-weights reasoning involve one such aid or the other (see §B.2).

**Our setting, by design, offers none of these aids.** Our paths have fixed lengths (hence, no implicit curriculum), all later token gradients eliminated (so no supervision from the intermediate hops), the paths disjoint (so, no test-train overlaps), and there is as much as a 10-fold composition (so, a daunting exponent). What we isolate, therefore, is a stronger and more sterile instance of implicit in-weights reasoning, one that is cleanly inconsistent within the associative view of parametric memory.

### 2.4. Two competing forms for parametric memory

The resolution to this paradox lies in a dramatically different view of parametric memory: a memory of atomic facts that does not take shape as a lookup over arbitrary embeddings

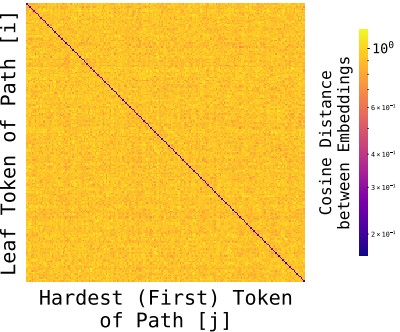

*Figure 4.* **Evidence of global geometry of Transformer in path-star task.** In the heatmap, entry $(i, j)$ is the cosine distance between the *leaf* embedding of path $i$ (row) and the *first-hop* embedding of path $j$ (col). The clear diagonal line implies that embeddings within each path are more aligned, reflecting global structure. Analogous plot for Mamba in Figure 13

but as a geometry of highly-organized embeddings. While the former directly stores the adjacency matrix $\mathbf{A}$ in the matrix $\mathbf{W}_{\texttt{assoc}}$ (up to some rotations defined by $\mathbf{\Phi}$), the latter factorizes the associations into $\mathbf{\Phi}_{\texttt{geom}}(u)^T \mathbf{\Phi}_{\texttt{geom}}(v)$. Importantly, whereas an associative memory only makes local information accessible, a geometric memory readily exposes global multi-hop relationships. We define the notion of geometric memory below, and compare the two forms of memory in Table 1. (The definition below intentionally leaves the notion of "multi-hop" abstract, which §4 makes more concrete).

> **Definition 2b.** *(Global geometric parametric memory)* *We say that a deep sequence model $f$ has memorized a graph $\mathcal{G} = (V, E)$ geometrically if for any pairs of vertices $u$ and $v$, the logit $f(u)[v]$ reflects some notion of multi-hop closeness in $\mathcal{G}$. At its simplest, this can be abstracted as $f(u)[v] = \mathbf{\Phi}_{\texttt{geom}}(u) \cdot \mathbf{\Phi}_{\texttt{geom}}(v)$, where $\mathbf{\Phi}_{\texttt{geom}}$ reflects some graph embedding of the vertices. A fuller definition is in §E.5.*

**The resolution.** We view the two parametric memories of Defs. 2a and 2b as two competing data structures, each yielding its own learning complexity. While the associative memory may incur an $\exp(\ell)$ cost to learn an $\ell$-hop lookahead, the geometric memory is powerful enough to drive this down to $\Theta(1)$. Suppose the embeddings of all nodes in path $i$ are clustered tightly around a unique path vector $\mathbf{z}_i$. In this data structure, learning the first token given the leaf node is a one-hop task: simply find a one-hop neighbor of the central node that is best-aligned with $\mathbf{\Phi}_{\texttt{geom}}(v_{\texttt{leaf}}^{(i)})$. A similar structure indeed materializes both in our massive graphs and in tiny graphs; this geometry turns out to be fundamental to any gradient-descent-trained deep sequence model and not specific to attention or residual connections:

**Observation 2.** *(Evidence of global geometry across various deep sequence models) For our large path-star graphs, the (token) embeddings of the leaf and first node of a path are clustered closer to each other than those of other paths in both the Transformer and Mamba (Figures 4 and 13). Similar geometries arise for a variety of small graphs for the Transformer, Mamba and a simple 3-layer neural network model—see visualizations (Figure 1 and §E.3: Figures 16 to 19) and node-node cosine-similarity heatmaps (§E.3; Figures 22 and 23).*

## 3. The memorization puzzle

In this section, we demonstrate that the geometry cannot be easily explained by well-understood learning pressures, be it from the architecture, the optimizer, or the supervision. During training, the two types of parametric memory—two equally valid ways to fit the training data—must compete with each other. Depending on the reader's prior, the rise of the geometric memory may seem familiar especially in light of well-known geometries of high-level concepts and features. While this impression is valid in part, we will carefully isolate aspects that are unexpected.

### 3.1. Supervisory pressure does not explain geometric memory

Geometric representations are known to arise in various tasks with "multi-hop" objectives, from arithmetic (Liu et al., 2022) to path-finding (Khona et al., 2024) to two-hop reasoning (Wang et al., 2024a). Reasoning supervision has also been shown to act as a regularizer that elicits useful structures in the model's storage (Huang et al., 2025). This trail of findings overwhelmingly directs us to the following guess for why we see a geometry:

> **Plausible Explanation 3a.** *(Pressure from global supervision) Perhaps, the model memorizes via a global geometry rather than via local associations due to the multi-hop objective.*

We refute this by ablating the reasoning supervision:

> **Refutation 3a.** *(Geometry arises even from memorizing local, atomic co-occurrences) A global geometry emerges even in models trained only on local edge-memorization, as seen in the heatmaps for the large path-star graph (Figures 5 and 13b). Such models can be subsequently finetuned purely on the hardest-token task and achieve high test accuracy on unseen paths (§F.3). Also, the geometries seen on the small graphs (Figure 1 & §E.3) are for various architectures trained only on edge-memorization.*

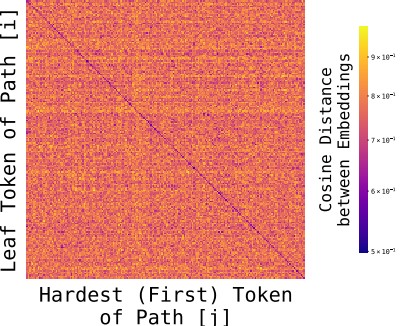

*Figure 5.* **Evidence of global geometry of Transformer in edge-memorization task.** The geometry observed in Figure 4 persists even when we train only on the edges of the path-star task, without path-finding supervision. Mamba shows an even stronger heatmap (Figure 13).

### 3.2. Capacity or optimizer pressures do not explain geometric memory

In theories of generalization, there are various intuitions for why models do not store a lookup table. We scrutinize (straightforward versions of) both these traditional and modern theories as plausible explanations for our geometry, and we refute them one by one. We begin with the traditional notion of explicit capacity pressures:

> **Plausible Explanation 3b.** *(Explicit capacity pressures) Perhaps, the model memorizes geometrically because a lookup of local associations is impossible to represent, either due to the parameter count or the design of the architecture, especially the embedding bottleneck.*

We refute this empirically by showing that associative memory can not only be represented, but be learned by models that memorize geometrically:

> **Refutation 3b.** *(Associative memory can be artificially learned with our architectures) First, a geometry arises even without the pressures of weight decay or dropout (§E.3.1). Next, in various sequence models (Transformers, Mamba, neural networks) and various tiny graphs (Figures 1 and 15), a geometric memory arises naturally even though the same training setup can fit the data associatively, with embeddings frozen and one trainable intermediate layer.*

Maybe then the optimizer has a role to play. For instance, it is known that in classification, gradient descent takes longer to discover a brute-force lookup (Arpit et al., 2017). A similar effect can be proposed here:

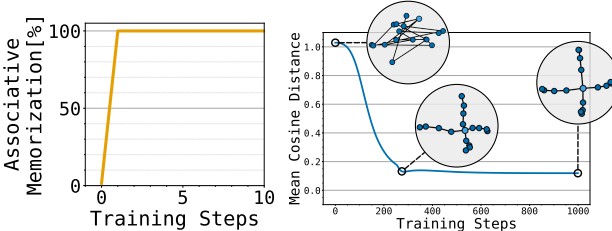

*(a)* Associative Memorization    *(b)* Geometric Memorization

*Figure 6.* **Associative first, geometric later:** Associative memory (**left**) is only two steps away under gradient descent whereas geometric memory (**right**) takes much longer to form for the tiny path-star graph. More details and similar plots for various other graphs and learning rates are in Figures 20 and 21.

> **Plausible Explanation 3c.** *(Optimizational pressure)*
> *Perhaps, the model memorizes geometrically because an associative storage takes longer to find by gradient descent.*

This is already (indirectly) refuted by various prior observations that models "memorize verbatim" well before learning representations that solve downstream tasks (Li et al., 2025; Ye et al., 2025; Wang et al., 2024a). Our intuition is that, intuitively, an associative memory merely stores the adjacency matrix $\mathbf{A}$ immediately available in the data (provably within 1 gradient step—see Nichani et al. (2024)); a geometric memory takes longer since the model has to unearth information implicit in the data. We empirically verify this for the cross-entropy loss on our tiny graphs:

> **Refutation 3c.** *(Associative memory is easier to find with gradient descent)* *Given sufficient embedding dimensionality and capacity, associative storage can be discovered in as little as two steps on all our tiny graphs (Figures 6a and 20) while a geometric memory takes much longer to form (Figures 6b and 21).*

Yet another plausible explanation for the geometry can be derived from modern intuition about generalization (Neyshabur et al., 2015; Zhang et al., 2017): gradient descent implicitly finds succinct fits of the data even without being explicitly told to do so, and thus evades a complex lookup. Indeed, such compressive or complexity-reducing effects have been brought up in recent analyses of geometry in language (Li et al., 2025) and reasoning (Wang et al., 2024a). We translate this into our terminology:

> **Plausible Explanation 3d.** *(Implicit capacity pressure)* *Perhaps, the model memorizes geometrically (over the course of training) because gradient descent converges to a more succinct fit of the data.*

To refute this, we challenge the underlying premise itself. Counterintuitively, a geometric storage is not necessarily more succinct than an associative storage. In a typical language task, which contains redundancies, a lookup table would scale rapidly in complexity with the dataset size while a geometric representation would scale more gracefully; in our *memorization* tasks, this gap becomes a constant or non-existent:

> **Refutation 3d.** *(Geometric and associative memory are equally succinct)* *For sparse graphs where the vertex and edge counts scale similarly (e.g., path-star or cycle), the complexity of associative memory both in terms of bits and $\ell_2$ norms is either equal to or at most twice those of geometric memory. Informal proof and more explanations in §G.2.*

Together, Refutations 3a to 3d lead to our ***memorization puzzle*** in sequence modeling: why does a geometric memory arise instead of an associative lookup even without pressures from the supervision, the architecture, and the optimizer?

## 4. Geometry arises from naturally-occurring spectral bias, without pressures

Setting aside the competition with the associative memory, we can still ask questions about the geometry: where does the global geometry come from, and how does it arise without explicit pressures? Recall that we find geometries not only in models with attention layers, but also in Mamba and in classical deep networks; to focus on the core phenomenon, we study the simplest yet non-trivial multi-layered gradient-descent-trained model: a 2-layer linear network consisting only of an embedding and unembedding layer (weight-tied), trained on the same loss (cross entropy) and the same dataset (of local edges). Equivalently, this is a 1-hop `Node2Vec` model with a 1-layer embedder. Here, associative memory is architecturally prohibited, so we can pursue a clean analysis of the geometry alone.

**Where does the global geometry come from?** Precisely what embeddings are learned even in such simple models is an open question. But a rich line of work—with key assumptions about various pressures outlined shortly—points to a *low-rank spectral bias* of gradient descent: the model learns a low-rank eigendecomposition of the adjacency matrix $\mathbf{A}$. We corroborate this in Appendix E.3, showing that the graph's top eigenvectors indeed match what our 2-layer, `Node2Vec` models learn. Our insight is that these top eigenvectors are the very source of global information: intuitively, these vectors are ones that should dominate the $\ell$-hop adjacency matrix $\mathbf{A}^\ell$.

**Deep sequence models vs. (non-associative) `Node2Vec`-style models.** Two more notable insights come from

comparing the geometries of the `Node2Vec` model against deeper sequence models like the Transformer (e.g., in Figure 1 or Appendix E.3). First, the geometries look alike suggesting that the known spectral bias of `Node2Vec` models is characteristic of all such deep models, regardless of architectural quirks. But strikingly, the `Node2Vec` geometries are much more well-organized than that which the deeper models exhibit (e.g., compare right two columns in Figure 1 or Appendix E.3). Thus, we conjecture a similar spectral geometry in sequence models like the Transformer but—in hindsight—mildly adulterated by the competition with local associative memory. Despite this adulteration, the implications are profound: unlike the two-layer models, the Transformer comes with the power to *reason* over the global geometry, an ability lacking in `Node2Vec`-style models.

**Where does the low-rank spectral bias come from?** While the geometry stems from a low-rank spectral bias, we argue it is not clear where the bias itself comes from. The literature suggests that `Node2Vec` models rely on the top eigenvectors assuming explicit pressures from a narrow bottleneck, regularization, or multi-hop supervision.

**Observation 3.** *(Spectral bias without explicit pressures) A 2-layer `Node2Vec`-style model under cross-entropy loss naturally exhibits a low-rank spectral bias even without explicit pressures (i.e., with large width, no regularization, and 1-hop supervision) as in Figure 7.* [2]

We hypothesize that explicit pressures are required only for the simpler-to-analyze losses studied in literature. In contrast, for the cross-entropy loss, we find that such a spectral bias naturally emerges (much like how such losses induce a max-margin bias (Soudry et al., 2018)):

How does this bias naturally arise? In §H, we provide an empirically-informed intuition (and a partial proof) for how the cross-entropy system gradually filters out lower eigenvectors, reaching a zero-gradient minimum. But we leave open a formal analysis of convergence and whether this can be characterized into a form of max-margin-style bias.

We emphasize that this 2-layer analysis does not divulge anything about the competition between associative and geometric memories that troubles deeper models. The 2-layer analysis illustrates how and what global information can arise naturally, out of local supervision, without any supervisory, bottleneck-driven, or regularizing pressure. These spectral dynamics must be relevant to any gradient-descent-trained deep sequence model.

---

[2] We emphasize that 2-layer models show a particularly strong low-rank behavior not seen in deeper models. For instance, compare the two-layer cycle graph model in Figure 7 with the three-layer ones in §E.3.1.

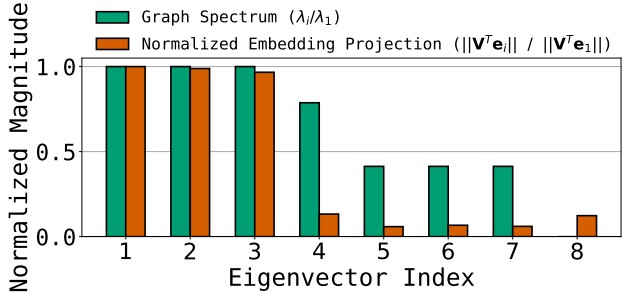

*Figure 7.* **Low-rank spectral bias arises naturally in $2$-layer cross-entropy-trained models even without explicit pressures**. We consider a wide model ($d_{\text{embedding}} = 100$, much larger than nodes in the tiny path-star graph). We report the projection of the embeddings along the eigenvectors (normalized by that of the top eigenvector). Observe that the model is naturally skewed towards the top vectors, in fact even more skewed than the eigenvalues themselves. Further details and plots for other graphs are in §H.5.

## 5. Related Work

We briefly discuss key related lines of work here, deferring details and many other lines to §B.

**Analysis of Transformer memory** Various works have analyzed how neural networks implicitly behave as associative memory stores, such as in autoencoders (Radhakrishnan et al., 2020) or Transformers (Bietti et al., 2023; Cabannes et al., 2024a; Nichani et al., 2024; Geva et al., 2021; Schlag et al., 2021; Sukhbaatar et al., 2019). We emphasize that the associative memory view is sufficient to understand Transformer behavior on disjoint facts, evidenced by the rich empirical literature built on this view. There have also been mechanistic studies of how Transformers perform fact recall (Lv et al., 2024; Geva et al., 2023; Meng et al., 2022) and where memory is stored (Geva et al., 2021). Directly related to us are the works of Khona et al. (2024); Yao et al. (2025); Biran et al. (2024) who interpret multi-hop recall circuits.

**Multi-hop question-answering** A line of work has looked at natural language based two-hop questions on pretrained models, where results here have been limited (Press et al., 2023; Biran et al., 2024) or mixed (Yang et al., 2024; 2025; Balesni et al., 2025). Yao et al. (2025); Wang et al. (2024a) find that these queries require exponential amounts of data, or a curriculum, or very long amounts of training. Studying the gap between these and the symbolic settings is important and left for future work.

**Analysis of `Node2Vec` embedding geometries.** Most earlier analyses are on a simpler, "negative sampling" loss (Levy & Goldberg, 2014), or involve explicit multi-hop objectives (Qiu et al., 2018), low-rank constraints (HaoChen et al., 2021; Tan et al., 2024; Jaffe et al., 2020) or early stopping (Karkada et al., 2025). Other analyses focus on specific types of graphs like stochastic block models (Davi-

son et al., 2024; Harker & Bhaskara, 2024). Only recently has the softmax loss become a focus under assumptions such as freezing one layer (Thrampoulidis, 2024) or a heuristic, regularized proxy (Zhao et al., 2025). Regardless, these establish insightful connections to SVMs which may prove to be useful in understanding the precise closed form solution.

## 6. Conclusion and Open Questions

While the associative view of parametric memory is a simple and highly effective view of neural networks, we demonstrate that a geometric view of storage is necessary to explain the behaviors of deep sequence models. Our findings give rise to various foundational and practical questions.

First, is the memorization puzzle of why models memorize geometrically even when there are no pressures. But second, in the more general, complex graphs, we ask what *notions of graph complexity*—such as its spectrum, connectivity, and size—determine to what extent the model memorizes geometrically vs. associatively. Third, is the practical question of what interventions in the data, architecture, or optimization can make memorization more geometric (if one wants approximate reasoning and creativity) or more associative (if one wants accurate retrieval). Broadly, we also expect our findings to inspire revisiting unstated associative assumptions underlying research on knowledge and memory in language models. We discuss limitations in §A.

**Acknowledgments.** We would like to thank Gaurav Ghosal, Gintare Karolina Dziugaite, George H Chen, Christina Baek, Jacob Springer for valuable feedback on earlier versions of this manuscript. We are also deeply grateful to Omar Salemohamed, Andrej Risteski, Alberto Bietti, Ekdeep Singh Lubana and Andrew Lampinen for many discussions and pointers to related work.

## Impact Statement

This paper presents work whose goal is to improve our understanding of sequence models. There are many potential downstream consequences of work in this field, none of which we feel must be specifically highlighted here.

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

*Table 1.* Comparison of two forms of parametric memory in a deep sequence model, given a dataset of edges from a graph $\mathcal{G} = (V, E)$

| | **Associative Memory** (Def. 2a) | **Geometric Memory** (Def. 2b) |
|---|---|---|
| **Storage mechanism** | A lookup table of pairwise edge associations | A graph embedding of the vertices |
| **Embedding $\Phi$** | Arbitrary (random or orthogonal); acts as keys for lookup; doesn't reveal the associations in itself | Reveals the graph structure |
| **Simplest neural implementation (of the learned similarity between input token, node $u$ and next-token, node $v$, denoted by $f(u)[v]$)** | $f(u)[v] = \mathbf{\Phi}(u)^T \mathbf{W}_{\text{assoc}} \mathbf{\Phi}(v)$ where $$\mathbf{W}_{\text{assoc}} \approx \sum_{(u,v) \in \mathcal{G}} \mathbf{\Phi}(u) \otimes \mathbf{\Phi}(v)$$ and $\mathbf{\Phi}(u) \cdot \mathbf{\Phi}(v) \approx 0$ for $u \neq v$. | $f(u)[v] = \mathbf{\Phi}_{\text{geom}}(u)^T \mathbf{\Phi}_{\text{geom}}(v)$ where $\mathbf{\Phi}_{\text{geom}}$ corresponds to eigenvectors of graph Laplacian (potentially, only the top eigenvectors). |
| **Relationship to graph $\mathcal{G}$** | $\mathbf{W}_{\text{assoc}}$ can be thought of as adjacency matrix (up to a rotation) | $\mathbf{\Phi}_{\text{geom}}$ can be thought of as a *factorization* of adjacency matrix, typically low-rank or skewed towards the top eigenvectors. |
| **Global information** | Only local edge information is readily available (or via a probe). That is, the logit $f(u)[v]$ high only for $(u,v) \in E$ | Global information readily available. Concretely, $f(u)[v]$ is proportional to multi-hop distance between $u$ and $v$ in $\mathcal{G}$ (but if factorization is full rank and not skewed towards the top eigenvectors, global information can still be accessed through a probe that focuses on the top directions). Intuitively, the top eigenvectors contain global information as they are also the components dominant in multi-hop adjacency matrix, $\mathbf{A}^\ell$ for large $\ell$. |
| **Novel information not present in data** | None; only stores seen associations. | Has synthesized novel information by integrating information scattered across multiple datapoints. |
| **Implicit compositional reasoning e.g., path-finding** | Intuitively, exponentially hard to learn. | Approximated as an easy-to-learn geometric navigation task. |
| **Accuracy of retrieval** | Precise | Approximate |
| **Description length** | Number of edges, $|E|$ (upto logarithmic factors) | Number of vertices times the embedding dimensionality, $|V|m$ |

# Appendix

## A. Limitations

1. Our positive result of implicit in-weights reasoning is on a purely symbolic task, and on a specific graph topology (path-star, and tree-star in §E.2). It is unclear how well this generalizes to other topologies, and to graphs of other sizes.

2. Whether our insights extend to natural language is highly non-trivial, since the way the entities are tokenized and the way relationships are presented are much more unstructured.

3. We use small to mid-sized Transformers trained from scratch (`GPT-mid`). We have not explored the effect of large model sizes or of large-scale pretraining.

4. We emphasize that all our arguments are empirical and informal (or left as propositions), and lack a precise, fuller characterization of our observations.

   (a) Perhaps, a slightly more nuanced form of architectural or statistical pressure (e.g., more nuanced norm complexity such as flatness of the loss) may indeed explain why associative memory is less preferred by the model.

(b) Conversely, perhaps the lack of pressures in the learning setup is indeed why the Transformer learns a sub-optimal kind of geometric memory compared to `Node2Vec`.

5. Although we illustrate a clear contrast between associative and geometric memory in their caricatured forms (i.e., as $\mathbf{\Phi}(u)^T \mathbf{W}_{\texttt{assoc}} \mathbf{\Phi}(v)$ vs. $\mathbf{\Phi}_{\texttt{geom}}(u) \cdot \mathbf{\Phi}_{\texttt{geom}}(v)$), it is unclear how to conceptually disentangle these two modes of storage in a given multi-layered deep network.

## B. Related work

Our work consolidates fragments of a nascent phenomenon and brings together distinct lines of theoretical and empirical work on memorization, learning compositional functions, and interpreting model representations. We elaborate on each of these threads below.

### B.1. Analysis of Transformer memory

**Associative memory.** The concept of associative memory dates back to theories of how information is stored in the brain (Longuet-Higgins et al., 1970; Willshaw et al., 1969). These ideas have since been explicitly modeled through various architectures such as Hopfield networks (Hopfield, 1982; Ramsauer et al., 2021), energy-based models (Krotov & Hopfield, 2016), and other modern Transformer-style inventions (Le et al., 2020; Hoover et al., 2023). Closest to us are works that analyze how architectures implicitly behave as associative memory storage, such as in autoencoders (Radhakrishnan et al., 2020) or Transformers (Bietti et al., 2023; Cabannes et al., 2024a; Nichani et al., 2024; Geva et al., 2021; Schlag et al., 2021; Sukhbaatar et al., 2019). Others have established connections between the attention mechanism and Hopfield networks (Ramsauer et al., 2021; Smart et al., 2025). We emphasize that this view is sufficient to understand Transformer behavior on disjoint facts, evidenced by the rich empirical literature built on this view. The geometric view only seems necessary when the facts become interdependent.

**Remark 1.** *In literature, associative memory is abstracted with the assumption that the embeddings are orthogonal as in (Wang et al., 2024b; Ghosal et al., 2024; Jiang et al., 2024a; Zhu et al., 2024; Cabannes et al., 2024b) or random in sufficiently high-dimensions (and thus, near-orthogonal) as in (Nichani et al., 2024; Cabannes et al., 2024a; Bietti et al., 2023; Wang & Sato, 2025).*

**Geometric memory as node representations.** Although geometric memory has not been explicitly labeled as a distinct form of parametric memory, we find scattered observations of it in literature in weaker forms or more limited settings. Jiang et al. (2024a); Khona et al. (2024); Ye et al. (2025); Nishi et al. (2025) show that nodes in graph-like datasets are embedded in a geometric way that encodes distances. These settings, which are on smaller-scale graphs, involve some form of global supervision or latent high-level structure, while we find that such a geometry can arise even with atomic facts. Jiang et al. (2024a) further juxtapose these structured embeddings against associative memory. However, they suggest that the structure becomes relevant only in the underparameterized setting, and that an associative memory suffices as an abstraction in the overparameterized setting; we find this is not the case. Independently, on two-hop datasets, Ye et al. (2025); Wang et al. (2024a) report how models first "memorize" the training data, before "generalizing" on their reasoning task; under the hood, this translates to associative and geometric memorization respectively. Nishi et al. (2025) argue and report that knowledge editing becomes tricky when memorizing cyclic graphs. Indeed, we suspect that a geometric memory poses problems to knowledge editing, since information is not stored loosely. This is an important open problem for future work to address.

**Geometric memory as factorized knowledge.** Recall that the geometric memory can be viewed as storing a low-rank factorization of the adjacency matrix. The advantages of such factorizations is again scattered in literature. Saxe et al. (2019; 2014) prove that a two-layer model naturally learns an SVD factorization of input-output associations. This is a powerful result demonstrating the value of depth in learning hierarchical representations; but the global, multi-hop nature of these representations was not highlighted. Also, unlike our setting, theirs requires explicit pressures to demonstrate a low rank spectral bias (either through a bottleneck or early-stopping) for they study the squared-error loss.[3] More recently, Huang et al. (2025) point out that two-hop reasoning emerges if and only if the key-query matrices in an attention layer are kept separate (as against being collapsed into a single matrix); we show that such favorable effects occur in more general

---

[3]This requires some nuance. Under their squared error loss (applied without a softmax layer), their (weight-untied) fully-trained 2-layered model would learn the 1-hop adjacency matrix perfectly, with no global, multi-hop information revealed in the logits. Although the associations are stored in a factorized form, the final hidden representations do not skew further towards the higher directions than the eigenvalues themselves. For the cross-entropy loss, there is such a skew (Figure 7).

architectures without attention layers. They also highlight the role of an implicit regularization effect from their reasoning supervision; while this may indeed explain the benefits of a reasoning objective, we emphasize that this does not resolve why a geometry arises under pure local supervision—see Remark 2.

We also interpret these prior analyses (Saxe et al., 2019; Huang et al., 2025) as a study of how 2-layered models memorize a special type of graphs: *bipartite* graphs (although they do not express it as such).[4] Importantly, as noted in our own analysis in §4, such 2-layered models do not capture the competition between associative and geometric memory.[5] For that, a study of deeper models is required, where the dynamics along the various spectral components become entangled; to keep them untangled, unnatural assumptions about the initializations need to be made, as shown in (Saxe et al., 2014). Another nuance in these prior 2-layered analyses is the lack of weight-tying, which creates significant differences in what is learned, as we identify in §E.4.

**Remark 2.** *(**Implicit regularization in matrix factorization is an insufficient explanation of geometric memory**) Huang et al. (2025) attribute the success of reasoning to implicit regularization effects that arise in matrix completion tasks; we argue that these effects do not explain why a geometry arises over associative storage in our experiments. The "matrix completion" view in Huang et al. (2025) is that during training the model attempts to fit certain observed entries in a 2-hop adjacency matrix $\mathbf{A}^2$; any test query in effect demands that the model "complete" a missing 2-hop entry. While many minima exist in such underdetermined matrix completion tasks, it is known (Gunasekar et al., 2017; Arora et al., 2019) that gradient descent on a deep factorized model finds the least nuclear norm matrix consistent with the observed entries. However, our edge-memorization-only task is a well-determined task that requires storing the adjacency matrix $\mathbf{A}$ as is, without any missing entries to infer; thus no such implicit pressure must exist. Yet a geometry arises in this task as shown in Refutation 3a. While it is still possible that there is a certain nuclear norm minimization effect even in edge-memorization tasks—perhaps because the logits themselves are not well-determined—the connection is not as obvious as it may seem.*

**Expressive capacity.** Theoretical works have quantified bounds on the expressive capacity of models when it comes to memorizing sequences (Mahdavi et al., 2024; Kajitsuka & Sato, 2025; Madden et al., 2025; Kajitsuka & Sato, 2024; Kim et al., 2023) as opposed to associations between pairs of bigrams. These do not comment on the learning dynamics. These works typically assume that the token embeddings are all well-separated from each other (an assumption that empirically breaks in our setting). In light of the geometric view, it is necessary to restate expressive capacity bounds in terms of geometric capacity of a network and the "geometric complexity" of the dataset.

**Empirical analyses.** Other works (Allen-Zhu & Li, 2025; Morris et al., 2025; Roberts et al., 2020; Pan et al., 2025) have performed careful empirical analyses of scaling laws for memorization, and quantified memorization in terms of "bits per parameter count", known as bit complexity (Vardi et al., 2022), which is related to our notion of bit count in Refutation 3d. All these analyses are worth revisiting with the newfound lens of a geometric memory. Zucchet et al. (2025) empirically analyze the dynamics behind how facts are memorized in a model. Others (Liu et al., 2024; Zhang et al., 2025) have proposed methodological improvements to acquiring knowledge in a Transformer; of relevance to us is the finding in Zhang et al. (2025) that training a model simultaneously on both facts and question-answering is a better way to integrate knowledge into the parameters.

**Mechanistic interpretability.** There have been mechanistic investigations into how Transformers perform fact recall (Lv et al., 2024; Geva et al., 2023) and where facts are stored in a transformer (Geva et al., 2021) and how it can be edited (Zhu et al., 2020; Meng et al., 2022). Similar attempts have been made in traditional classifier networks (Baldock et al., 2021; Maini et al., 2023; Stephenson et al., 2021). Directly related to us are the works of Khona et al. (2024) and Yao et al. (2025); Biran et al. (2024) who perform a mechanistic interpretability analysis of how multi-hop recall works.

### B.2. In-weights reasoning tasks

**Synthetic graph tasks.** Our work consolidates positive results of in-weights reasoning in literature, and presents a stronger instance of it, removing various confounders that make composition-learning easy. Khona et al. (2024) report successful

---

[4]Concretely, Saxe et al. (2019) study mappings between "entity" tokens and binary feature vectors; we translate this as a bipartite graph where "entity" nodes are connected to various "feature" nodes. (Huang et al., 2025) study a mapping between "subject" nodes and "answer" nodes.

[5]At least not in its full glory; in 2-layered weight-tied models, there is only a contrived form of associative memory (Proposition 5); if weight-untied, a natural form can arise in a single step (Proposition 4).

path-finding on 200 nodes-large in-weights graphs, with varying path lengths and test-train overlap. Others (Wang et al., 2024a; Feng et al., 2024; Ye et al., 2025; Huang et al., 2025) report positive results on much shorter 2-hop tasks over a thousand entities or fewer. Geerts et al. (2026) look at in-weights transitive inference, a special type of $\ell$-fold composition query where the model is trained on local comparisons and is queried on more distant comparisons. On settings with 7 objects, Geerts et al. (2026) find a clear difference between an in-context version of their task (where the model struggles) and an in-weights version (where the model succeeds). Our task of finding the first node is a much harder *search* task akin to finding the smallest node in an ordered relationship. Nagarajan et al. (2025) discuss the limitations of next-token prediction, including on open-ended in-weights tasks, in lower data regimes. The fact that their next-token predictor achieves non-trivial performance on their in-weights task could be attributed to the effects of a geometric memory. Tangentially related is the positive finding in Yin & Wang (2025) that the Transformer can compose in-weights knowledge given in-context demonstrations. It is worth noting that the (theoretical) arguments in both these works (Nagarajan et al., 2025; Yin & Wang, 2025) rest on the associative memory view.

As a negative result, Wang et al. (2024b) report that, on in-weights graphs of less than $500$ nodes, models are only able to infer already-seen paths or sub-paths, but not beyond them. We suspect this may stem from the fact that their model is only trained on the paths themselves (while our work and Khona et al. (2024) make the model memorize edge bigrams).

**Multi-hop question-answering.** A line of work has looked at how pretrained models respond to two-hop questions in natural language e.g., "What is the calling code of the birthplace of Frida Kahlo?" (example from Press et al. (2023)). Results here have been limited (Press et al., 2023; Biran et al., 2024) or mixed (Yang et al., 2024; 2025; Balesni et al., 2025). Success has been shown on synthetic knowledge provided there is exponential amounts of data, or a curriculum, or very long amounts of training (Yao et al., 2025; Wang et al., 2024a). Orthogonally, Wang et al. (2025a) identify that such in-weights implicit reasoning can be hurt by scaling up the parameters, perhaps due to "excessive memorization", which we interpret as associative memorization. It is possible that some of these negative results may be attributed to the reversal curse (Berglund et al., 2024; Allen-Zhu & Li, 2023), or the lack of extended computation e.g., we use pause tokens (Goyal et al., 2024). But a dedicated study of this gap between our synthetic settings and these settings is important and left for future work.

**Reversal curse and (a)symmetric knowledge.** The reversal curse (Berglund et al., 2024; Allen-Zhu & Li, 2023) is a well-known out-of-distribution, in-weights failure mode of next-token-trained Transformers. Such models are unable to recall $u$ given $v$, when trained to recall $v$ given $u$, suggesting asymmetric storage in parametric memory. Fixes for the reversal curse have involved reversed or permuted data augmentation (Guo et al., 2024; Lu et al., 2024; Kitouni et al., 2024; Golovneva et al., 2024). In our settings, we find that reverse edges are critical to elicit implicit reasoning in our large path-star tasks (see §F.1), but is not necessary to elicit a geometry in the smaller graphs.[6] Various theories have been proposed to understand this failure (Lin et al., 2024; Zhu et al., 2024; Wang & Sun, 2025), which may be worth revisiting under a geometric view. One may also view our contrast between associative and geometric memory (of a generic graph data) as a generalization of the aforementioned contrast between the asymmetric and symmetric knowledge storage (of a more specific, disjoint set of associations). Future work may discover a nuanced connection between our work and the reversal curse.

### B.3. Failure of end-to-end composition learning

While $\ell$-fold composition functions are surprisingly easy to *express* in transformers (Sanford et al., 2024b), empirical results have time and again demonstrated that they are hard to *learn* through gradient-based methods, both in traditional deep network settings (Shalev-Shwartz & Shashua, 2016; Abbe & Sandon, 2020a; Gülçehre & Bengio, 2016; Glasmachers, 2017; Abbe & Boix-Adsera, 2022; Abbe & Sandon, 2020a) and more recently in language models too (Nye et al., 2021; Ling et al., 2017; Cobbe et al., 2021; Piekos et al., 2021; Zelikman et al., 2022; Recchia, 2021; Cobbe et al., 2021; Hsieh et al., 2023; Shridhar et al., 2022). Others (Bachmann & Nagarajan, 2024; Hu et al., 2025; Shalev-Shwartz & Shashua, 2025) demonstrate how next-token learning can trap training at a stage where composition learning becomes a problem. Theoretical works have attempted to formalize these failures by demonstrating limits due to expressivity (Malach, 2023; Chen et al., 2024; Peng et al., 2024) or sample complexity (Shalev-Shwartz & Shashua, 2016) or computational complexity (Wies et al., 2023; Hu et al., 2025) or in terms of statistical queries (Wang et al., 2025b).

**Remark 3.** *(**Theoretical hardness of learning compositions**) Existing theoretical hardness results typically only show that some worst-case function in a class of compositional functions is hard-to-learn; no proclamations are made about how*

---

[6]It appears that the observations on small graphs in Khona et al. (2024) do not require memorizing the reverse edges, which aligns with our findings on small graphs.

*hard it is to learn a fixed function we care about. Indeed, with contrived initial conditions, a singleton function class can be provably learned (Abbe et al., 2021; Abbe & Sandon, 2020b; Nachum & Yehudayoff, 2020). Yet, in practice, learning a fixed function is hard, proving which has been an open question—until the recent negative results of Abbe et al. (2025); Shoshani & Shamir (2025). These show how the (fixed) full parity function is hard to learn with gradient descent which, through a reduction, proves that in-context composition tasks are hard to learn (Hu et al., 2025). We leave it as an open question to link our in-weights composition task to one of these hardness results.*

### B.4. In-context graph tasks

Graph tasks have been studied extensively in the setting where each context corresponds to a unique graph. We emphasize that this is a very different setting. Indeed, a takeaway from our work is to be deliberate not to conflate insights from the in-context setting with that of the in-weights setting (a distinction that is rarely made explicit in literature).

While Bachmann & Nagarajan (2024) identify the path-star topology as a failure case for next-token learning, Frydenlund (2024; 2025) demarcate the extent of this failure in the same in-context setting, whereas Brinkmann et al. (2024) report positive path-finding results in other graph topologies. Others (Saparov et al., 2024; Sanford et al., 2024a; Yehudai et al., 2025) study Transformers in various in-context graph tasks like search and counting. Connections between in-context graph tasks and spectral biases exist (Cohen et al., 2025; Park et al., 2025a) but should not be confused with the spectral bias in in-weights tasks. While all these works study symbolic graph tasks, other works have empirically identified the limitations on graphs described in natural language (Guo et al., 2023; Wang et al., 2023; Dai et al., 2025). Kim et al. (2022); Ying et al. (2021b) propose algorithmic ideas for encoding graphs as inputs to Transformers. Finally, various failures (Momennejad et al., 2023; Dziri et al., 2024; Valmeekam et al., 2023a;b;c; Shojaee et al., 2025) have been reported on in-context tasks, including planning tasks, framed as word problems.

#### B.4.1. IN-CONTEXT VS. IN-WEIGHTS LEARNING

The dichotomy between drawing information from context vs. drawing information from the weights has been studied in various angles. Some have looked at this from the aspect of two competing circuits relying on one source vs. the other (Chan et al., 2022b;a; Neeman et al., 2023; Cheng et al., 2024). Others have looked at it in the context of the learning paradigms of in-context learning and finetuning the weights (Mosbach et al., 2023; Lampinen et al., 2025). Closer to us, the stark in-context vs. in-weights disparity when it comes to handling global relationships has been emphasized in Wang et al. (2024a); Geerts et al. (2026), for which our results provide further evidence.

### B.5. Analyses of graph and word embedding methods

Much attention has been given to characterizing what embeddings are learned by various contrastive losses such as `Node2Vec`. Most of these are on losses simpler than the cross-entropy loss, with the exception of a recent line of work (Zhao et al., 2025; Zhao & Thrampoulidis, 2025; Thrampoulidis, 2024) which studies the geometry of cross-entropy-trained (weight-untied) `Word2Vec` models. However, certain assumptions are made, such as freezing one of the layers (Thrampoulidis, 2024) or heuristically approximating via a regularized proxy model (Zhao et al., 2025). Curiously, (Zhao et al., 2025) find that their system only directionally converges, while the weights themselves diverge; we however point out that it's reasonable to expect the model to converge to a zero-gradient solution. Regardless, (Zhao et al., 2025) suggest an insightful connection to the converged direction to support vector machines; this may inspire better characterizations of what solution is found by our unregularized `Node2Vec` system.

The connection to a graph spectrum has been made in many other analyses. These analyses focus on a simpler loss called the negative sampling loss where the closed form expression for the inner products is straightforward (namely, the so-called pointwise mutual information (PMI) matrix, as discovered in Levy & Goldberg (2014)). This analysis does not however tell us what exactly the embeddings are. The connection between these embeddings and the graph spectrum has been established in adjacent settings, like with DeepWalk (Qiu et al., 2018) (where the objective is explicitly multi-hop), in low-rank settings like SimCLR (HaoChen et al., 2021; Tan et al., 2024) or with quadratic losses with early stopping (Karkada et al., 2025) or the softmax loss with rank 1 (Jaffe et al., 2020). Other analyses of `Node2Vec` focus on specific types of graphs such as stochastic block models (Davison et al., 2024; Harker & Bhaskara, 2024). We refer the reader to Goyal & Ferrara (2018) for a survey of graph embedding methods. Finally, we clarify that these methods and our insights must not be confused with graph neural networks (Ying et al., 2021a; Zhou et al., 2020), where the graphs are not stored in parametric memory, but presented as input.

**B.6. Other foundational works on generalization and memorization**

**Spectral and simplicity bias.** The type of spectral bias we study in memorization and sequence modeling must be distinguished from the one studied in generalization and traditional classification and regression settings (Rahaman et al., 2019; Xu, 2018; Kalimeris et al., 2019; Arpit et al., 2017; Basri et al., 2019; Bietti & Mairal, 2019). In these earlier studies, the spectrum is that of a continuous function or a decision boundary (e.g., say, the Fourier components of a polynomial), whereas the spectrum we are concerned with is of a discrete, combinatorial object, namely the graph adjacency matrix. This corresponds to a difference in the nature of representations as remarked below.

**Remark 4.** *(On the nature of representations in classification vs. sequence modeling) The geometric representations learned in our setting (i.e., sequence modeling of atomic facts) are of a fundamentally different nature from that learned in the classification/regression setting. The representations in classification/regression correspond to features inherent to each datapoint, such as the edges detected in an image. No such features are inherent to a one-hot token; a token's representation is synthesized from its co-occurrence patterns with other tokens. This difference is central to why the analogy between the generalization puzzle and the memorization puzzle breaks down in §3.2, both in terms of succinctness and ease of discovery with gradient descent.*

**Memorization.** Our work is also orthogonal to the seminal works of Zhang et al. (2017); Neyshabur et al. (2015) who were concerned with classical generalization tasks that possess statistical redundancies. Their argument is that explicit pressures cannot explain why representations arise in such tasks, but implicit pressures may. Our memorization task on the other hand is in sequence modeling, and lacks statistical redundancies; both explicit and implicit pressures do not suffice to explain the geometric representations. Our work is also orthogonal to the foundational work of Feldman (2020) who argue that memorizing the quirks of a training set can be *necessary* for generalization in long tail datasets.

**Linear representation hypothesis.** A long-studied geometric concept in language models is the concept of linear representations in analogies (Park et al., 2024; 2025b; Elhage et al., 2022; Mikolov et al., 2013). The introduction of certain concepts often takes a linear direction, surprisingly, independent of context i.e., going from `cow` to `calf` takes the same direction as `cat` to `kitten`, independent of the source in the context, (`cow`, `cat`). This linear structure, is related, but neither reducible to nor reducible from geometric memorization. Many theories have been proposed to model the linear geometry and semantics of these embeddings (Gittens et al., 2017; Allen & Hospedales, 2019; Ethayarajh et al., 2019; Allen et al., 2019; Hashimoto et al., 2016; Jiang et al., 2024b). These studies are orthogonal since their contribution lies in identifying what structures exist in word-context relationships for such geometries to arise in the embeddings. This is akin to identifying structures in the adjacency matrix, while our analysis is agnostic to such structures. There are many other complementary analyses of these embeddings, both theoretical (Grohe, 2020) and empirical (Mimno & Thompson, 2017; Chen et al., 2021; Chang et al., 2022; Li et al., 2020).

# C. Detailed background on the path-star task

A path-star graph $\mathcal{G} = (V, E)$ is a special tree graph from Bachmann & Nagarajan (2024) that has a central root named $v_{\texttt{root}}$ with multiple disjoint, uniform-length paths emanating from it. One of the leaf nodes is specified as $v_{\texttt{goal}}$. The task is to find the unique path from $v_{\texttt{root}}$ to $v_{\texttt{goal}}$. To solve this task, one may either plan—by searching over the paths and backtracking—or execute a much simpler solution by following the unique path back from $v_{\texttt{goal}}$, and then outputting the reverse of this path. Although this solution is algorithmically straightforward, and although a Transformer can even be shown to learn this solution under a simple multi-token modification to the objective, the next-token objective itself fails to learn this task even with sufficient amounts of data. Thus, the failure comes from optimization under the next-token objective. In this sense, the path-star task is known to be a minimal textbook adversarial example to next-token learning.

**In-context path-star task.** In the in-context version of the task, the model is given the prefix $\boldsymbol{p} = (\texttt{adj}(\mathcal{G}), v_{\texttt{root}}, v_{\texttt{goal}})$ that provides a randomized adjacency list, and indicates the start and goal nodes, in the model's context. The model must then produce the true path as response, $\boldsymbol{r} = (v_{\texttt{root}}, \ldots, v_{\texttt{goal}})$. To cast this as a learning task, we define a distribution $\mathcal{D}_{d,l}$ corresponding to graphs of the same topology, degree $d$ and path length $l$. To sample $\boldsymbol{p} \sim \mathcal{D}_{d,l}$, we uniformly sample node values from a vocabulary $\mathbb{V}$ to create the graph $G$; given this, we can randomize the adjacency list $\texttt{adj}(\mathcal{G})$.

Figures 8 and 9 contrast the two evaluation regimes we use throughout: in-weights (edge memorization & path finetuning) versus in-context (graph in the prompt).

### C.1. Failure of next-token learning in the in-context path-star task

Next-token learners trained on samples from the above task are known to fail *in-distribution* i.e., even on unseen graphs of the *same* topology (with only variations in the node identities and adjacency list randomization), the model uniformly at random picks a path to follow from the center. The mechanism of failure plays out in two stages during training.

**The Clever Hans Cheat.** Early on during training, the model fits the later tokens in the target — concretely nodes $v_2, \ldots, v_{\texttt{goal}}$ — as the unique child of the previous token provided in the input during next-token training. This is known as a *Clever Hans cheat*: the model uses a local rule that relies on witnessing part of the ground truth prefix $(\boldsymbol{p}, \boldsymbol{r}_{<i})$ to predict the next token $r_i$ — as against only using the prefix $\boldsymbol{p}$ to predict the answer tokens. Arguably, this happens because the cheat is much simpler to learn — as an induction head (Olsson et al., 2022) — than the more complex solutions that rely only on $\boldsymbol{p}$ (which involves search-and-backtracking or the simpler lookahead-and-reverse). Simpler predictive features are prioritized early in training (Arpit et al., 2017; Shah et al., 2020; Rosenfeld & Risteski, 2024), which starves gradient signals from more complex features (Pezeshki et al., 2021).

**The Indecipherable Token.** Therefore, in the second stage towards failure, the model attempts to learn the first token $r_i$—the key decision-making token—purely based on information about the graph in the prefix $\boldsymbol{p}$, and without any gradient signal about the later tokens. This is however a computationally hard "needle-in-the-haystack problem" (Wies et al., 2023): the model needs to find a complex end-to-end algorithm with $\ell$ subroutines in it, without any intermediate supervision for those subroutines. The first token thus becomes "indecipherable" and the model simply memorizes it on the training data; during test-time the model subsequently predicts the first token as a random neighbor, and then continues following down the path by applying the local follow-the-child rule it learned through the Clever Hans cheat.

Figure 11 demonstrates these failure modes empirically, showing that next-token prediction achieves only chance-level performance on path-star graphs of varying sizes, with the first token remaining particularly difficult to learn in isolation.

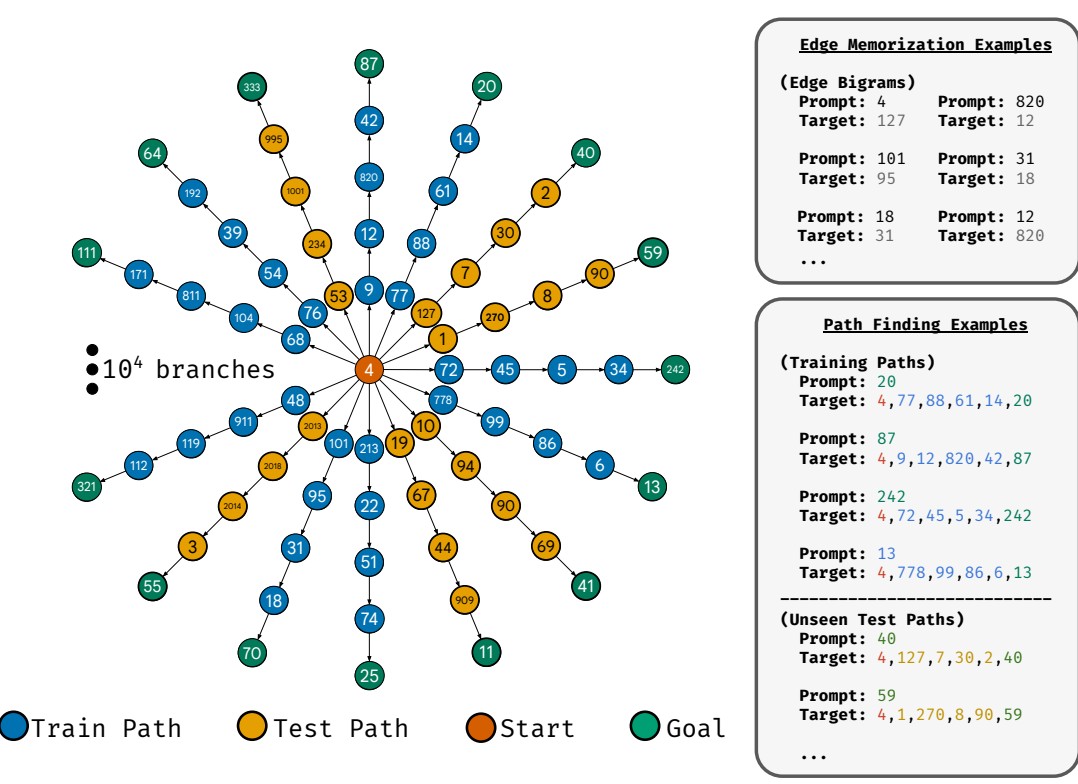

*Figure 8.* **Overview of our in-*weights* path-star task.** All examples are derived from a fixed path-star graph. Training involves two types of examples: (i) **edge memorization** examples; (ii) **path-finding** examples, where the prefix is some leaf, and the target is the full path. Test examples are path examples corresponding to a held-out set of leaves.

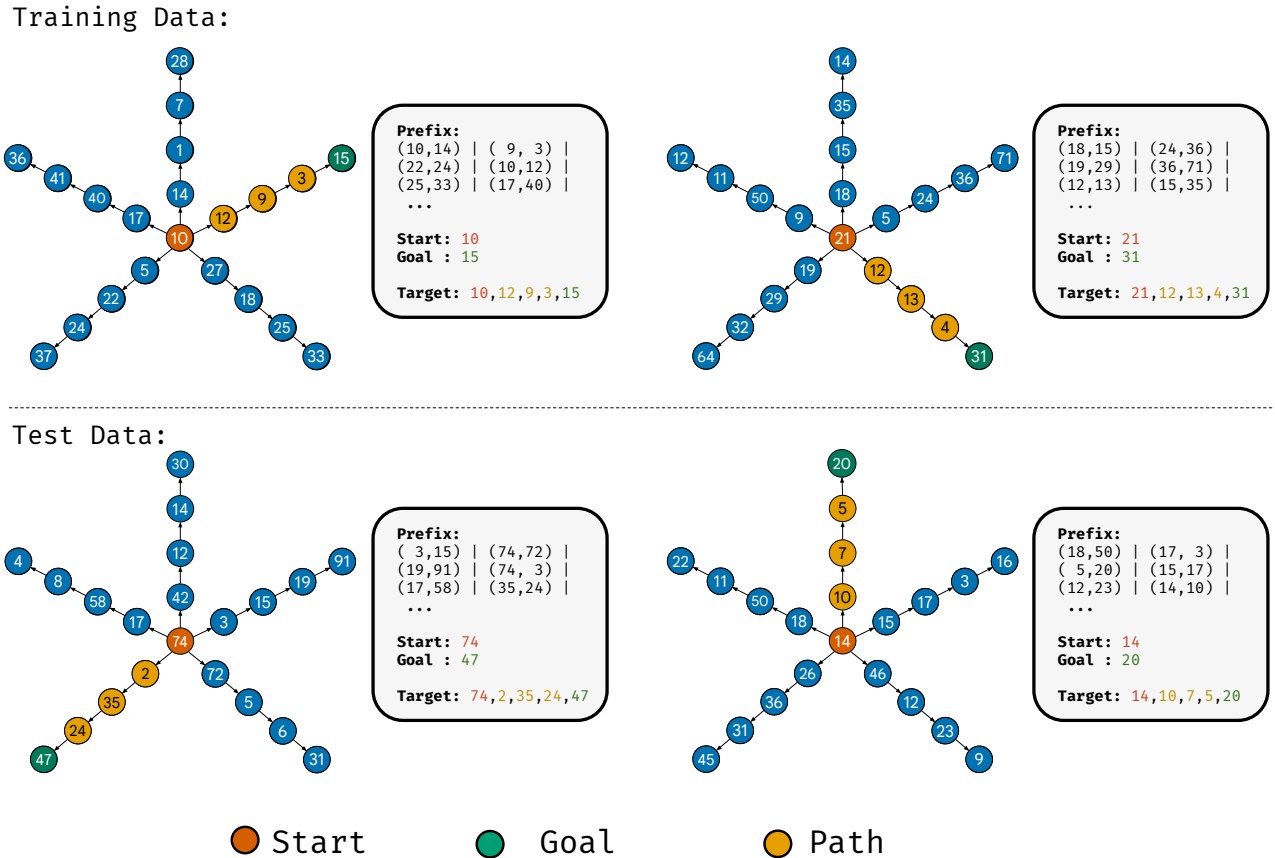

*Figure 9.* **Overview of in-context path-star task of B&N'24.** Each training and test example corresponds to a fresh, randomly-labeled path-star graph (a tree graph where only the root node branches into $d$ paths of length $\ell$). For each example, the prefix specifies a randomized adjacency list (of edge bigrams) of the corresponding graph, followed by $(v_{\text{root}}, v_{\text{goal}})$. The target is the full path $(v_{\text{root}} \rightarrow v_{\text{goal}})$ in that graph.

**Observation 4a.** *(**Success of implicit in-weights reasoning**)* *On in-weights path-star graphs of as many as $5 \times 10^4$ nodes, trained on $75\%$ of the total $10^4$ paths, both the Transformer and Mamba are able to predict unseen paths when conditioned on held-out leaves with as much as 100% accuracy (see left plot of Figure 10 for Transformer), and left plot of Figure 12 for Mamba). Similar positive results for some harder graph topologies are in §E.2.*

Shortly, we will isolate an even stronger instance of implicit reasoning from this task for analysis. To get there, let us scrutinize where the argument of B&N'24 may go differently in the in-weights setting for the model to succeed in Observation 4a.

### C.1.1. WHERE DOES THE PATH-STAR-FAILURE ARGUMENT GO WRONG IN THE IN-WEIGHTS SETTING?

Recall that the path-star failure unfolds in two stages. Perhaps one of these stages does not play out in the in-weights setting. A first possibility could be that the Clever Hans cheat is not picked up here. The cheat was a simple, left-to-right pattern that fits all but the first token as the unique neighbor of the preceding ground-truth token that was present as input. Such left-to-right cheats are simpler than the true right-to-left solution, and are thus quickly learned. However, perhaps when the target we train on is an *in-weights* path, the cheat is not easy to learn—say, due to the nature of recalling from parametric memory. A complex cheat may not be learned quickly, allowing gradients from future tokens to reach the first token representation, in turn allowing the correct, right-to-left solution to compete and emerge. We summarize this hypothesis out below:

**Plausible Explanation 4a.** *(**Model may experience future-token gradients**)* *The in-weights path-star task is solved (Observation 4a) because Clever Hans cheats are not learned quickly enough, allowing future-token gradients to persist, teaching*

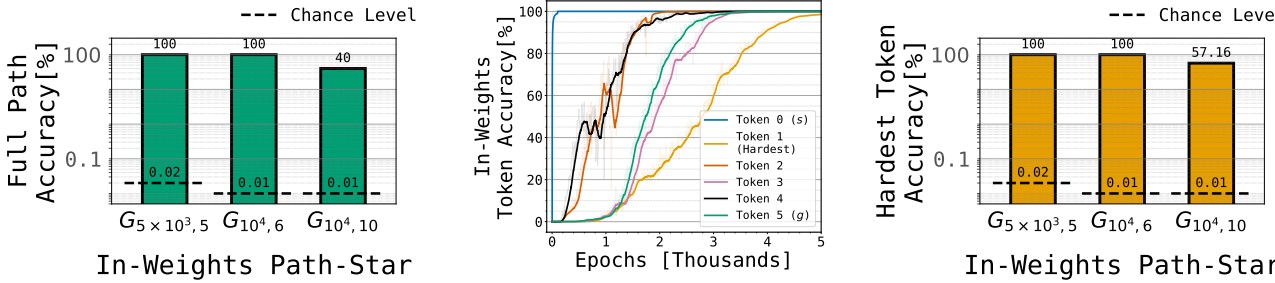

*Figure 10.* **Success of Transformer in in-weights path-star task. (§2) (left)** A next-token-trained Transformer achieves perfect or highly non-trivial accuracy on large path-star graphs $\mathcal{G}_{d,\ell}$. **(middle) Learning order of tokens.** The tokens of a path are not learned in the reverse order i.e., the model does not learn the right-to-left solution. Thus, gradients from the future tokens are not critical for success. **(right) Success of hardest-token-only task.** In fact, the hardest token (the first) given the leaf is learned in isolation to non-trivial accuracy (Observation 1). (This is a duplicate of the same plot in Figure 3)Success of this $\ell$-fold composition task is hard to explain within the associative memory view (§2.3). Analogous plots for Mamba are in Figure 12. We invite the reader to contrast these in-weights task results with the in-context task ones in Figure 11.

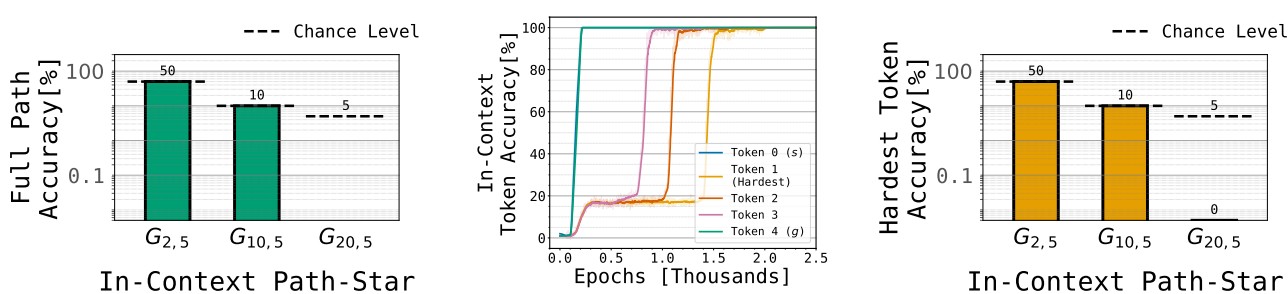

*Figure 11.* **Failure of Transformer in in-context path-star task. (B&N'24)** We report the failure of next-token Transformers in the in-context version of the path-star task, reproducing results from B&N'24. **(left)** Full path accuracy remains at chance level across different small graph sizes. **(middle)** Learning order of tokens with teacherless (multi-token-trained) objective shows a clear right-to-left learning cascade. **(right)** The hardest (first) token given the leaf fails to be learned in isolation, contrasting sharply with in-weights success shown in Figure 10.

*the model to find the right-to-left solution.*

We can indirectly test for this hypothesis as follows. If the model learned the right-to-left dependencies, it must also learn the tokens in the reverse order: the unique predecessor of the goal node is the easiest to identify (under the right-to-left dependencies) and will thus be learned first; the next-easiest is the next predecessor and so on, until the first node (which depends on all that has been learned so far). Indeed, in the in-context setting of B&N'24 where they explicitly switch off the Clever Hans cheat (under teacherless training), such a reverse-learning cascade is observed (Figure 11). However, we do not see this in the in-weights setting.

**Observation 4b.** *(**No reverse-learning cascade**) The target tokens in the in-weights path-star task are learned in no particular order by the Transformer and Mamba models (see middle plots of Figures 10 and 12 in contrast with Figure 11).*

The above observation weakens Expl. 4a, encouraging us to search for another one. A second possibility is that the second stage of the failure of B&N'24 does not trouble us in the in-weights task. Recall that in this stage, the first token is to be learned in isolation without future-token gradients. But this is an $\ell$-fold composition task that is theoretically well-understood to be computationally hard (as we elaborate later). Perhaps, that is not the case for our in-weights task:

**Plausible Explanation 4b.** *(**First-token may be easy to learn**) Learning the key decision-making token (the first token) in the in-weights path-star task is not computationally hard.*

Testing Expl. 4b is easy: we train the model simply on the first token loss, instead of the loss over the full path sequence. We find (as discussed in the main paper) that this task is successful here, affirming Expl. 4b, and isolating a much stronger and cleaner instance of implicit in-weights reasoning.

# D. Experimental setup

## D.1. Graphs, tokens, and data construction

**Path-star graphs.** We denote by $\mathcal{G}_{d,\ell}$ a path-star graph with central node $v_{\texttt{root}}$, degree $d$ (number of arms), and path length $\ell$ per arm. The $i$-th arm consists of the node sequence $(v_{\texttt{root}} = v_0^{(i)}, v_1^{(i)}, \ldots, v_{\ell-1}^{(i)})$ where the last node in the sequence is $v_{\ell-1}^{(i)} = v_{\texttt{leaf}}^{(i)}$.

**Vocabulary.** Each node is represented by a unique numerical token. We reserve several special tokens: [PAUSE] (a compute/pause token), [PAD] (a padding token), optional directional tokens (>, <), and task-specific prefix tokens ([EDGE], [PATH]). The effective vocabulary size is $|\mathbb{V}| = |\text{nodes}| + 9$.

**Edge-memorization datasets for local supervision.** We form directed bigrams $(u, v)$ when $v$ is a child of $u$ to define a distribution $\mathcal{D}_{\texttt{edge}}^{\rightarrow}$. Conversely, we use examples of the form $(v, u)$ where $u$ is the parent of $v$ to define $\mathcal{D}_{\texttt{edge}}^{\leftarrow}$; and $\mathcal{D}_{\texttt{edge}}$ is their union. Training sequences of edge bigrams are simple two-token sequences "$u\ v$" sampled uniformly. All these examples, be it forward or backward, provide only local supervision.

**Path-finding datasets.** We define a path distribution $\mathcal{D}_{\texttt{path}}^{\rightarrow}$, where an example is the pair $(\boldsymbol{p}, \boldsymbol{r})$ with $\boldsymbol{p} = (v_{\texttt{leaf}}^{(i)}, \overbrace{\texttt{[PAUSE]}, \ldots, \texttt{[PAUSE]}}^{P \text{ tokens}})$ and $\boldsymbol{r} = (v_{\texttt{root}}, v_1^{(i)}, \ldots, v_{\ell-1}^{(i)})$. The leaf node is sampled uniformly. We finetune on a subset of leaves and *evaluate on held-out leaves*. Unless stated otherwise, decoding is greedy (top-1). The arrow in $\mathcal{D}_{\texttt{path}}^{\rightarrow}$ denotes that this is the *forward* path; we also experiment with predicting the reverse goal-to-start path, denoted by the distribution $\mathcal{D}_{\texttt{path}}^{\leftarrow}$.

**In-context datasets.** Adapted from B&N'24, the prefix contains a randomized adjacency serialization and $(v_{\texttt{root}}, v_{\texttt{goal}})$; the target is the full path. Where NTP fails in-context, we use the teacherless objective of B&N'24 for comparison plots (details in §D.3).

## D.2. Model architecture

### D.2.1. IN-WEIGHTS PATH-STAR TASK EXPERIMENTS

**Backbone.** The main experiments appearing in §2 use a decoder-only Transformer (GPT-mid) with a causal mask, pre-norm LayerNorm, sinusoidal positional embeddings (Vaswani et al., 2017), GELU MLPs, and tied input/output token embeddings. Additionally, experiments in §E.1 employ a Mamba sequence model (Gu & Dao, 2023) of comparable scale.

**Default Transformer configuration.**

- Layers $N_{\texttt{layer}} = 12$, model width $m_{\texttt{width}} = 384$, heads $m_{\texttt{head}} = 8$.
- Dropout 0 on attention and MLP blocks (synthetic setting), label smoothing 0.
- Context length set to accommodate the longest training sequence (edges: 2; paths: $\ell+1+N_{\texttt{pause}}$) with margin.

**Mamba configuration.** The Mamba models in §E.1 use an equivalent depth and hidden dimension to the Transformer baseline. The sequence model parameters include the state dimension $d_{\text{state}} = 16$, convolution kernel size $d_{\text{conv}} = 4$, and expansion factor expand $= 2$, which follow the standard values from the original Mamba implementation.

**Embedding layer.** Token embeddings are stored in $\mathbf{V} \in \mathbb{R}^{|\mathbb{V}| \times m_{\texttt{width}}}$, with output projection weights tied to $\mathbf{V}$.

**Variants.** For larger graphs $\mathcal{G}_{10^4,10}$, we scaled $m_{\texttt{width}}$ proportionally to 784 in our hyperparameter grid search. For the Mamba architecture, we also varied the expansion factor to 4 and the state dimension to 32.

### D.2.2. TINY MODEL ARCHITECTURES

For toy experiments on small-scale graphs (§E.3), including the *Tiny Path–Star*, *Tiny Grid*, *Tiny Cycle*, and *Tiny Irregular* graphs, we used reduced-size architectures.

**Models.** We evaluated three model types: (1) a Transformer (TinyGPT), (2) a feed-forward *linear* neural network with the same configuration but without residual connection or attention (TinyNN) and (3) a Mamba model (TinySSM) of comparable scale to (1). Note that the TinyNN is initialized the same way as the TinyGPT: all trainable embedding and linear

weights are initialized from $\mathcal{N}(0, 0.02^2)$, while linear biases are initialized to zero. With tied input/output embeddings, the language-model head shares the token embedding matrix and computes output logits via its transpose. Unless otherwise noted, all experiments use this `GPT`-style token embedding initialization. In some analyses (see §E.3.1), we also consider the PyTorch default token embedding initialization, where embeddings are sampled from $\mathcal{N}(0, 1)$. Exact configurations of each of the models mentioned are described below.

**Configuration.**

- For all graphs, all models used a single layer ($N_{\texttt{layer}} = 1$) besides the embedding/unembedding layers, but the model specifications differed across graph types.

For `TinyGPT` and `TinySSM` in particular:

- For all tiny-graph experiments, `TinyGPT` used a single attention head ($m_{\texttt{head}} = 1$) and embedding dimension $m_{\texttt{width}} = 32$.

- For *Tiny Path–Star*, *Tiny Grid*, and *Tiny Cycle*, `TinySSM` used embedding dimension 8 with $d_{\text{state}} = 2$, $d_{\text{conv}} = 2$, and expand = 1. For *Tiny Irregular*, `TinySSM` used embedding dimension 8 with $d_{\text{state}} = 2$, $d_{\text{conv}} = 4$, and expand = 1.

For `TinyNN`, we used a large embedding dimension 512 with 1 head for the sake of empirical proofs of Section 3. We elaborate why. First, `TinyNN` architectures are a clean architecture where one can be certain that freezing the (un)embedding layers results in a purely associative storage; thus they are appropriate for our empirical proofs. Next, in our proofs, we wanted to make two demonstrations. First, that the model memorize geometrically even without bottleneck pressures (as is the case with a large embedding dimension); second, that in sufficiently wide-neck settings, the associative storage is realized within just 2 learning steps (which we find is true here)

We clarify that even when reducing the embedding dimension of `TinyNN` to match that of `TinyGPT` (*Tiny Path–Star* and *Tiny Grid*: $m_{\texttt{width}} = 32$; *Tiny Irregular*: $m_{\texttt{width}} = 16$), we found that the geometric structure still persisted and that the associative memory variant continued to fully memorize the edge bigrams, albeit with longer optimization. In particular, *Tiny Path–Star* and *Tiny Grid* required roughly 500 optimization steps with learning rate 0.01, while *Tiny Irregular* required roughly 1000 optimization steps with learning rate 0.005. For *Tiny Cycle*, we continued to use a wider embedding dimension of 512, which required roughly 10,000 optimization steps at learning rate 0.01. In contrast, in sufficiently wide settings (e.g., embedding dimension 512), the associative storage mechanism is realized within just 2 learning steps, which we find to hold both in these toy settings and in the associative path-star experiments of Section 3..

**Associative memory variants of these architectures.** In associative-memory settings (e.g., left-column visualizations in Figure 1), token embeddings were frozen in all these architectures. All models employed weight tying between input and output embeddings. Note that since we employ only one trainable layer in these settings (the embedding/unembedding layers frozen), we can be certain, at least in the case of `TinyNN` that storage is associative.

**Two-layer (`Node2Vec`-style) architectures.** For the shallow `Node2Vec`-style archirtectures (two weight-tied layers) used in §4, we use wide models with embedding dimension $m = 100$ to ensure that the models have no explicit bottleneck/rank pressure. We train these models with a learning rate of 0.01, a batch size equal to the number of nodes in the graph, and 2000 steps, which is well beyond fitting the training data to full accuracy.

### D.3. Training and optimization

**Objective.** We use next-token cross-entropy over the causal prefix. For first-token-only experiments, the loss is restricted to the first target position. Although for large graph experiments we train all our models to 50,000 epochs, we *only need about a couple of thousand epochs* to see accuracy gains (e.g., see the per-token accuracy plots in Figure 10).

**Optimizer and schedule.** We use the `AdamW` optimizer with a weight decay of 0.01. The learning rate follows a cosine decay schedule with a linear warm-up. In the two-phased edge-memorization ablation experiment of §F.3, the peak learning rate for edge memorization (Phase 1) is $1 \times 10^{-2}$ and for path finetuning (Phase 2) is $5 \times 10^{-5}$.

**Tiny-graph optimization details.** For the tiny-graph experiments in §E.3, optimization hyperparameters varied by graph type. For *Tiny Path–Star* and *Tiny Grid*, both `TinyGPT` and `TinyNN` were trained for 500 optimization steps with learning

rate 0.01. For *Tiny Cycle*, `TinyGPT` was trained for 700 steps with learning rate 0.01, while `TinyNN` used embedding dimension 512 and required 10,000 optimization steps at learning rate 0.01. For *Tiny Irregular*, `TinyGPT` was trained for 2000 steps with learning rate 0.01, while `TinyNN` used embedding dimension 16, learning rate 0.005, and 1000 optimization steps.

For all experiments in Appendix E.3.1, we use a fixed training budget of 10,000 optimization steps across all settings to ensure comparability between regularization strategies.

**Batching.** We used a range of different batch sizes of $\{64, 128, 256, 512, 1024\}$.

**PAUSE tokens.** We append $N_{\texttt{pause}} \in \{0, 2, 4, 6, 10\}$ pauses in $\mathcal{D}_{\texttt{path}}^{\rightarrow}$ to provide compute budget (no labels on pause positions). The chosen $N_{\texttt{pause}}$ for each $\mathcal{G}_{d,\ell}$ is given in Table 2.

*Table 2.* Default hyperparameters by graph size. Values denote the settings used.

| Graph $\mathcal{G}_{d,\ell}$ | $N_{\texttt{layer}}$ | $m_{\texttt{width}}$ | $m_{\texttt{head}}$ | $N_{\texttt{pause}}$ | Peak LR |
|---|---|---|---|---|---|
| $\mathcal{G}_{5 \times 10^3, 5}$ (In-Weights) | 12 | 384 | 8 | 5 | $5 \times 10^{-5}$ |
| $\mathcal{G}_{10^4, 6}$ (In-Weights) | 12 | 384 | 8 | 6 | $5 \times 10^{-5}$ |
| $\mathcal{G}_{10^4, 10}$ (In-Weights) | 12 | 784 | 8 | 10 | $5 \times 10^{-5}$ |
| $\mathcal{G}_{2,5}$ (In-Context) | 12 | 384 | 8 | n/a | $1 \times 10^{-4}$ |
| $\mathcal{G}_{10,5}$ (In-Context) | 12 | 384 | 8 | n/a | $1 \times 10^{-4}$ |
| $\mathcal{G}_{20,5}$ (In-Context) | 12 | 784 | 8 | n/a | $1 \times 10^{-4}$ |

## D.4. Path-finding evaluation

**Forward vs. reverse.** We evaluate forward generation ($v_{\texttt{root}} \rightarrow v_{\texttt{leaf}}$) and reverse generation ($v_{\texttt{leaf}} \rightarrow v_{\texttt{root}}$). Reverse is algorithmically trivial after edge memorization; forward is the non-trivial case we care about.

**Metrics.**

- *Exact-match path accuracy* / *Full path accuracy*: fraction of held-out leaves whose entire path is generated correctly.

- *First-token accuracy* / *Hardest token accuracy*: accuracy of the first hop (hardest token) given the leaf.

- *Per-token accuracy over epochs* / *Token accuracy*: accuracy at each target position, tracked through training.

**Baselines.** Random choice among $d$ branches gives $1/d$ first-token accuracy and near-zero exact-match.

## D.5. Tests for (global) geometric vs associative storage

Path-finding is one way to test for the presence of a global geometry. But in our smaller models, we have used various simpler tests. These tests detect whether the embedding matrix indeed has information about the graph or whether it is arbitrary (if it is, then we know that the storage is associative at least in 3-layered `TinyNN` models).

**Remark 5.** *(**Global vs. local geometry**) If there is a geometry in the embeddings, we can ask whether it is global or relatively local e.g., is the embedding only close to its immediate neighbors but orthogonal to the rest?[7]). As a concrete example, consider the vertices of a path graph embedded along the edges of a hypercube as $(1, 1, 0, 0, \ldots), (0, 1, 1, 0, \ldots), (0, 0, 1, 1, \ldots)$ and so on. Such a solution encodes a local geometry in that each vertex has high cosine-similarity to its neighbors, but to no other vertices. On this regard, each of our tests comes with its own false positive or negative as discussed below. We use the sum total of all tests to ascertain the presence of global geometries as against local geometries.*

1. **3D visualizations**: We found 3D visualizations like in Figure 1 to help track many qualitative behaviors e.g., whether the embeddings are arbitrary, the presence of negative eigendirections in the form of zigzags, the varying quality of geometries under various interventions etc., However, we warn the reader that these visualizations can suffer from a false positive on the local/global test: models with local geometries could still display a global geometry in the 3D visualization since the visualization method may inject some global information into it.

---

[7]We wish to thank Omar Salemohamed for suggesting the possibility of such local geometries.

2. **Cosine heatmaps**: In plots such as Figures 4 and 22, we provide heatmaps of the cosine distances between embeddings. If the heatmap resembles multi-hop adjacency, it affirms a global geometry; if it appears random, the embedding matrix has no useful information. However this can yield false negatives: if the heatmap resembles a spiky one-hop adjacency matrix, the geometry may still be global (which is possible if the embeddings are a full-rank factorization of the adjacency matrix).

3. **Eigenvector projections**: We also provide projections of the embeddings onto the eigenvectors of the graph, like in Figure 7. A random embedding matrix would be roughly equally distributed along all directions, whereas a more geometric embedding would reflect the underlying graph spectrum. In fact, a global geometry would correspond to a spectrum that is even more skewed towards the topmost directions (e.g., uses only the top two directions in the cycle). Our plots are designed as follows. We index the eigenvectors for the negative graph Laplacian by their *signed* eigenvalues. We ignore the top eigenvector which is a degenerate eigenvector that assigns near-uniform values to all nodes. Then, for various indices $i$, we plot the magnitude of $\|\mathbf{V}(t)^T \mathbf{e}_i\|/\|\mathbf{V}(t)^T \mathbf{e}_1\|$, which is the projection of the embeddings along the $i$'th direction relative to the projection along the second-top direction. We compare the spread of these relative magnitudes against that of the eigenvalues themselves as $\lambda_i/\lambda_1$.

4. **Geometric margin**: As a more quantitative test for geometries, for our 3-layered `TinyNN` model, we compute the fit of a modified model that is purely geometric. We do this by eliminating all cross-eigenvector connections in the middle layer $\mathbf{W}$. Specifically, if $\hat{\mathbf{V}}$ are the right-singular directions of the learned embedding matrix $\hat{\mathbf{\Phi}}$, we replace the middle layer with $\hat{\mathbf{V}}^T \texttt{diag}(\hat{\mathbf{V}}\mathbf{W}\hat{\mathbf{V}}^T)\hat{\mathbf{V}}$. We then compute the *margin* for this modified model on the training set as $\min_u\{\min_{w\in\texttt{nbr}(u)} f(u)[w] - \max_{v\notin\texttt{nbr}(u)} f(u)[v]\}$. If this is positive, it means that the embedding matrices encode the graph structure through cosine-similarity alone (modulo some sign and magnitudes prescribed by the diagonal values in the intermediate layer). If the geometric margin is negative however, it may be a false negative because there may be different diagonal values that result in a fit of the data.

## D.6. Implementation and compute

Code is implemented in `PyTorch` with standard Transformer components. All runs fit on a single modern GPU (e.g., `A100-40GB` or `A100-80GB` for the largest batch-size); per-figure training time depends on $(d, \ell)$ and batch size. Code for reproducing the experiments is available at: `https://github.com/shahriarnz14/Geometric_Memory`.

# E. Experiments on broader settings

In this section, we demonstrate that our findings generalize to various other deep sequence architectures and various other large and smaller graphs.

## E.1. Path-star task on Mamba `SSM`

This section (Figures 12 and 13) demonstrates that the implicit reasoning on path-star graphs observed for Transformers in §2.1, generalizes to the Mamba `SSM` architecture (Gu & Dao, 2023) too. A notable difference here is that the Mamba model presents a strong geometry even when trained only on the edges (Figure 13b).

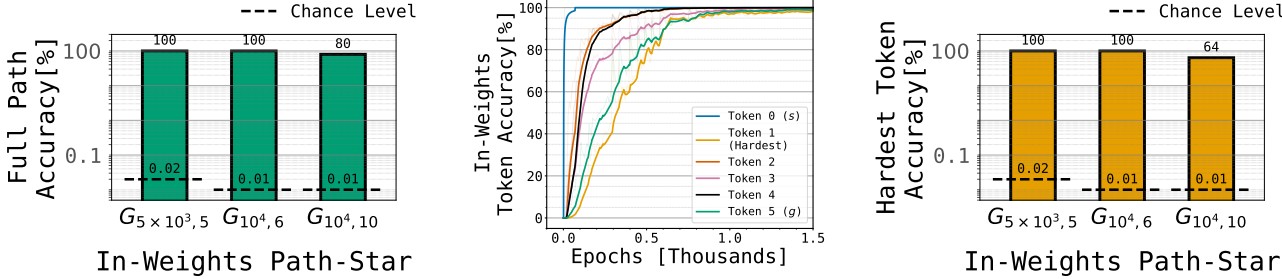

*Figure 12.* **(left) Success of in-weights path-star task for *Mamba*.** This figure is a counterpart to Figure 10. A next-token-trained Mamba achieves perfect or highly non-trivial accuracy on large path-star graphs $\mathcal{G}_{d,\ell}$ (Observation 4a). **(middle) Learning order of tokens.** The tokens of a path are not learned in the reverse order i.e., the model does not learn the right-to-left solution. Thus, gradients from the future tokens are not critical for success (Observation 4b). **(right) Success of hardest-token-only task.** In fact, the hardest token (the first token) given the leaf is learned in isolation to non-trivial accuracy (Observation 1). Success of this $\ell$-fold composition task is hard to explain within the associative memory (§2.3).

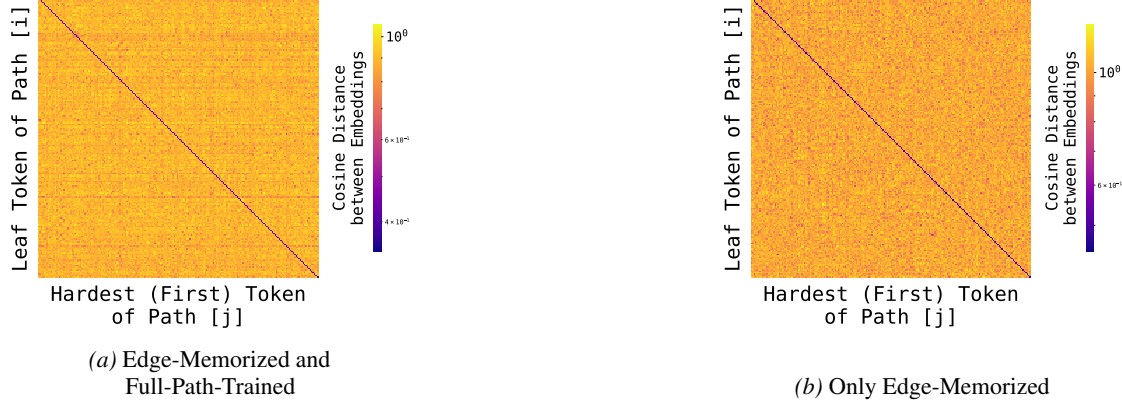

*(a)* Edge-Memorized and
Full-Path-Trained

*(b)* Only Edge-Memorized

*Figure 13.* **Evidence of global geometry in path-star task for *Mamba*.** This figure is a Mamba counterpart to the Transformer heatmaps in Figures 4 and 5. Recall that entry $(i, j)$ is the mean cosine distance between the leaf token in (an unseen) path $i$ (row) and first/hardest token on (an unseen) path $j$ (col). **Left** is trained on edges and path-finding task $(\mathcal{D}_{\text{edge}} \cup \mathcal{D}_{\text{path}}^{\rightarrow})$; **right** on edges only $(\mathcal{D}_{\text{edge}})$. We find that even on these unseen paths, the leaf and first token embeddings cluster together, regardless of whether path-finding supervision exists. Interestingly, compared to the Transformer in Figure 5, the Mamba trained only on local supervision (right) exhibits a much stronger geometry.

### E.2. Other large, harder path-finding graphs

Next, we consider variants of the path-finding graph. While the path-star graph is adversarially constructed in certain ways, there is only one decision-making step, which makes planning simpler in a certain way. To make the task harder, we introduce a branching at every node along each path. (Similar tree variants have been considered in (Frydenlund, 2025) for the in-context task, but we are interested in in-weights tasks.)

In the *tree*-star graph, $\mathcal{T}_{d,\ell}$, there is a central node with degree $d$ and each child node except the leaf has 2 children, as visualized in Figure 14a. The path-length from the root to any leaf is $\ell$. In this graph, two types of learning tasks can be considered, depending on how we split the test and training paths. For a no-overlap setting, we could sample all training paths from one subset of trees, and the test paths from the remaining trees; we call this the *split at first token* setting, since the test/train split is determined by the first token. A second setting—with some test-train overlap—is one where we reserve some leaves as training goals, and the rest as test-goals. Here, on any test path, a prefix may have participated in a training path. We call this the *split at leaf* setting. Note that in both variants, all nodes will be sampled as part of the edge-memorization task.

In Figure 14b, we find that on both variants the Transformer achieves non-trivial path-finding accuracy, generalizing our results beyond the path-star task. However, we note that the test-train split at the first token is much harder to succeed at, likely due to no overlaps.

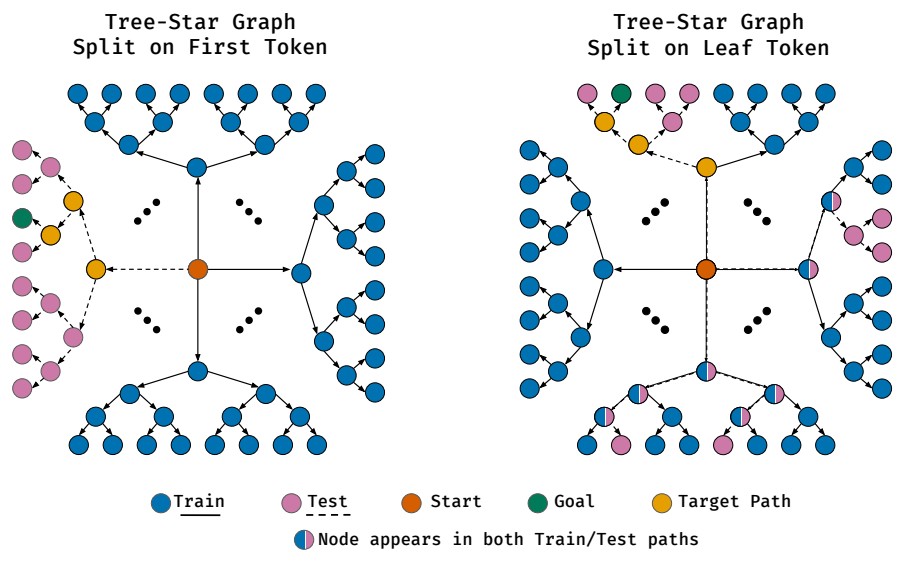

*(a)* In-Weights Tree-Star

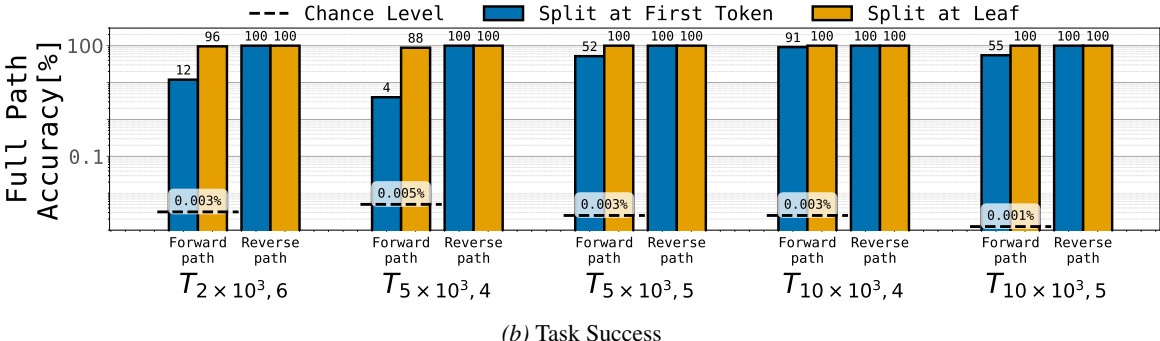

*(b)* Task Success

*Figure 14.* **Transformer achieves non-trivial accuracy on the harder in-weights tree-star task.** The tree-star task of §E.2 introduces decision-making at every step of the path, not just the first token. There are two variants of this task based on the test-train split. In the *split on first token* variant **(top-left)**, we reserve some of the trees for generating training paths, and the rest for test paths. In the *split on leaf token* variant **(top-right)**, we reserve some leaves for training and the rest for testing; some nodes may be sampled for path-finding during both test and train time. In both tasks, we have visualized a single target path. In the **bottom** figure, we report non-trivial path-finding accuracies on both tasks, above random chance defined as $1/\texttt{num\_leaves}$.

### E.3. Tiny graphs

**Graphs** Besides the large path-star graph (of §2) and tree-star graphs (of E.2), we also report the embeddings learned on four tinier graphs for various architectures (details in D.2.2); the figures here are an extension of the Transformer embeddings in Figure 1. In these experiments, we train the models purely on local supervision (the edges, presented in both directions) to $100\%$ edge-memorization accuracy: for each vertex, we ensure that its $d$ neighbors appear in the top $d$ softmax probabilities. The graphs include (a) a tiny path-star graph with four paths of length $4$, (b) a $4 \times 4$ grid graph, (c) a $15$-node cycle graph and (d) an irregular graph with two asymmetric components.

**List of figures.** Our visualization techniques are based on the tests described in §D.5. Our first set of figures (Figures 16 to 19) are 3D visualizations of the embeddings from various architectures: an associative memory model (implemented with a neural network with only one trainable matrix sandwiched between (un)embedding layers), a Node2Vec model, the eigenvectors of the graph Laplacian, a Transformer's token embeddings, a neural network's first layer, and a Mamba SSM's token embeddings. For the Node2Vec model we use the top eigenvectors, and for the rest we use UMAP (McInnes et al., 2018) to choose the top directions. As noted in §D.5, these visualizations by themselves do *not* guarantee the presence of global information. As a rigorous test, in Figures 22 and 23, we provide node-node cosine-similarity heatmaps exhibiting multi-hop information. Next in Figures 20 and 21, we present detailed plots of how fast a model stores associatively/geometrically over training time. In §E.3.1, we provide plots demonstrating the effect of various optimization hyperparameters. Finally, in §E.4, we explain the presence of zigzagging directions in some settings, and the effect of weight-tying.

**Consolidated Observations.** For easy reference, we consolidate all our observations from these plots below:

**Observation 5.** *In the tiny graphs of Figures 16 to 22 and 23 and §E.3.1 on various architectures (Node2Vec, graph spectrum, Transformer, neural network, Mamba SSM), we find that:*

1. *A geometry arises in all these architectures even without global supervision from a path-finding task (Refutation 3a).*

2. *The global information in these geometries can be traced back to the eigenvectors of the graph spectrum (§4).*

3. *The geometry arises in all three deep sequence models (Transformer, Mamba SSM, neural network) even though these models can learn the data associatively using the same learning setup, with just the (un)embedding matrices frozen (Refutation 3b).*

4. *While the geometry is significantly aided by higher weight decay or dropout, these regularizing effects are not necessary (Refutation 3b); (demonstrated in §E.3.1).*

5. *The geometry of the Node2Vec model (which precludes associative memory), is much stronger than the deep sequence models, suggesting that the deep sequence models may be adulterated with associative memory .*

6. *Associative storage can be quickly found by gradient descent (Figure 20), as claimed in Refutation 3c, whereas geometric memorization takes longer to discover (Figure 21)*

7. *Geometries arise even with only one direction of the edges presented (Figure 31).*

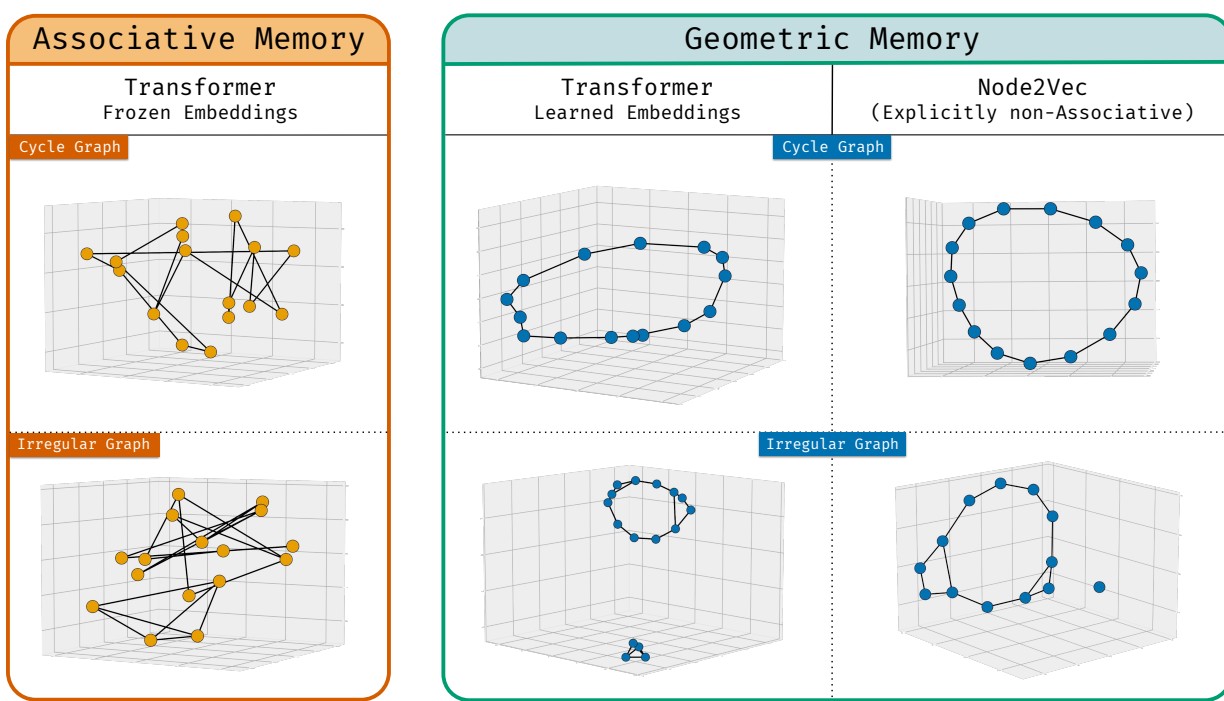

*Figure 15.* **Quality of geometric memory across graph types.** Extending Figure 1, we show that the contrast between associative memory (left), Transformer geometric memory (middle), and `Node2Vec` geometric memory (right) holds across different graph topologies. `Node2Vec` models, where associative memory is architecturally prohibited, consistently exhibit cleaner and more structured geometries than Transformers. This points to significant headroom for improving the geometric nature of Transformer memory. Details of the Transformer architecture used for this visualization are provided in §D.2.2.

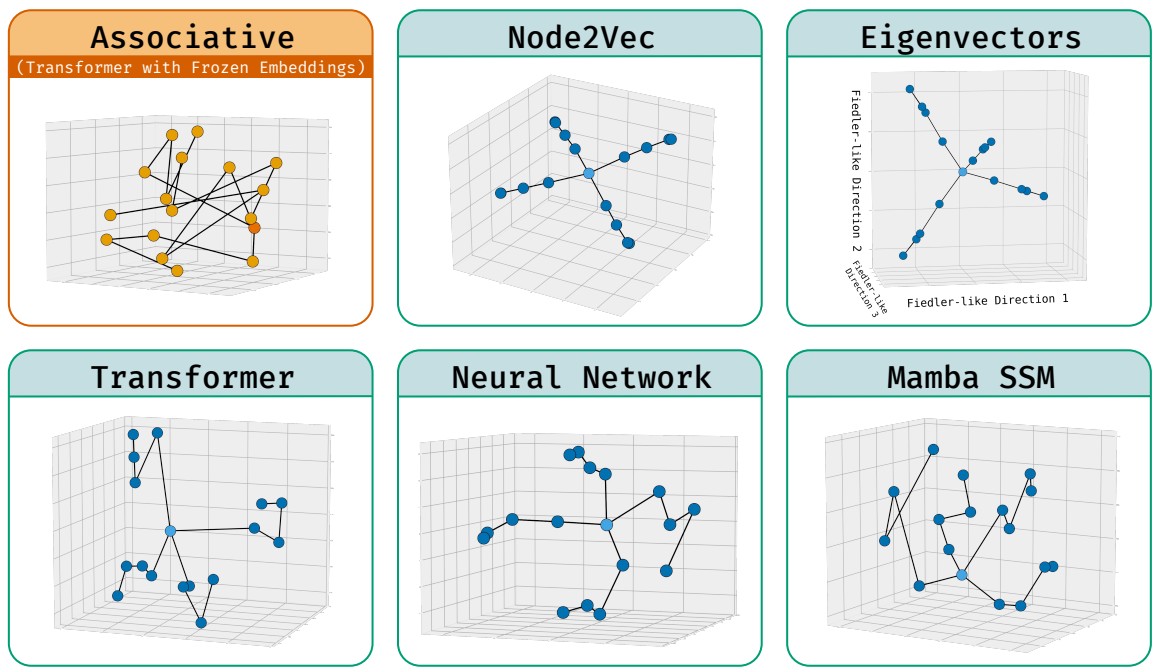

*Figure 16.* **Tiny path-star**: Geometries of various architectures on a smaller version of the path-star graph. See Observation 5.

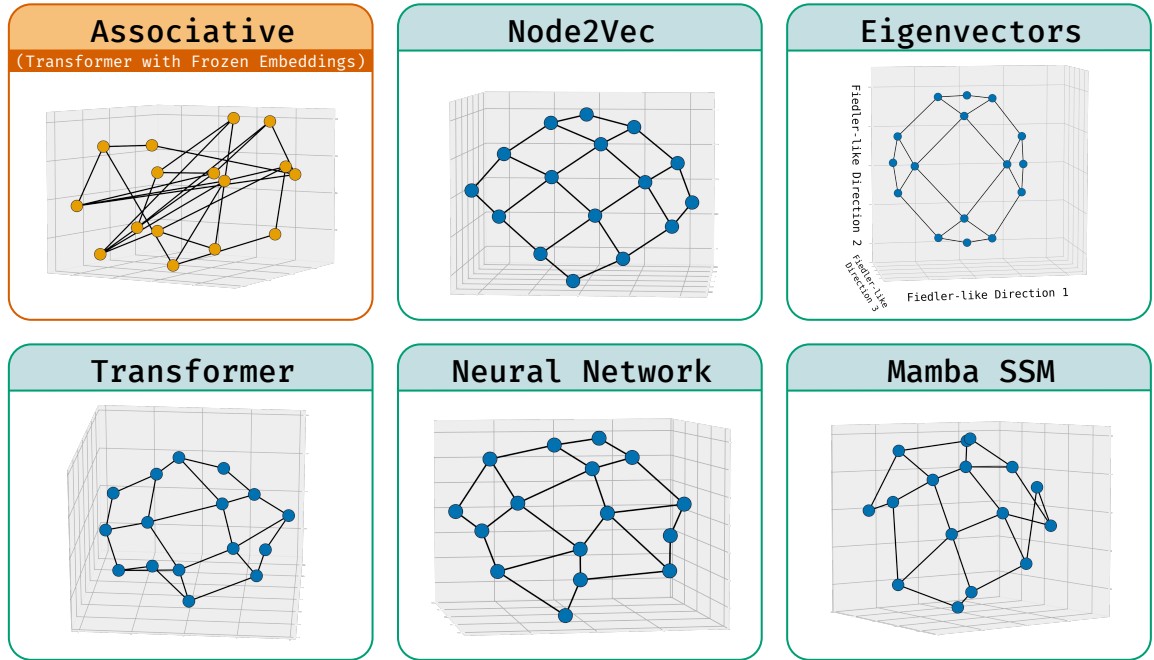

*Figure 17.* **Tiny grid**: Geometries of various architectures on a small $4 \times 4$ grid graph. See Observation 5.

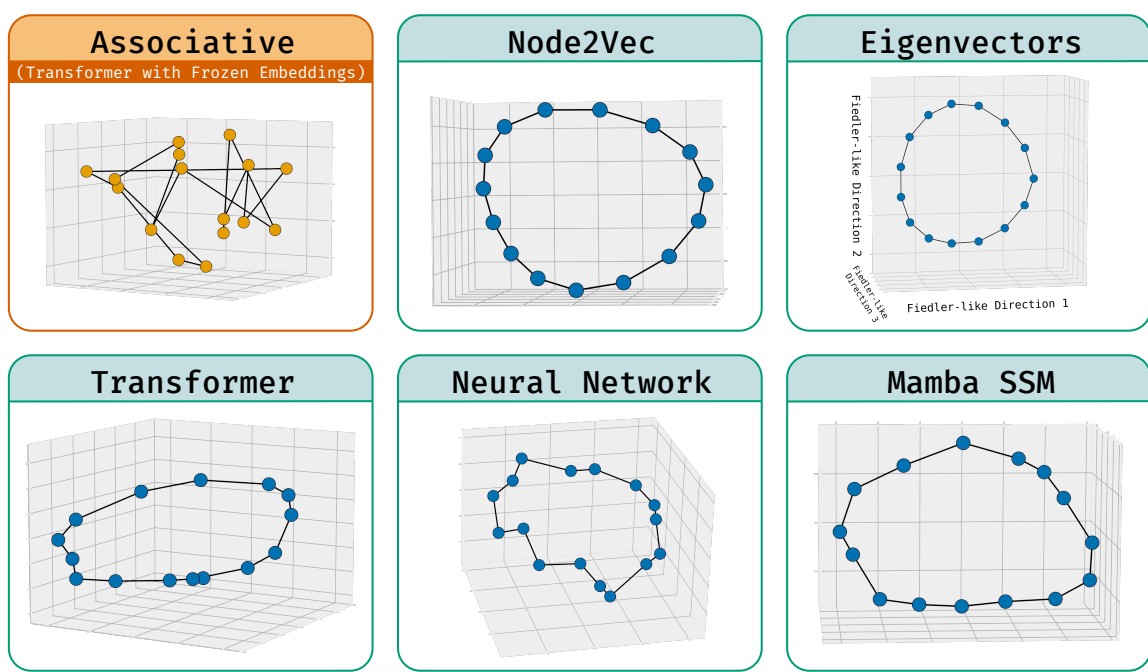

*Figure 18.* **Tiny cycle**: Geometries of various architectures on a small cycle graph. See Observation 5.

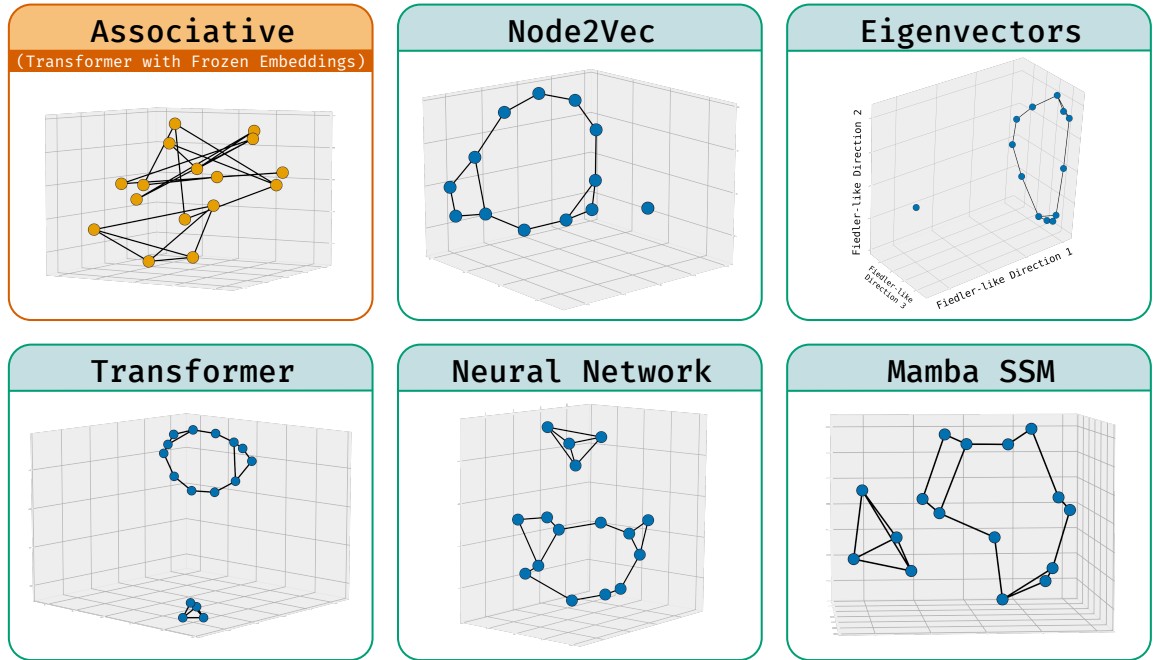

*Figure 19.* **Tiny irregular graph**: Geometries of various architectures on a small irregular graph of two connected components, both asymmetric. See Observation 5.

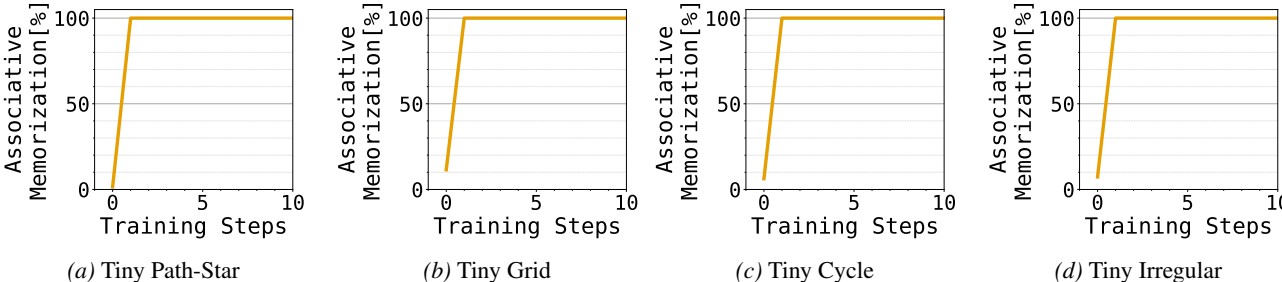

*(a)* Tiny Path-Star        *(b)* Tiny Grid        *(c)* Tiny Cycle        *(d)* Tiny Irregular

*Figure 20.* **Associative memory can be discovered quickly by gradient descent, given a sufficiently wide model, and a sufficiently large learning rate**: For the various tiny graphs described in §E.3, and for our `TinyNN` model (with frozen embedding/unembedding layers and one wide trainable weight matrix to prevent a geometry from taking over; see §D.2.2), we report memorization over timesteps of training with *full-batch* gradient descent and constant learning rate of $0.1$. For each vertex, we compute what fraction of its $k$ neighbors appear in the top $k$ next token probabilities; this is averaged across vertices (where $k$ varies across vertices). This value, which quantifies associative memorization, reaches its maximum within $2$ steps of gradient descent; compare this to the longer time for geometric memory to form in Figure 21. This is evidence for our Refutation 3c: ease-of-discovery does *not* dictate what memory is formed.

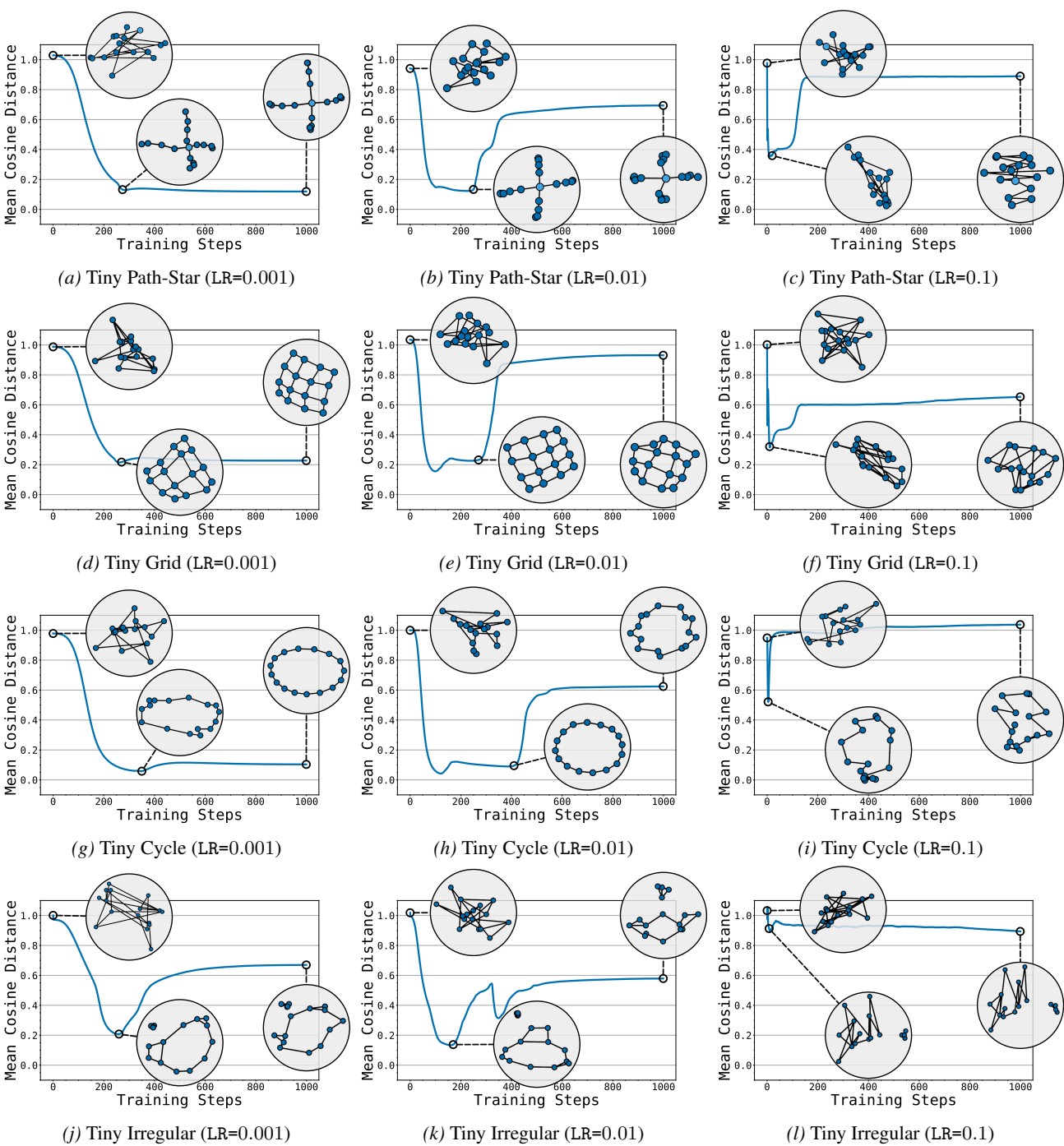

*(a)* Tiny Path-Star (`LR=0.001`)

*(b)* Tiny Path-Star (`LR=0.01`)

*(c)* Tiny Path-Star (`LR=0.1`)

*(d)* Tiny Grid (`LR=0.001`)

*(e)* Tiny Grid (`LR=0.01`)

*(f)* Tiny Grid (`LR=0.1`)

*(g)* Tiny Cycle (`LR=0.001`)

*(h)* Tiny Cycle (`LR=0.01`)

*(i)* Tiny Cycle (`LR=0.1`)

*(j)* Tiny Irregular (`LR=0.001`)

*(k)* Tiny Irregular (`LR=0.01`)

*(l)* Tiny Irregular (`LR=0.1`)

*Figure 21.* **For various learning rates, geometric memorization takes much longer for gradient descent to discover**: This is an extended version of Figure 6b for two more learning rates and all our tiny graphs. Recall that for the same `TinyNN` architecture as in Figure 20 but without frozen embeddings, we roughly quantify geometric memorization over time via cosine distance across *all* pairs of vertices, alongside 2D visualizations. Each row is a different graph, and each column is a different learning rate. For both the learning rate 0.001 and 0.01, the geometries take about 200 steps to appear; the larger learning rate of 0.1 is too aggressive to create the geometry. Recall that, for associative memory, two steps with learning rate of 0.1 sufficed (Figure 20). This is evidence for Refutation 3c: ease of discovery does not determine what sort of memory is preferred.

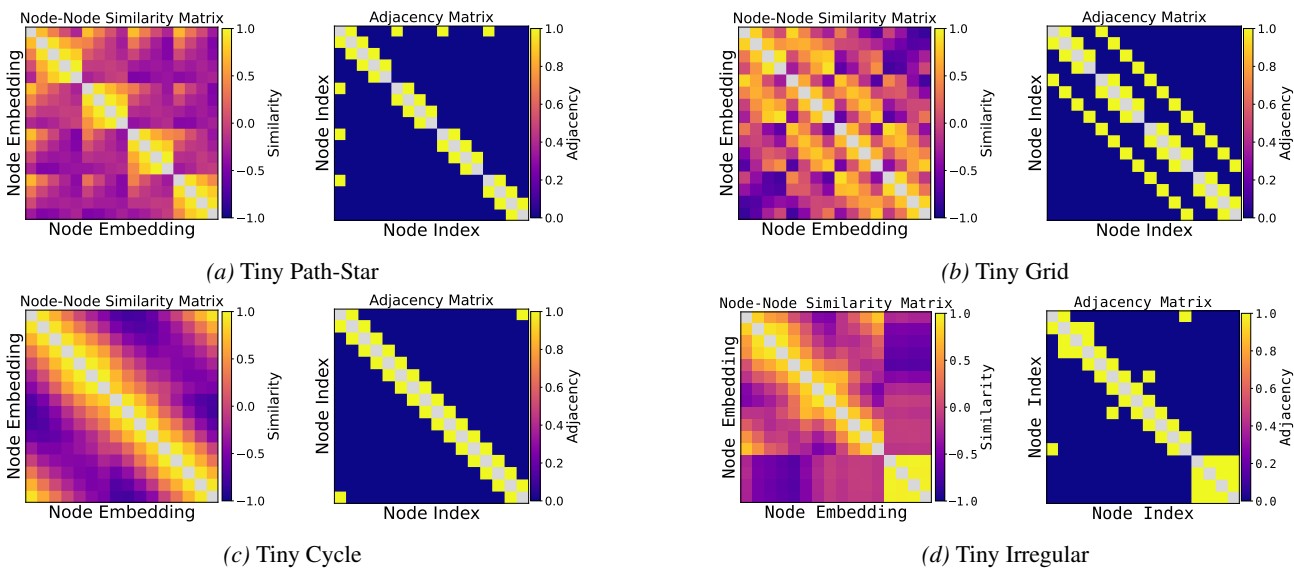

*(a)* Tiny Path-Star

*(b)* Tiny Grid

*(c)* Tiny Cycle

*(d)* Tiny Irregular

*Figure 22.* **Node-node cosine similarity vs. adjacency matrix for the Transformer**: For each graph, we plot the cosine similarity between all node embeddings and compare it against the adjacency matrix. Observe that the cosine similarities exhibit a richer structure than the adjacency matrix, reflecting some notion of multi-hop distance e.g., in the cycle graph, there is a gradual decrease in similarity as we walk towards the diametrically opposite node in any given row.

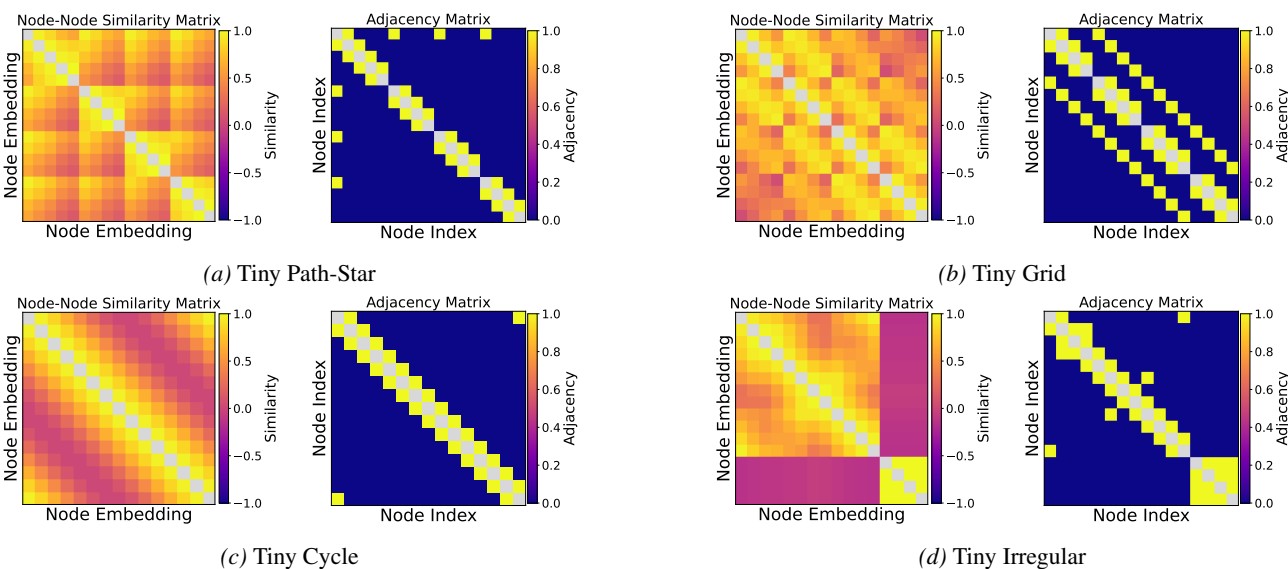

*(a)* Tiny Path-Star

*(b)* Tiny Grid

*(c)* Tiny Cycle

*(d)* Tiny Irregular

*Figure 23.* **Node-node cosine similarity vs. adjacency matrix for the `Node2Vec` model**: Like in the Transformer (Figure 22), we again observe that cosine similarities exhibit a rich, multi-hop structure absent in the adjacency matrix.

E.3.1. EFFECT OF EXPLICIT REGULARIZERS AND SELF-EDGES

We provide an analysis of how the geometry is affected by various aspects of the optimization setting: adding self-edges, varying the initialization, the learning rate, weight decay, and dropout. For all experiments in this section, we train for 10,000 optimization steps to ensure comparability across settings. For this study, we use the `TinyNN` architecture with embedding dimension 512 trained on the easy-to-visualize cycle graph. For each setting, we provide a visualization of (a) the embeddings, (b) the cosine heatmap of the embeddings, and (c) the embeddings' projections onto the graph's eigenvectors, along with the geometric margin (defined in §D.5), which when positive confirms that the storage is indeed geometric.

Our main observations are as follows:

1. **Self-edges:** Adding self-edges to the training graph makes the visualizations significantly cleaner and less zigzaggy (see Figure 24). This aligns with our understanding of the negative eigendirections in §E.4: adding self-edges makes such negative directions less dominant by bringing them closer to zero. Indeed, we see this in the eigenvector projections in Figure 24c.

2. **Initialization:** The scale of initialization has a strong effect on what is learned by the model. Specifically, our default `GPT`-style initialization of embeddings ($\mathcal{N}(0, 0.2^2)$) consistently provided more geometric results, *even without any explicit regularization* (Figure 25). Whereas, a `unit-normal` initialization of the embeddings (Figure 27) resulted in a wider range of behaviors, more sensitive to training conditions e.g., in Figure 27, we show a setting that produces an associative storage.

3. **Learning rate**: We found that higher learning rate tends to improve the geometry. This result is most pronounced for the `unit-normal` initialization of embeddings with $\mathcal{N}(0, 1)$, (compare Figure 27 with the higher learning rate Figure 28), but is also seen for `GPT`-style initialization (compare Figure 26 with the higher learning rate Figure 25).

4. **Explicit regularization**: On the `unit-normal` initialization of embeddings, which we found to be sensitive to optimizer hyperparameters, we find stronger geometries with increasing the weight decay (Figure 29); increasing dropout also helps, though the effect is less pronounced (Figure 30). As noted earlier, on the `GPT`-style initialization, *such regularizers were not necessary* (Figure 25).

5. **Forward vs. reverse edges**: The geometry does not seem contingent on there being both directions of the edges on the training set for our Transformer model (Figure 31)

**Graph memorization as a minimal example of grokking.** These observations point to the fact that geometric memorization of a graph dataset is a minimal instance of grokking, a phenomenon studied for more complex algorithmic datasets (where closed form solutions to the weights are hard to obtain). The helpful effect of weight decay may be related to its effects on grokking (Liu et al., 2023) or rank-minimization (Wang & Jacot, 2024; Galanti et al., 2025; Yunis et al., 2024); similar rank-minimizing effects have been identified for dropout (Cavazza et al., 2018) and learning rate (Andriushchenko et al., 2023). The effect of initialization has also been pointed out in grokking (Liu et al., 2023), and connected to the idea of lazy vs. rich regimes (Kumar et al., 2024) from the perspective of neural tangent kernels, where rich feature-learning is analogous to not geometric representations. However such connections between grokking and the lazy/rich dichotomy is contentious because associative memory, unlike lazy learning, requires the weights to move substantially; similar points have been made in (Zheng et al., 2025; Chou et al., 2025).

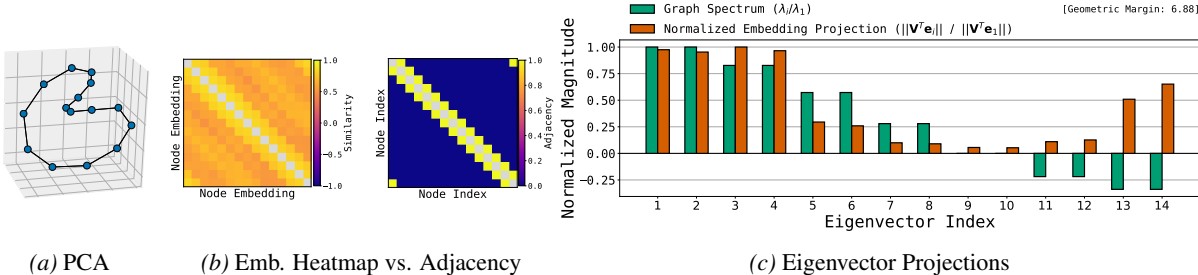

*(a)* PCA          *(b)* Emb. Heatmap vs. Adjacency          *(c)* Eigenvector Projections

*Figure 24.* **Self-edge reduces zig-zagging:** When self-edges are added to the training set, the model relies less on the bottommost eigendirections that are zig-zagging. This is evidenced by the smooth heatmap and the top-heavy barplot; compare this with zigzagging heatmaps and more bottom-heavy barplot in the no-self-edge setting of Figure 25. For this setup, we use a GPT-style initialized network with LR: 0.01; Embedding Init: GPT-Default ($\mathcal{N}(0, 0.02^2)$), Weight Decay: Off, Dropout: Off

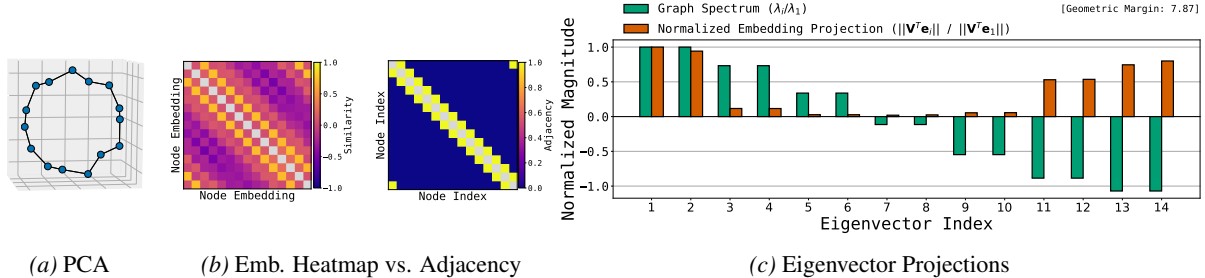

*(a)* PCA          *(b)* Emb. Heatmap vs. Adjacency          *(c)* Eigenvector Projections

*Figure 25.* **Geometry arises without regularization:** Without self-edges or weight decay or dropout, we still see a global geometry in the heatmap, and a skew (towards the highest magnitude) eigendirections. Quantitatively, the geometric margin of this model is indeed positive (7.87), affirming that the graph is indeed stored in the embedding matrices. For this setup, we use a LR: 0.01; Embedding Init: GPT-Default ($\mathcal{N}(0, 0.02^2)$), Weight Decay: Off, Dropout: Off

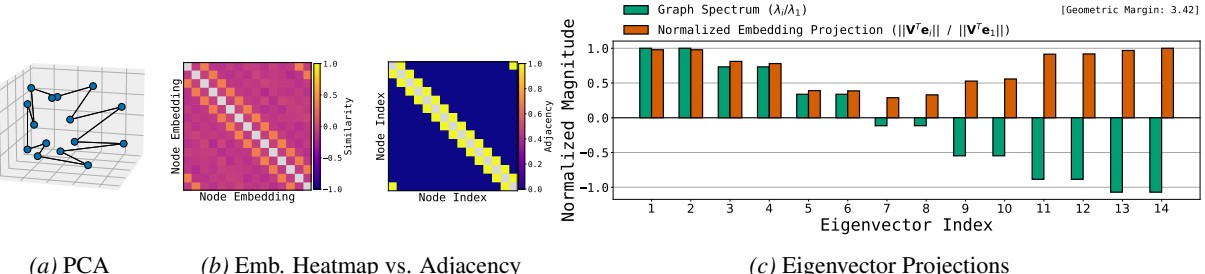

*(a)* PCA          *(b)* Emb. Heatmap vs. Adjacency          *(c)* Eigenvector Projections

*Figure 26.* **Lower learning rate weakens the global geometry:** Reducing the learning rate from $10^{-2}$ in Figure 25 to $10^{-4}$ still results in a geometry, but the eigendirections are less skewed towards the global directions, and the heatmap weaker. For this setup, we use LR: 0.0001; Embedding Init: GPT-Default ($\mathcal{N}(0, 0.02^2)$), Weight Decay: Off, Dropout: Off

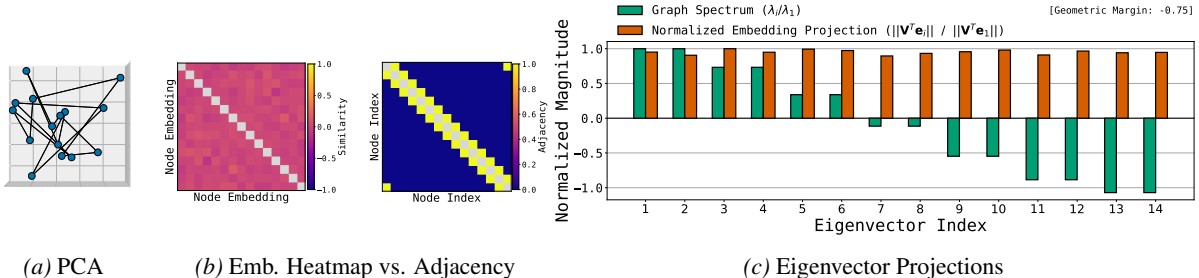

*(a)* PCA          *(b)* Emb. Heatmap vs. Adjacency          *(c)* Eigenvector Projections

*Figure 27.* Unit-Normal-**init learns less geometric solution:** Changing the initialization in Figure 26 results in what is now a purely-associative solution: the heatmap indicates orthogonal embeddings and equally distributed along all directions. The geometric margin of this solution turns out to be negative ($-0.75$). For this setup, we use a network with LR: 0.0001; Embedding Init: PyTorch-Default ($\mathcal{N}(0, 1)$), Weight Decay: Off, Dropout: Off

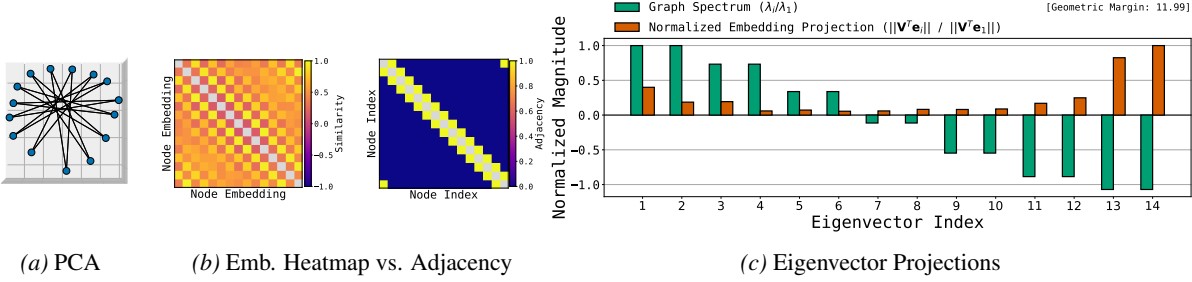

*(a)* PCA       *(b)* Emb. Heatmap vs. Adjacency       *(c)* Eigenvector Projections

*Figure 28.* **Higher learning rate results in more geometry:** Increasing the learning rate from $10^{-4}$ in Figure 27 to $10^{-1}$ results in a more geometric solution. For this setup, we use a network with `LR: 0.1; Embedding Init: PyTorch-Default` ($\mathcal{N}(0,1)$)`, Weight Decay: Off, Dropout: Off`

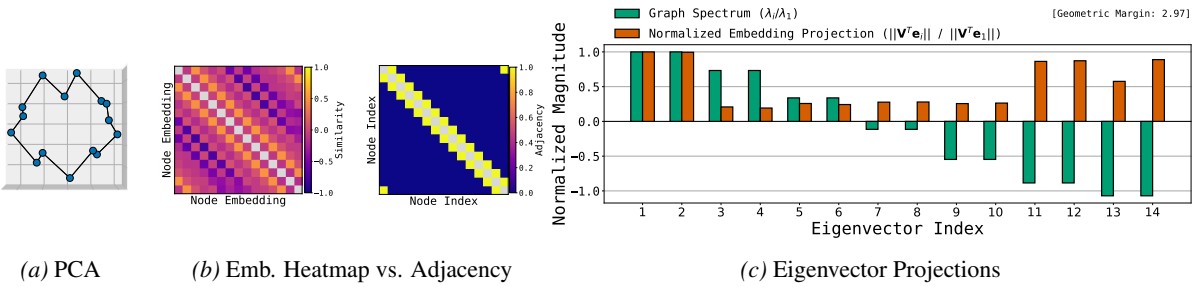

*(a)* PCA       *(b)* Emb. Heatmap vs. Adjacency       *(c)* Eigenvector Projections

*Figure 29.* **Higher weight decay results in more geometry:** Adding weight decay to the setting of Figure 27 results in a more geometric (although zigzagging) solution. For this setup, we use a network with `LR: 0.0001; Embedding Init: PyTorch-Default` ($\mathcal{N}(0,1)$)`, Weight Decay: 10, Dropout: Off`

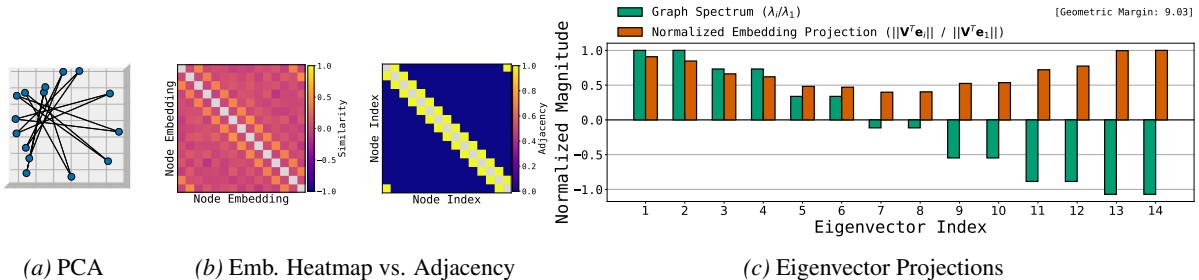

*(a)* PCA       *(b)* Emb. Heatmap vs. Adjacency       *(c)* Eigenvector Projections

*Figure 30.* **Higher dropout results in more geometry:** Adding dropout to the setting of Figure 27 and a slight increase in the learning rate results in a more geometric (although zigzagging) solution. For this setup, we use `LR: 0.001; Embedding Init: PyTorch-Default` ($\mathcal{N}(0,1)$)`, Weight Decay: Off, Dropout: 0.4`

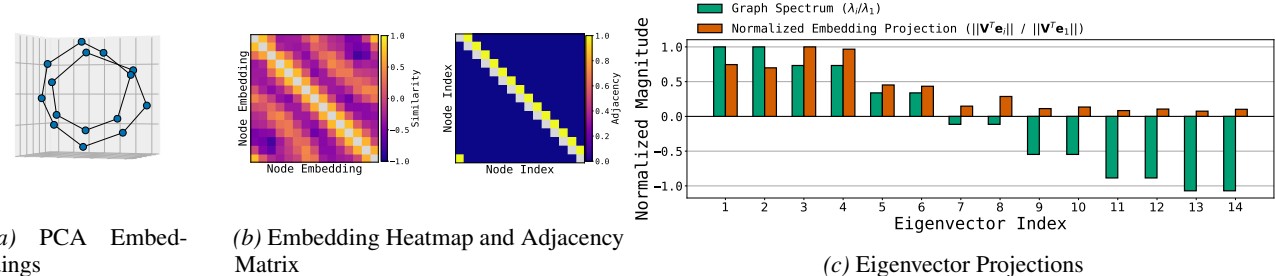

*(a)* PCA Embeddings       *(b)* Embedding Heatmap and Adjacency Matrix       *(c)* Eigenvector Projections

*Figure 31.* **Unidirectional graph can still result in geometry**: At least on small graphs, and for the Transformer (`TinyGPT`), training on only one direction of the edges suffices for the model to store geometrically. Similar results hold for our other graphs.

### E.4. Zigzagging and bipartite memory

Sometimes, we observe in various settings a zigzagging geometry: along certain top directions, adjacent vertices get embedded far away, while vertices that are exactly two-hop away get embedded close. This happens noticeably in two settings: (a) multi-layered models (especially when there is no weight decay) and/or (b) models with untied (un)embedding weights (e.g., Figure 32). We explain why zigzagging occurs, and clarify that this still falls within our extended definition of a geometric memory in §E.5. In short, zigzagging corresponds to negative eigendirections in the adjacency matrix, and these directions are picked up only when the architecture has the expressive power for it.

**Zigzagging directions and negative eigenvalues.** A key insight is that zigzagging directions correspond to *negative* eigendirections in the adjacency matrix. An adjacency matrix (without self-edges) has both positive and negative eigenvalues. Typically, the most negative/bottommost eigendirections assign, alternatingly, $+1$ and $-1$ values as we walk along a path. Likewise, if we were to consider a singular value decomposition (SVD) of the adjacency matrix—which becomes relevant for weight-untied models—such negative eigendirections are "folded up": all strong eigendirections, with the most positive and most negative eigenvalues $\lambda$, become top singular directions with singular value $|\lambda|$ (see (Zhang, 2018) for relevant facts on spectral graph theory). Thus, in all these cases, even though the geometry looks unruly, and even though the cosine heatmaps don't reveal a multi-hop structure, the graph data has been stored by the model in a factorized form.

As discussed in §E.5, this storage is still geometric. Furthermore, the well-behaved "global" directions may still exist, and extractable via an appropriate probe on the embeddings. One way to test this is to add self-edges to the graph and see if the zigzags disappear from the top directions (as this would reduce the singular value of the zigzagging direction). We find this to be the case in §E.3.1 Figure 24. This also offers an explanation for recent observations that adding identity statements improves two-hop reasoning (Lin et al., 2025).

**Why are zigzagging directions learned?** We claim that such directions are picked up by the model (a) because they help drive the loss down and (b) because they can be expressed only by certain models. A model relying only on the top positive eigendirections would recover a positive definite approximation of the adjacency matrix, whose diagonals are highly positive i.e., the logit $f(u)[u]$ would be high since $\mathbf{\Phi}_{\text{geom}}(u) \cdot \mathbf{\Phi}_{\text{geom}}(u)$ would be high. Such a model would suffer a high softmax loss if there are no self-edges in the dataset. The negative eigendirections $\mathbf{e}_{\text{neg}}$ help correct for these highly positive diagonal logits by introducing a negative definite component of the form $\mathbf{e}_{\text{neg}}\lambda_{\text{neg}}\mathbf{e}_{\text{neg}}^T$ where $\lambda_{\text{neg}} < 0$. Indeed, when we add self-edges to the dataset, the zigzagging usually disappears!

Such zigzagging however can only be expressed by some architectures. A weight-tied `Node2Vec` model cannot, for such a model can only express positive definite matrices. A weight-untied model can express such directions as $(\mathbf{e}_{neg})(-\mathbf{e}_{neg})^T$; indeed, weight-untied two-layer models are known to learn an SVD decomposition (Zhao et al., 2025; Karkada et al., 2025; Saxe et al., 2014) which must include such directions as top singular directions. A multi-layered weight-tied model can also express these directions: it can use the intermediate layer to express the negative $\lambda_{\text{neg}}$. Thus, we see zigzagging only where it is helpful (when no self-edges exist) and only when the model can express it (weight-untied or multi-layered).

**Open question.** Although visually unruly, these zigzagging eigendirections are highly-structured: in spectral graph theory, they are understood to be an approximate bipartite cut of the graph. Thus, they correspond to a form of geometric memory distinct from the "global" one, best termed as a *bipartite* geometric memory. Next, while the "global" geometry endows the model with navigational powers (as witnessed in §2), what benefits does the bipartite memory bring to a model's reasoning abilities?

### E.5. Extended definitions of geometric memory

Below, we provide a more complete and general definition of geometric memory.

**Alternative factorizations.** The term "geometric memory" can be applied to broader forms of factorized storage where multi-hop information is not present in the logits but can be accessed via a probe. The definition in Def. 2b, for convenience, implicitly considers a *low-rank, positive definite* factorization of the adjacency matrix. Two variations in this factorization are possible. One is a *full-rank* (or non-low-rank) factorization; another factorization is one that is not positive definite but takes the form $\mathbf{\Phi}_{\text{geom}}(u)^T\mathbf{\Lambda}\mathbf{\Phi}_{\text{geom}}(v)$ where $\mathbf{\Lambda}$ is a diagonal matrix of both positive and negative values; such factorizations can arise in multi-layered networks under various settings. However, these factorizations may not yield nice "multi-hop" logit values (as we will see in §E.4). Yet, both these factorizations are geometric in that the nodes are embedded in a highly non-trivial way; one can still extract multi-hop information via a probe trained to detect the global directions of $\mathbf{\Phi}_{\text{geom}}$. (The reason why the dominant directions correspond to multi-hop information is explained in §4).

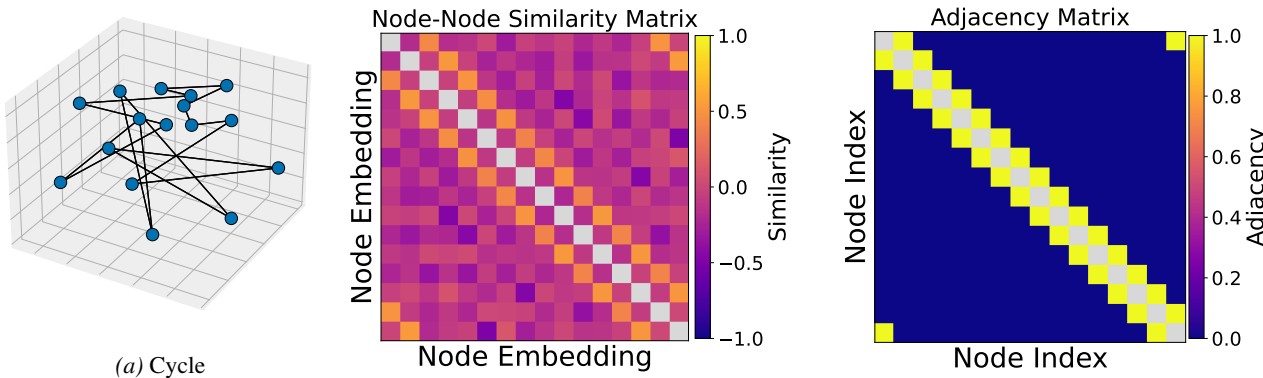

*Figure 32.* **Embeddings of a weight-*untied* Transformer shows a zigzagging geometry.** Both the 3D visualizations of the embeddings and their cosine similarities show a zigzagging pattern (adjacent vertices have very low, even negative, similarity).

**Location of geometry.** In the simple settings of this paper, the geometric embeddings $\mathbf{\Phi}_{\texttt{geom}}$ in practice are found in the token embeddings themselves. But more generally, such an embedding may reside in any other layer. We consider even that to be geometric memorization.

**Self-edges.** The simplistic abstraction in Def. 2b may assign the highest output logit to the input vertex itself (i.e., $\arg\max_v \mathbf{\Phi}_{\texttt{geom}}(u) \cdot \mathbf{\Phi}_{\texttt{geom}}(v) = u$) regardless of whether a self-loop $(u, u)$ exists in the graph. Neural networks typically seem to handle this well through an auxiliary circuit that suppresses such loops. While we discuss one such mechanism in §E.4, for a tidy discussion, we will ignore it as a pedantic detail.

# F. Edge supervision and training dynamics

There are tangential aspects of our large path-star graph training that are worth elaborating on: the roles of reverse edges (§F.1), of pause tokens (§F.2), of interleaving edge-memorization (§F.3).

## F.1. The role of reverse edges

Reverse edges seem to play a nuanced role in our observations. On the large path-star task in §2, we find it necessary to augment training on the reverse edges; on the other hand, for the tiny graphs, a geometry arises even without these reverse edges. Perhaps, reverse edges are needed for larger tasks; or perhaps, they are necessary to perform implicit reasoning and retrieval. We leave it for future work to gain greater clarity on this effect, which is tied to the reversal curse (Berglund et al., 2024; Allen-Zhu & Li, 2023).

### F.1.1. THE CRITICAL ROLE OF REVERSE EDGES IN THE LARGE PATH-STAR TASK

**Edge supervision regimes.** We evaluate three edge supervision regimes for the fixed in-weights graph: (i) *forward-only* edges $\mathcal{D}_{\text{edge}}^{\rightarrow}$, (ii) *backward-only* edges $\mathcal{D}_{\text{edge}}^{\leftarrow}$, and (iii) their mixture $\mathcal{D}_{\text{edge}} = \mathcal{D}_{\text{edge}}^{\rightarrow} \cup \mathcal{D}_{\text{edge}}^{\leftarrow}$, each combined with path supervision.

We consider two types of path-finding tasks. The first, as discussed in Section 2.1, is a *forward generation* ($v_{\text{root}} \rightarrow v_{\text{leaf}}$) task defined by $\mathcal{D}_{\text{path}}^{\rightarrow}$. Another task is *reverse generation* ($v_{\text{leaf}} \rightarrow v_{\text{root}}$), denoted by $\mathcal{D}_{\text{path}}^{\leftarrow}$. Forward path generation is non-trivial to learn as it involves planning or look-ahead, and is adversarial towards next-token learning; the reverse path however is trivial to learn on path-star graphs because each node has a unique predecessor along the target path. We must also clarify that the presence of reverse edges in itself *does not trivialize* the forward path-finding task—these edges provide only local information; thus, the success of the global path-finding task is still *non-trivial*.

We enumerate our observations from these various edge-supervision regimes below:

**Observation 6.** *(**Role of reverse edges**) We find in Figure 33 that:*

1. *A Transformer trained on only the forward edges, struggles on both forward and reverse path-finding tasks (see the middle color in Figure 33).*

2. *A Transformer trained on only the reverse edges, achieves non-trivial accuracy on the reverse path-finding task; however, it fails on the forward path-finding task (see the third color in Figure 33).*

We suspect that the lack of reverse edges either hurts the geometry *or* hurts the retrieval ability of the model. On the other hand, the success of the reverse path-finding task with reverse-only edge memorization could be explained by the fact that the task requires no planning, as discussed in the remark below.

**Remark 6.** *We note that the asymmetry between forward and reversed path-generation tasks stems from their algorithmic complexity. The reversed path generation is algorithmically trivial on path-star graphs because each node has a unique predecessor along any target path—the model simply needs to follow the unique backward edges. Forward generation, however, requires planning (examine each outgoing path) or lookahead (track the reverse path without explicit chain-of-thought and reverse it). Indeed, for the in-context task of B&N'24, the model fails on the forward task, but strikingly succeeds on the reverse task, as corroborated in Figure 34 (right). Even in the in-weights setting Figure 34 (left), the reverse path is generally quicker to learn and yields higher accuracy.*

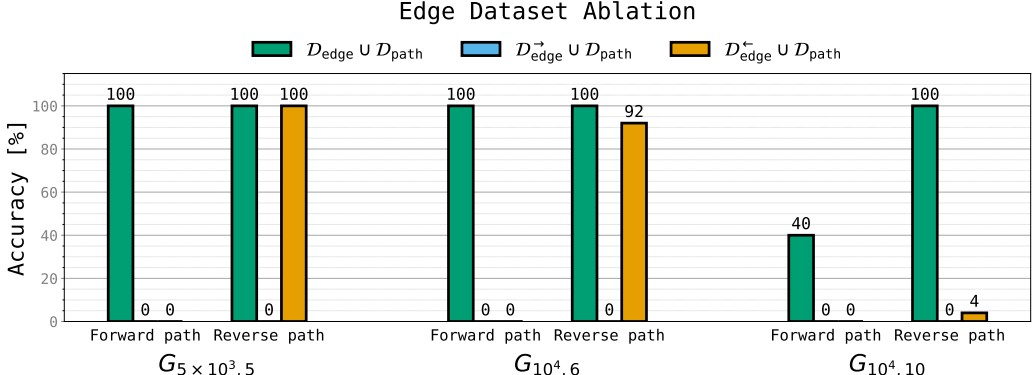

*Figure 33.* **Mixed edge supervision enables forward path generation while forward-only fails due to reversal curse.** Exact-match accuracy on held-out leaves for multiple path-star graphs (varying degree $d$ and path length $\ell$). As established, training on *mixed* edges $\mathcal{D}_{\text{edge}}$ yields high non-trivial *forward* accuracy across graphs. But training on *forward-only* $\mathcal{D}_{\text{edge}}^{\rightarrow}$ fails on both the forward and reverse tasks. This is indicative of the reversal curse. With *backward-only* edges ($\mathcal{D}_{\text{edge}}^{\leftarrow}$) the model attains high accuracy primarily on *reverse* path generation for smaller graphs. This can be reconciled by noting that generating the reverse path is an easier retrieval task. Random forward path accuracy is $1/d$.

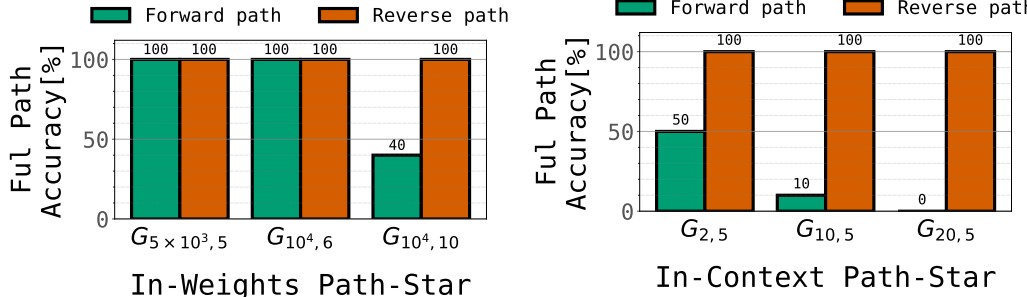

*Figure 34.* **Forward vs. reverse path generation:** The figure contrasts the model's performance on forward (start→leaf) and reverse (leaf→start) path generation tasks for path-star graphs learned either **in-weights** (left) or **in-context** (right). While both methods achieve perfect accuracy on the algorithmically simple reverse path task, their performance on the forward task differs dramatically. **(left)** The in-weights model succeeds at the forward task, which requires planning and look-ahead, demonstrating high accuracy even on large graphs with thousands of nodes. **(right)** In contrast, the in-context model completely fails at forward path generation. This stark difference highlights the superior capability of in-weights learning to internalize and utilize complex graph structures.

## F.2. Pause tokens for computational slack

In the same in-weights path-finding task of §2, we find that it is helpful to insert pause tokens (Burtsev et al., 2020; Goyal et al., 2024) to achieve quicker accuracy gains during training. Pause tokens are added by appending dummy tokens to the prefix of the path-finding task both during training and inference.

Figure 35 shows that adding a short sequence of pause tokens after the prompt reliably boosts exact-match accuracy across graphs, for a given amount of training time. Increasing the number of pause tokens increases speed of convergence.

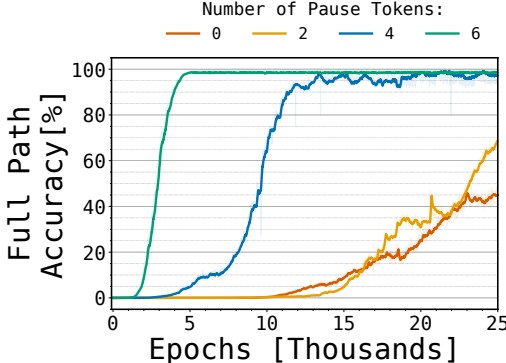

*Figure 35.* **Pause tokens boost convergence speed of in-weights path-star path-finding task of §2.**

## F.3. (Not) Interleaving edge-memorization

In all our experiments, we have interleaved edge-memorization examples with path-finding examples. An alternative training method would be a two-phased approach, where we first enforce edge-memorization, and then follow up by finetuning on the path-finding task. We found this to be less stable, e.g., the model achieves a peak accuracy momentarily, only to deteriorate dramatically right after. This is a manifestation of the well-known effect that finetuning has on parametric memory (Li et al., 2024; Luo et al., 2025). Since this is a confounding effect, we do not choose this regime for our experiments.

However, we confirm that even in this regime, our models do achieve a high peak accuracy (see Figure 36). The fact that the composition task is learnable in this regime implies that the edge-pretrained model must have come with an adequate global geometry despite being trained only on local supervision (Refutation 3a).

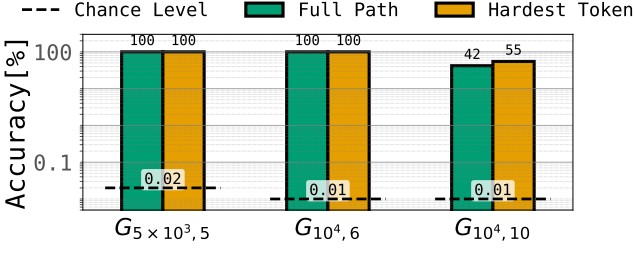

*Figure 36.* **Locally supervised model succeeds at path-finding task.** We report the *peak* accuracy under finetuning an edge-memorizing model on our path-finding task of §2. We emphasize that this accuracy value is only reached momentarily, and typically deteriorates quickly during further finetuning. Nevertheless, this suggests that local supervision alone was adequate to synthesize a global geometry (Refutation 3a). Learning rates of the two phases are given in §D.3.

# G. Proofs about representational complexity

## G.1. Empirical failure of composition learning under associative memory

As discussed in §2.3, it is well-known that certain compositional tasks are hard to learn empirically; theoretically, this has been proven in a certain sense. However, these prior discussions involve composing information available in the context, rather than in the weights. To test this intuition in our in-weights composition task, we design an experiment where we freeze the embeddings of a Transformer and train it on our path-star task. If the model succeeded in this task, it may mean one of two things: either the model develops a geometric memory in a subsequent layer, or the model does in fact efficiently learn how to compose associative matrix operations—going against our intuition derived from prior limits on composition learning. However, we find that even after 50,000 training epochs (using the same `GPT-mid` architecture described in Table 2 and §D.2.1, except with frozen token embeddings; and the optimization hyperparameter grid search reported in §D.3), the model fails to learn the in-weights path-star task. We believe, this is preliminary evidence that associative memory indeed struggles to learn compositional in-weights tasks. A fleshed-out proof of this negative result is left for future work.

## G.2. Succinctness does not break the tie: Proof of Proposition 3d

In datasets where redundancies exist, the complexity of a lookup table scales quickly with the training set size (say $n$), whereas the more succinct solution does not (or at worst, grows polynomially slower). For example, if the data is linearly separable in some constant dimensionality, the linear classifier can be described in $\Theta(1)$ bits, whereas a lookup table (such as a nearest neighbor model) would require $\Theta(n)$ bits. However, this wide disparity in complexity does not necessarily surface in our setting, which is a *memorization* task without redundancies. Concretely, at least in terms of bits and $\ell_2$ norms, there are simple graphs where both the geometric and associative views are equally complex— and at worst, differ by a constant factor of at most 2 or so depending on whether edge/weight symmetries are constrained during training on not. Thus, the gap does not scale as $\Theta(n)$, which in our case would be the edge count or the vertex count of our graph.

We make two notes about the constant multiplicative factor of 2 in the gap between the two complexities. First, the factor does not appear if only direction of the edges are stored, or if the embedding weights are untied. Second, it may seem that this constant factor may partly explain the geometric bias; however, a geometry appears even when memorizing only one direction of the edges in the smaller graphs (see Figure 31), where both forms of storage must be equally succinct.

**Proof intuition.** We first roughly derive the bit and norm complexity for a general graph. The key idea is that associative memory scales with the edge count, whereas a geometric memory scales with the vertex count. Then, we argue how this resolves to similar values for graphs like the path-star or a cycle, where the edge count and the vertex count are the same (plus or minus 1).

**Notation.** We let $|V|$ be the number of entities and $|E|$ the number of associations.

**Proposition 1.** *(**Bit complexity**) Storing a graph $\mathcal{G} = (V, E)$*

- *with associative memory requires $|E| \log |V|$ many bits (with a multiplicative factor of 2 if both direction of the edges must be stored).*

- *with geometric memory requires $|V| m \log \Delta$ many bits where $m$ is the embedding dimensionality, and $\Delta$ is the number of cells along each dimension required to avoid collision. This is doubled if (un)embedding weights are not tied.*

*Proof.* In the local associative view, given as input any vertex $u$, we must be able to lookup the vertex IDs of its neighbors. Thus, at the position of this vertex, we need a total of $d(u) \log |V|$ many bits (where $d(u)$ is the degree, and $\log |V|$ is the bit length of each ID). Summing this over all vertices gives us $|E| \log |V|$ many bits (since the sum of all degrees must equal the edge count). Note that an extra factor of 2 appears if both the direction of the edges must be stored.

In the geometric embedding, each vertex is stored as a vector in $m$ dimensions. Each dimension must store one of $\Delta$ values, which requires $\log \Delta$ bits. Summing this up across all dimensions and vertices gives us the result. This is doubled if the unembedding matrix is not weight-tied.

$\square$

**Proposition 2.** *($\ell_2$ **norm complexity**) Given a graph $\mathcal{G} = (V, E)$, and without loss of generality, given the margin constraint that if $u$ is a neighbor of $v$, then $f(u)[v] - \arg\max_{w \notin nbr(u)} f(u)[w] \geq 1$ ($f(u)[v]$ denotes the logit of predicting $v$ given $u$;*

*and $w$ is a non-neighbor), then*

1. *associative memory requires $\ell_2$ norm of at most $\sqrt{|E|}$ with an extra factor of $\sqrt{2}$ if both directions must be stored.*

2. *geometric memory requires an $\ell_2$ norm of at least $\sqrt{|V|}$ with an extra factor of $\sqrt{2}$ if (un)embedding weights are not tied.*

*Proof.* Recall that associative memory takes the form $f(u)[v] = \mathbf{\Phi}(v)^T \mathbf{W}_{\text{assoc}} \mathbf{\Phi}(u)$. Without loss of generality, if we assume that the embeddings are one-hot vectors in $\mathbb{R}^{|V|}$, we can set $\mathbf{W}_{\text{assoc}}$ to be the adjacency matrix to satisfy our margin constraint. The $\ell_2$ norm (of the free parameters in $\mathbf{W}_{\text{assoc}}$) is $\sqrt{|E|}$.

In the geometric view, recall that $f(u)[v] = \mathbf{\Phi}_{\text{geom}}(v)^T \mathbf{\Phi}_{\text{geom}}(u)$. To have $\mathbf{\Phi}_{\text{geom}}(v)^T \mathbf{\Phi}_{\text{geom}}(u) > 1$, we need $\|\mathbf{\Phi}_{\text{geom}}(v)\|^2 + \|\mathbf{\Phi}_{\text{geom}}(u)\|^2 > 2$. Thus, roughly, all embedding norms must be at least 1, implying that $\sum_u \|\mathbf{\Phi}_{\text{geom}}(u)\|^2 \geq |V|$. $\square$

## Proof of Main Refutation 3d

*Proof.* Our proof follows from the fact that for graphs like the path-star graph and the cycle graph, the edge count and vertex count are nearly equal. Thus, both notions of complexity—the associative scaling with the edge count and the geometric with vertex count—can be shown to reduce to similar values here from the above propositions.

This is straightforward to see for $\ell_2$ norm based complexity based on Proposition 2. For the bit complexity estimates, from Proposition 1, we know that associative memory costs $|V| \log |E|$ many bits; for the geometry, we need to pin down the values of the embedding dimension $m$ and the cell count $\Delta$.

The path-star graph can be embedded such that each path is stored along a unique dimension (thus totally $m = d$ dimensions), and each dimension can be gridded into $\ell$ many cells. This requires a bit complexity of $|V| d \log \ell$. For a more ambitious geometry, we could further squeeze this into $\log d$ dimensions while still keeping the paths well-separated (by the Johnson–Lindenstrauss lemma), resulting in $|V| \log d \log \ell$ bits. This is still greater than the cost of associative memory which is approximately $|V|(\log d\ell) = |V|(\log d + \log \ell)$.

A similar argument works for a cycle graph. Here, we can embed in $m = 2$ dimensions, with a cell count of $|V|/2$, thus totaling $2|V| \log(|V|/2)$ bits for geometric memory, again greater than $|V| \log |E|$ when $|V| = |E|$. $\square$

## G.3. Weight-tied dual-encoder models can represent certain special or contrived forms of associative memory

In the standard associative memory view that we have discussed, associations are represented through the function $\mathbf{\Phi}(v)^T \mathbf{W}_{\text{assoc}} \mathbf{\Phi}(u)$. However, weight-tied dual encoder models like `Node2Vec` can only represent functions of the form $\mathbf{\Phi}(v)^T \mathbf{\Phi}(u)$. When weight-tied, this precludes the form of associative memory we care about; but it still allows two forms of associative memories under special conditions: if the graph is bipartite (Proposition 3) or if the embedding dimensionality is sufficiently large (in which case, the construction is contrived; see Proposition 5). We can also show that non-weight-tied models can represent any graph associatively (see Proposition 4). In all these results below, we will show that our constructions are associative in that they achieve a logit of 1 on adjacent vertices, but a logit of 0 on any non-adjacent vertices. In this sense, only local associations are captured, whereas no global geometry is.

**Proposition 3.** *Weight-tied dual encoder models of the form $\mathbf{\Phi}(v) \cdot \mathbf{\Phi}(u)$ can store a graph $\mathcal{G}$ associatively when the graph is bipartite.*

*Proof.* Assume that the graph partition consists of vertices $V_1$ and $V_2$. The construction is to represent one set of vertices with a one-hot representation, and the other by a one-hot adjacency vector. That is, consider an embedding $\mathbf{\Phi} : V \to \mathbb{R}^{|V_1|}$ that has dimensionality $|V_1|$. For any $v_i \in V_1$, assume $\mathbf{\Phi}(v)$ is such that it is 1 only at index $i$. For $w \in V_2$, $\mathbf{\Phi}(v)$ at index $i$ is 1 if and only if $v$ is adjacent to $v_i$ i.e., $\mathbf{\Phi}(v)_i = \mathbf{1}[(w, v_i) \in E]$.

Thus, if $w$ and $v$ belong in the same partition $\mathbf{\Phi}(v) \cdot \mathbf{\Phi}(w) = 0$; if in different partitions, $\mathbf{\Phi}(v) \cdot \mathbf{\Phi}(w) = \mathbf{1}[(w, v) \in E]$. $\square$

**Proposition 4.** *Weight-untied dual encoder models of the form $\mathbf{\Phi}_1(v) \cdot \mathbf{\Phi}_2(u)$ can store any graph associatively when the embedding dimension is at least as large as the vertex count. Furthermore, this can be approximately learned in one-step of gradient descent under the correlation loss $\sum_{(u,v) \in E}(\mathbf{\Phi}_1(u)) \cdot \mathbf{\Phi}_2(v)$ with one tower frozen.*

*Proof.* The proof parallels that of Proposition 3. We set one of the embeddings to a one-hot representation of the vectors; the other is set to embed the adjacency row. In matrix notation, the resulting model (which outputs a vector of logits for a given input $\mathbf{u}$) looks like $f(\mathbf{u}) = \mathbf{W}_2 \mathbf{W}_1 \mathbf{u} = \mathbf{A}\mathbf{I}\mathbf{u} = \mathbf{A}\mathbf{u}$.

To show that this can be learned in one-step of gradient descent, consider $\mathbf{\Phi}_1$ initialized randomly. Given sufficiently high embedding size, these embeddings are orthogonal. Under the correlation loss, if one were to freeze $\mathbf{\Phi}_1$, $\mathbf{\Phi}_2(v)$ will be updated in the direction of $\sum_{u \in \mathtt{nbr}(v)} \mathbf{\Phi}_2(u)$, which is equivalent to the adjacency row up to a rotation. For a sufficiently large learning rate, the resulting model would effectively store $\mathbf{A}$ up to a rotation defined by $\mathbf{\Phi}_1$. $\qquad\square$

Now, for a generic graph, and for a weight-tied dual encoder model, we show that `Node2Vec` can represent a contrived form of associative memory when there is a much larger embedding dimensionality scaling with edge count. [8]

**Proposition 5.** *Weight-tied dual encoder models of the form $\mathbf{\Phi}(v) \cdot \mathbf{\Phi}(u)$ can store a graph associatively given an embedding dimensionality of $|E|$ where $E$ is the set of pairwise associations.*

*Proof.* For each node, we take its corresponding row from the $|V| \times |E|$ *incidence* matrix. In other words, the embedding is $|E|$-dimensional, where the $i$th dimension corresponds to the $i$th edge; for edge $(u, v)$, the embedding of $u$ and $v$ both are set to 1 along the dimension corresponding to $(u, v)$. Notationally, if the edges are indexed as $1, 2, \ldots$, then $\mathbf{\Phi}(u)_i = \mathbf{1}[u \in e_i]$, where $\mathbf{1}$ is the indicator function. Then, we have that $\mathbf{\Phi}(u) \cdot \mathbf{\Phi}(v) = \mathbf{1}[(u, v) \in E]$. Thus, the dot products capture only local information. $\qquad\square$

---

[8]We suspect this should also be a lower bound i.e., such a large dimensionality must be needed to represent associatively in `Node2Vec`.

## H. Detailed analysis of spectral bias in `Node2Vec`

Let $G$ be a graph of $n$ nodes $\{1, 2, \ldots, n\}$. Let $\mathbf{A} \in \mathbb{R}^{n \times n}$ be the adjacency matrix, $\mathbf{D} \in \mathbb{R}^{n \times n}$ the diagonal degree matrix, and let the embedding of the nodes be denoted by $\mathbf{V} \in \mathbb{R}^{n \times m}$, where $m$ is the embedding dimensionality. Let $\mathbf{L} = (\mathbf{I} - \mathbf{D}^{-1}\mathbf{A}) + (\mathbf{I} - \mathbf{D}^{-1}\mathbf{A})^T$ denote the *asymmetrically normalized random walk graph Laplacian*. The second topmost eigenvectors of $-\mathbf{L}$ are called the Fiedler vectors; we refer to them and the next few eigenvectors as Fiedler-like eigenvectors. The topmost eigenvector of $-\mathbf{L}$ is a degenerate eigenvector of (approximately) all 1s.

`Node2Vec` **setup.** We consider the simplest `Node2Vec` model, where the embeddings are directly parameterized by $\mathbf{V}$. We consider a 1-hop objective (where the neighborhood is defined by the immediate neighbors rather than by more distant ones discovered by a random walk). Note that our objective uses the full softmax loss:

$$\mathcal{J}_{\texttt{Node2Vec}}(\mathbf{V}) = \max_{\mathbf{V}} \sum_i \frac{1}{|\texttt{nbr}(\cdot)|} \sum_{j \in \texttt{nbr}(i)} \log \underbrace{\frac{\exp(\mathbf{v}_i^T \mathbf{v}_j)}{\sum_k \exp(\mathbf{v}_i^T \mathbf{v}_k)}}_{p(i,j)}, \tag{1}$$

where $\texttt{nbr}(\cdot)$ denotes the neighboring vertices in graph $\mathcal{G}$. The above (degree-normalized) objective resembles optimizing over a sequence dataset where we sample the first vertex uniformly, and the second vertex uniformly from its neighborhood.

Let $\mathbf{P} \in \mathbb{R}^{n \times n}$ be the matrix of probabilities $p(i, j)$, where:

$$\mathbf{P} = \texttt{row\_softmax}(\mathbf{V}\mathbf{V}^{\mathsf{T}}). \tag{2}$$

The dynamics of the `Node2Vec` algorithm can be expressed as below:

**Lemma 1.** *The update on the representations under gradient maximization of the* `Node2Vec` *objective in Eq 1 can be written as:*

$$\Delta\mathbf{V}(t) = \eta\mathbf{C}(t)\mathbf{V}(t) \text{ where, } \mathbf{C}(t) = \underbrace{(\mathbf{D}^{-1}\mathbf{A} - \mathbf{P}(t)) + (\mathbf{D}^{-1}\mathbf{A} - \mathbf{P}(t))^T}_{\textit{co-efficient matrix}} \tag{3}$$

We prove this in §H.4.

### H.1. Challenges of analyzing the dynamics

Unlike previously-studied dynamics which simplify nicely, this system may behave in one of many ways. For one, it may simply diverge, but if we are a bit lucky, it may at least converge in direction (like in logistic regression (Soudry et al., 2018)); but then, this direction may be degenerate—an all-one representation could potentially be a stable direction—and perhaps nice directions are visible only if we analyze with early-stopping. One way to get a handle on this would have been to show that, in the limit, we have $\mathbf{C}(t) \to 0$; solving this could then spell out the (limit) probability matrix $\mathbf{P}$, if not the inner products $\mathbf{V}\mathbf{V}^T$ themselves. However, we find that $\mathbf{C}$ cannot be zero as that would require the self-probability term $p(i, i) = 0$, which is infeasible. This closes all obvious analytical routes to understanding this system, so we turn to an empirical study.

### H.2. Empirical intuition of the dynamics

Empirically, we find that the model tends towards a gradient-zero state by working its way toward satisfying a two-fold constraint in Observation 7. First, the column space of $\mathbf{V}(t)$ converges to the top eigenvectors of $\mathbf{C}(0)$—which is approximately the negative of the Laplacian $\mathbf{L}$. Concurrently, $\mathbf{C}(t)$ itself converges such that its null space matches these eigenvectors. Together then, the update $\Delta\mathbf{V}(t)$ in Lemma 1 must become zero.

**Observation 7.** *We find that $\Delta\mathbf{V}(t) \to 0$ through the following concurrent behaviors:*

- *The null space of $\mathbf{C}(t)$ spans the top eigenvectors of $-\mathbf{L}$.*

- *The column space of $\mathbf{V}(t)$ converges to the top eigenvectors of $-\mathbf{L}$.*

Crucially, we find that this can happen (a) even without a constraint on the dimensionality $m$ and (b) this requires *no* early-stopping (see Remark 7 for a more nuanced discussion of this).

We lay out our empirical intuition below, deferring a more mathematical description of the same to the following section. First, we postulate a key invariant during training: the eigenvectors of the co-efficient matrix $\mathbf{C}(t)$, the probability matrix $\mathbf{P}(t)$ and the embeddings $\mathbf{V}(t)$ all remain (inexplicably) stable during training. In particular, since the system begins with $\mathbf{P}(0) \approx \mathbf{I}$, and so $\mathbf{C}(0) \approx -\mathbf{L}$ all these eigenvectors are then fixed as the eigenvectors of the normalized random walk graph Laplacian.

Next, we find that the eigenvalues of the co-efficient matrix $\mathbf{C}(t)$ begin *negative*, gradually approaching zero. The top eigenvectors reach zero first, achieving the second condition in Observation 7. That the values begin negative follows from the fact that the co-efficient matrix begins as the negative graph Laplacian. That these values approach zero follows from the fact that embedding vectors become less orthogonal over time; this in turn reduces the eigenvalues of $\mathbf{P}(t)$, which increases the eigenvalues of $\mathbf{C}(t)$.

Next, due to the negative eigenvalues of $\mathbf{C}(t)$, the embeddings $\mathbf{V}$ along the lowermost eigendirections quickly diminish, achieving our first condition. Note that this means we do not want early-stopping; unlike in the quadratic loss formulation of Karkada et al. (2025), it is longer training that filters out the lower eigenvectors. (Although, the existence of a degenerate eigenvector complicates this; see Remark 7). This achieves the first condition in Observation 7.

Observe that this argument does not require any upper bound on the size of the embedding space. It is unclear if a more succinct, margin-maximizing or norm-minimizing view of these dynamics is expressible.

**Remark 7.** *(**The degenerate vector and early-stopping**) When the graph Laplacian is symmetrically normalized (e.g., $\mathbf{D}^{-1/2}\mathbf{A}\mathbf{D}^{-1/2} - \mathbf{I}$), the top-most eigenvector of the graph Laplacian is a degenerate vector that assigns a constant value to all nodes, and provably corresponds to a zero eigenvalue. However, in our setting, this eigenvalue is slightly above zero, likely due to the asymmetric nature of our Laplacian. Therefore, as we train for longer, the model would become degenerate thus requiring early-stopping. However, this is a conceptually different reason to early-stop than the one in Karkada et al. (2025). Here we may need to early-stop to prevent collapse to the top eigenvector, whereas in Karkada et al. (2025), it is to prevent expansion to bottom eigenvectors.*

### H.3. Mathematical description

Below, we provide a more mathematical description of the above summary by dividing it up into various propositions. Our proofs for these propositions are highly informal. However, our propositions hold in practice without our simplifying assumptions (at least in the graphs we study). We leave it for future work to deliver a rigorous proof and a more clearly characterized theorem statement.

First, we note that the co-efficient matrix approximately begins as the negative graph Laplacian for an appropriately large initialization. (Without this assumption, we may still make a connection to a graph Laplacian-*like* object).

**Assumption 1.** We assume a sufficiently large magnitude or embedding dimensionality of random initialization such that the initial embeddings are nearly orthogonal as $\mathbf{V}(0)\mathbf{V}(0)^T \approx c\mathbf{I}$.

**Fact 1.** *Under Assumption 1,*
$$\mathbf{C}(0) \approx -\mathbf{L} = (\mathbf{D}^{-1}\mathbf{A} + (\mathbf{D}^{-1}\mathbf{A})^T - 2\mathbf{I}). \tag{4}$$

*Proof.* At time $t = 0$, the embeddings $\mathbf{V}(0)$ are all random, and hence nearly orthogonal to each other i.e., $\mathbf{V}(0)\mathbf{V}(0)^T \approx c\mathbf{I}$, where $c$ is some constant that depends upon the magnitude of the random initialization. Since $\mathbf{P} = \texttt{row\_softmax}(\mathbf{V}\mathbf{V}^\mathtt{T}(\mathtt{t}))$, for a sufficiently large $c$, $\mathbf{P}(0) \approx \mathbf{I}$, proving our claim. $\square$

Next, we make the empirical observation that the eigenvectors of $\mathbf{P} + \mathbf{P}^T$ match the eigenvectors of the embedding inner products $\mathbf{V}\mathbf{V}^T$. Note that $\mathbf{P}$ is related to the inner product via a non-linear row softmax operation, rendering a proof of this observation highly non-trivial. We assume this observation (without even an intuitive proof) for the rest of our discussion.

**Observation 8.** *(**Eigenvectors remain unchanged under a row-softmax transform**) The eigenvectors of $\mathbf{P}(t) + \mathbf{P}(t)^T$ at any time $t$, are also approximately the eigenvectors of the embeddings $\mathbf{V}(t)\mathbf{V}(t)^T$, appearing in the same order.*

From the above observation, we can conclude that the eigenvectors of the system match the Laplacian throughout training. This follows by how the updates reduce to multiplications between matrices sharing the same eigenspaces.

**Proposition 6.** *(Time-invariant eigenvectors match that of the Laplacian)* *With Assumption 1 and by assuming Observation 8 as a given, we have that for all t, the quantities* $\mathbf{C}(t), \mathbf{P}(t) + \mathbf{P}(t)^T, \mathbf{V}(t)\mathbf{V}(t)^T$ *have the same eigenvectors as that of the negative Laplacian* $-\mathbf{L}$.

*Proof.* At any time $t$, we can write the embedding vectors as

$$\mathbf{V}(T) = \prod_{t=0}^{T-1}(1 + \eta\mathbf{C}(t))\mathbf{V}(0), \tag{5}$$

and so the inner product as

$$\mathbf{V}(T)\mathbf{V}(T)^T = \prod_{t=0}^{T-1}(1 + \eta\mathbf{C}(t)) \underbrace{\mathbf{V}(0)\mathbf{V}(0)^T}_{\approx c\mathbf{I} \text{ by Assumption 1}} \prod_{t=0}^{T-1}(1 + \eta\mathbf{C}(t))^T \tag{6}$$

$$\approx c \prod_{t=0}^{T-1}(1 + \eta\mathbf{C}(t))(1 + \eta\mathbf{C}(t))^T. \tag{7}$$

From here, we inductively prove our claim. At $t = 0$, it is indeed the case that $\mathbf{C}(t), \mathbf{P}(t), \mathbf{V}(t)\mathbf{V}(t)^T$ all have the same eigenvectors as $\mathbf{L}$ either by Fact 1 for $\mathbf{C}(0)$, or trivially since $\mathbf{P}(t)$ and $\mathbf{V}(t)$ are orthogonal matrices. We assume this is true for all $t$ until $T - 1$. Then, by the above equation, it is also true that the inner product $\mathbf{V}\mathbf{V}^T$ shares these eigenvectors. By invoking Observation 8, we can say that the same is true of the probability matrix $\mathbf{P} + \mathbf{P}^T$. Subsequently, this is true of $\mathbf{C}(t)$, which equals $\mathbf{D}^{-1}\mathbf{A} + (\mathbf{D}^{-1}\mathbf{A})^T + (\mathbf{P} + \mathbf{P}^T)$. (Note that the first term here has the same eigenvectors as $-\mathbf{L}$ as it is off only by the identity matrix.) This proves our inductive assumption. $\square$

Next, we begin to bound the eigenvalues of the system. For the sake of our informal proofs we make some simplifying assumptions that make our matrices approximately symmetric; however, we do not need these assumptions in practice.

**Assumption 2.** For theoretical convenience, we assume that:

- $\mathbf{P} \approx \mathbf{P}^T$.

- the embeddings (i.e., the rows of $\mathbf{V}$) are of equal $\ell_2$ norms.

- the degrees of all nodes are roughly equal.

Now, we can observe a bound on the eigenvalues of the probability matrix.

**Proposition 7.** *(Eigenvalues of the probability matrix)* *Under Assumption 2, the eigenvalues of* $\mathbf{P}(t) + \mathbf{P}(t)^T$ *are such that:*

1. *their sum is upper bounded by* $2n$ *(where $n$ is the number of nodes).*

2. *they are each approximately bounded in* $[0, 2]$

*Proof.* For the first result, recall the fact the sum of eigenvalues is the trace of the matrix. Since each diagonal term is at most 2 (it is $2p(i, i)$), the trace is at most $2n$. This requires no special assumptions.

For the bounds on each eigenvalue, we can rely on the Gershgorin Circle theorem, which states that the eigenvalues lie in the union of discs centered at the diagonals $p(i, i)$, each with radius equal to the sum of the absolute off-diagonal terms, $\sum_{j \neq i} p(i, j)$. The upper bound is then equal to the sum of the rows. For $\mathbf{P}(t)$, this sum is equal to 1 due to the row-softmax operation. Assuming $\mathbf{P}^T \approx \mathbf{P}$—which is approximately true in practice, especially if the node degrees are uniform (but not always)—, we can conclude that the upper bound is approximately 2.

For the lower bound, if we have that the self-probabilities $p(i, i)$ are the largest in any row, then the lower bound $p(i, i) - \sum_{j \neq i} p(i, j)$ is at least zero. This is indeed the case if the embeddings of all nodes are of approximately equal norms, in which case the inner product $\mathbf{V}\mathbf{V}^T$ is highest along the diagonal. $\square$

**Proposition 8.** *The eigenvalues of* $\mathbf{D}^{-1}\mathbf{A} + (\mathbf{D}^{-1}\mathbf{A})^T$ *approximately lie in* $[-2, 2]$ *assuming that the nodes have approximately uniform degree as in Assumption 2.*

*Proof.* The diagonal of $\mathbf{D}^{-1}\mathbf{A}$ is 0 (assuming no self-loops in the graph), while the off-diagonal values are all positive and sum up to 1 in each row. When the node degrees are approximately uniform, $\mathbf{D}^{-1}\mathbf{A} \approx (\mathbf{D}^{-1}\mathbf{A})^T$. From the Gergshgorin circle theorem, the eigenvalues lie in the union of discs centered at the diagonal (from the above, 0) with radii equal to the sum of the absolute off-diagonal terms (from the above, 2), thus proving our claim. □

Now, we can establish the conditions in Observation 7, namely, the convergence of the null space of the co-efficient matrix from Observation 1, and then the convergence of the embedding vectors.

**Proposition 9.** *Under Assumption 2 and Assumption 1, at any time instant t, the eigenvalues of* $\mathbf{C}(t)$ *are all strictly negative (except for the topmost eigenvalue, which is of a degenerate all-1 eigenvector, and is approximately zero), and this is so until when the top eigenvectors of the Laplacian converge into the null space of* $\mathbf{C}(t)$ *(i.e., their eigenvalues become zero).*

*Intuition.* Recall that $\mathbf{C}(t) = \mathbf{D}^{-1}\mathbf{A} + (\mathbf{D}^{-1}\mathbf{A})^T - (\mathbf{P} + \mathbf{P}^T)$. The eigenvalues of the first term lie approximately in $[-2, 2]$ from Proposition 8, while that of the probability term lie in $[0, 2]$, from Proposition 7. Note that eigenvalues of the probability matrix all begin uniformly at 2 in the beginning (as $\mathbf{P}(0) = \mathbf{I}$, by Fact 1 under Assumption 1), as a result of which the initial eigenvalues of $\mathbf{C}(t)$ start at or below 0.

While the embeddings are initialized orthogonally under Assumption 1, they become less orthogonal during training, leading to a gradual decrease of the diagonal self-probability terms in $\mathbf{P} + \mathbf{P}^T$. Intuitively, this also means that the eigenvalues of $\mathbf{P} + \mathbf{P}^T$ must themselves all decrease from the initial value of 2 (based on Proposition 7). In turn, the eigenvalues of $\mathbf{C}(t)$, which begin negative must gradually inch toward zero. The topmost eigenvalues—which are closest to zero—are the first to reach zero.[9] □

**Proposition 10.** *(**Embeddings converge to top eigenvectors**) Assuming Observation 8 as a given, for a sufficiently small learning rate $\eta$, with increasing timestep t, the column space of* $\mathbf{V}(t)$ *converges to the top eigenvectors of the negative graph Laplacian* $-\mathbf{L}$*, independent of the embedding dimensionality.*

*Proof.* We can examine the dynamics of each embedding dimension separately.[10] For the embedding dimension $j = 1, 2, \ldots, m$, let $\mathbf{r}_j \in \mathbb{R}^n$ denote the $j$th column of the embedding matrix $\mathbf{V}$. The dynamics of this column (we drop the index $j$ for the moment) at any timestep $t$ can be isolated as:

$$\mathbf{r}(t) = \prod_{t=0}^{T}(1 + \eta\mathbf{C}(t))\mathbf{r}(0). \tag{8}$$

Given that $\mathbf{C}(t)$ have time-invariant eigenvectors (by Proposition 6), this can be further simplified as

$$\mathbf{r}(t) = \mathbf{E}\left(\prod_{t=0}^{T}(1 + \eta\mathbf{\Lambda}(t))\right)\mathbf{E}^T\mathbf{r}(0). \tag{9}$$

Given that the eigenvalues in $\mathbf{\Lambda}$ are all less than or equal to zero (by Proposition 9), the term $(1 + \eta\mathbf{\Lambda}(t))$ must consist of a diagonal of values in $[0, 1]$ for an appropriately small learning rate. Furthermore, as $T$ becomes large, the values of the top eigenvectors (which have the least eigenvalues, and therefore, the largest value of $1 + \eta\lambda_i(t)$) must come to dominate. Then, as $t$ increases, we can express the embedding dimension as an affine combination of some top $K$ eigenvectors (where the coefficients depend on how the embedding dimension was initialized):

$$\mathbf{r}(t) \approx \sum_{k=1}^{K}\left(\prod_{t}(1 + \eta\lambda_k(t))\mathbf{r}(0) \cdot \mathbf{e}_k\right)\mathbf{e}_k. \tag{10}$$

---

[9]Note that this is not straightforward to show. It is possible that even if the initial eigenvalues are very close to zero, they approach 0 slower than farther off values.

[10]This is possible only in a dual-encoder, `Node2Vec` style architecture. In a Transformer for example, there are cross-dimensional interactions, due to the associative weight matrix $\mathbf{W}_{\text{assoc}}$ that interfaces between the embedding and unembedding layers.

□

### H.4. Deriving the dynamics

We provide proof of Lemma 1 which expresses the dynamical system of our `Node2Vec` objective in Eq 1.

*Proof.* For a pair of nodes with embeddings $\mathbf{u} \in \mathbb{R}^m, \mathbf{v} \in \mathbb{R}^m$, the probability value of the edge $(\mathbf{u}, \mathbf{v})$ can be written as:

$$p(\mathbf{u}, \mathbf{v}) = \frac{\exp(\mathbf{u} \cdot \mathbf{v})}{\sum_{\mathbf{v}'} \exp(\mathbf{u}, \mathbf{v}')}. \tag{11}$$

Let $N(u)$ denote the neighborhood of the node $u$, and $N_u$, its degree. Let $\mathcal{J}_u$ denote the summand in the objective function specific to that node:

$$\mathcal{J}_u(\mathbf{V}) = \frac{1}{|N(u)|} \sum_{\mathbf{v} \in N(u)} \log p(\mathbf{u}, \mathbf{v}) \tag{12}$$

We now compute the derivative of $\mathcal{J}_u$ with respect to itself $\mathbf{u}$:

$$\frac{\partial \mathcal{J}_u(\mathbf{V})}{\partial \mathbf{u}} = \frac{1}{N_\mathbf{u}} \sum_{\mathbf{v} \in N(u)} \left( \underbrace{\mathbf{v}}_{\texttt{numerator}} - \underbrace{\sum_{\mathbf{v}' \neq \mathbf{u}} p(\mathbf{u}, \mathbf{v}')\mathbf{v}' - 2p(\mathbf{u}, \mathbf{u})\mathbf{u}}_{\texttt{denominator}} \right) \tag{13}$$

$$= \frac{1}{N_\mathbf{u}} \sum_{\mathbf{v} \in N(u)} \mathbf{v} - \sum_{\mathbf{v}' \neq \mathbf{u}} p(\mathbf{u}, \mathbf{v}')\mathbf{v}' - 2p(\mathbf{u}, \mathbf{u})\mathbf{u} \tag{14}$$

Next, we compute the derivative of $\mathcal{J}_w$ with respect to $\mathbf{u}$ for nodes $w \neq u$:

$$\frac{\partial \mathcal{J}_w(\mathbf{V})}{\partial \mathbf{u}} = \frac{1}{N_w} \sum_{\mathbf{v} \in N(w)} \left( \underbrace{\mathbf{1}[u = v]\mathbf{w}}_{\texttt{numerator}} - \underbrace{p(\mathbf{w}, \mathbf{u})\mathbf{w}}_{\texttt{denominator}} \right) \tag{15}$$

$$= \frac{1}{N_w} \sum_{\mathbf{v} \in N(w)} \mathbf{1}[u = v]\mathbf{w} - p(\mathbf{w}, \mathbf{u})\mathbf{w} \tag{16}$$

$$\tag{17}$$

By writing the above expressions as a matrix formula, we get the dynamical system claimed in Lemma 1. □

## H.5. Empirical validation of low-rank spectral bias across graph topologies

### H.5.1. LOW-RANK SPECTRAL BIAS

In Figure 37, we extend Figure 7, showing that for various tiny graphs, a cross-entropy-trained weight-tied two-layer model exhibits a low-rank spectral bias despite there being no explicit pressure; the model is wide enough (embedding size 100 larger than the graph size, eliminating bottleneck pressure) and is trained only on local supervision (eliminating supervisory pressure) and has no explicit regularization (such as $\ell_2$ norm regularization etc.,).

For these plots, we index the eigenvectors for the negative graph Laplacian by their *signed* eigenvalues. We ignore the top eigenvector which is a degenerate eigenvector that assigns near-uniform values to all nodes. Then, for various indices $i$, we plot the magnitude of $\|\mathbf{V}(t)^T \mathbf{e}_i\|/\|\mathbf{V}(t)^T \mathbf{e}_1\|$, which is the projection of the embeddings along the $i$'th direction relative to the projection along the second-top direction. We compare the spread of these relative magnitudes against that of the eigenvalues themselves as $\lambda_i/\lambda_1$. The greater skew in the projections implies a stronger filtering of the bottom directions in the adjacency matrix.

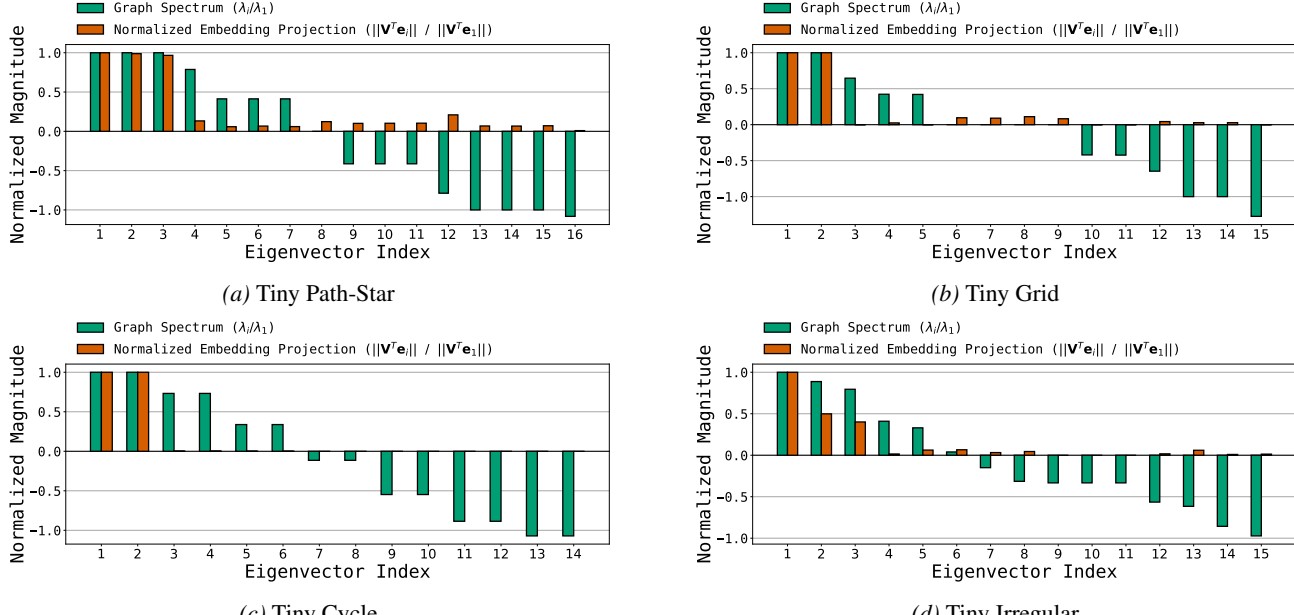

*(a)* Tiny Path-Star

*(b)* Tiny Grid

*(c)* Tiny Cycle

*(d)* Tiny Irregular

*Figure 37.* **Low-rank spectral bias arises naturally in** $2$**-layer cross-entropy-trained models even without explicit pressures (extended version of Figure 7)**. We consider a wide model ($d_{\text{embedding}} = 100$, much larger than nodes in graph, so no rank constraint) trained on local supervision. We report the projection of the weights along the eigenvectors (normalized by that of the top eigenvector). As stated in Observation 3, observe that the model is naturally skewed towards the top vectors, in fact even more skewed than the eigenvalues themselves. (Note that our observations are specific to the skew within the positive eigenvectors; the absence of negative directions in `Node2Vec` is simply because the model can only express positive semi-definite matrices.

### H.5.2. DYNAMICS

In Figures 38 to 41 below, we validate the two core claims of our above analysis intuiting how a low-rank spectral bias emerges without explicit pressures. We show that over the course of training with the cross-entropy loss, the embeddings $\mathbf{V}(t)$ converge more and more towards the top eigenvectors $\mathbf{e}_i$ of the negative graph Laplacian (by observing how $\|\mathbf{V}(t)\mathbf{e}_i\|_2$ evolves over time $t$). Simultaneously, we also show that the co-efficient matrix $\mathbf{C}(t)$ evolves such that its projection along the top eigenvectors decrease toward zero (by observing $\|\mathbf{C}(t)\mathbf{e}_i\|_2$). In effect, this means that the gradient updates which are given by $\mathbf{C}\mathbf{V}$, must also diminish over time. Note that the topmost eigenvector is a degenerate eigenvector that assigns a near-constant value to all nodes; the second-top eigenvector(s) are referred to as Fiedler-like eigenvectors. We refer to these vectors, and the next few ones that follow them, together as "Fiedler-like eigenvectors".

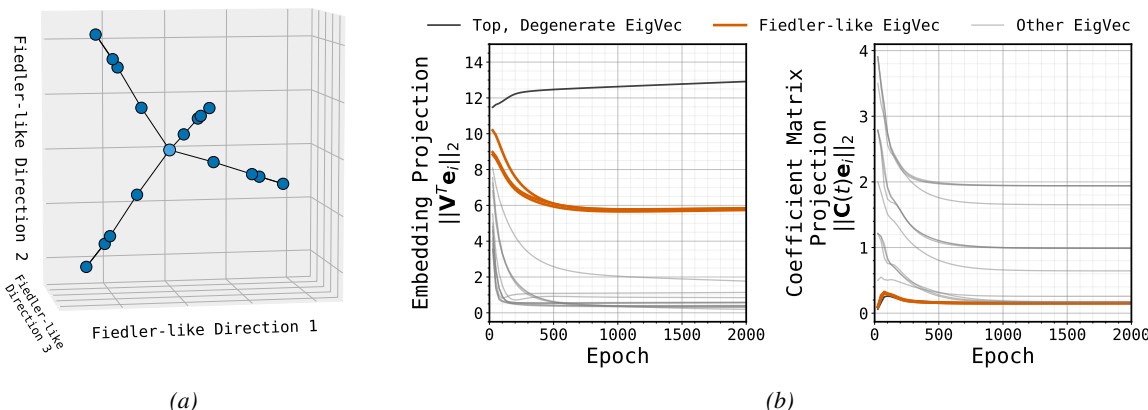

*(a)*          *(b)*

*Figure 38.* **How spectral geometry arises in `Node2Vec` even without low-rank pressure (Observations 3 and 7 for path-star Graph**. (a) The graph's Fiedler vectors shown here closely mirrors the `UMAP` directions of the `Node2Vec` embeddings in Figure 1 (right). (b) The evolution of eigenvector projections during training. *(left)* The embedding matrix $\mathbf{V}$ aligns with the Fiedler-like eigenvectors (and a top, degenerate vector) evidenced by the projection norm $||\mathbf{V}^T\mathbf{e}_i||_2$ converging to a stable, non-zero value. Projections of other eigenvectors diminish towards zero. *(right)* Concurrently, the Fiedler-like eigenvectors move into the null space of the co-efficient matrix $\mathbf{C}$ in that the norm $||\mathbf{C}\mathbf{e}_i||_2$ converges to 0. Crucially, this spectral bias arises without a low dimensional constraint ($d_{\text{embedding}} = 100$, much larger than nodes in graph).

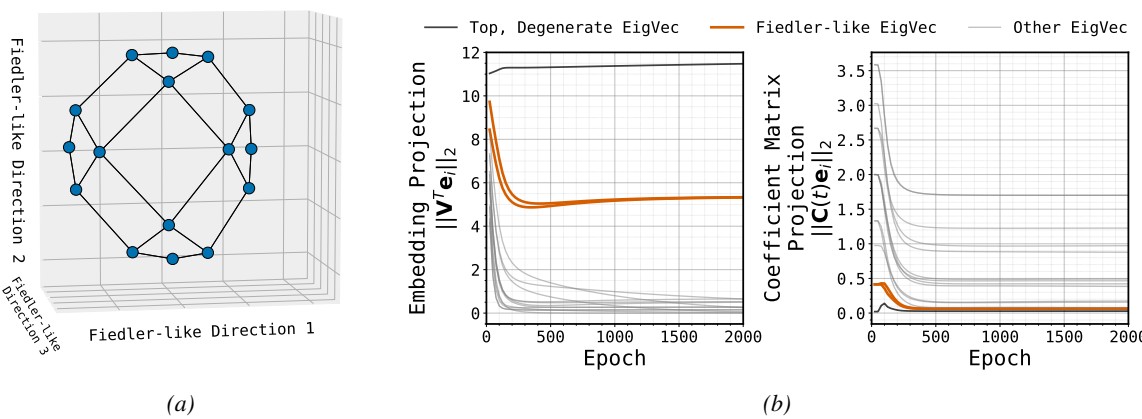

*(a)*          *(b)*

*Figure 39.* **How spectral geometry arises in `Node2Vec` even without low-rank pressure (Observations 3 and 7 for grid graph. (a)** The Fiedler vectors of a grid graph capture spatial locality and connectivity patterns, closely mirroring the `Node2Vec` embedding shown in Figure 17, where a similar geometry emerges. **(b)** Training dynamics show the same two-fold convergence pattern as path-star graphs, where the embeddings align with the top eigenvectors (left) while concurrently, the null space of the co-efficient matrix aligns with the eigenvectors (right).

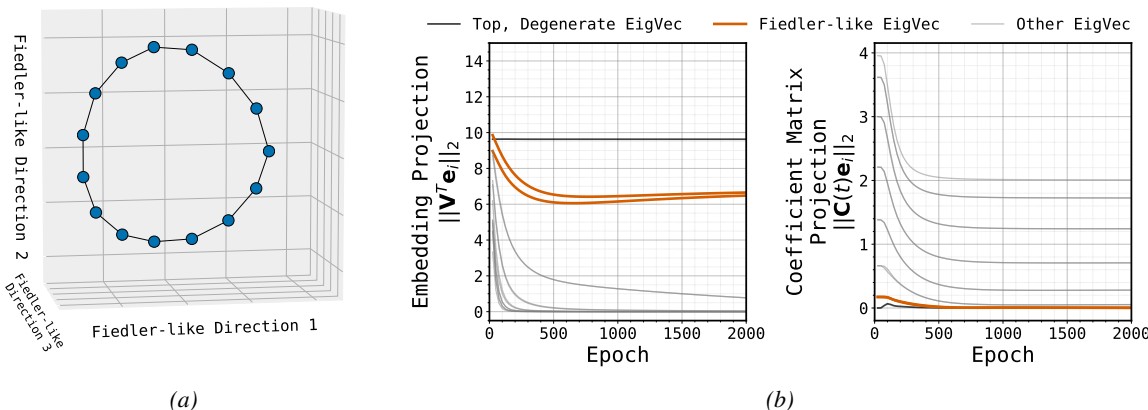

*(a)*                 *(b)*

*Figure 40.* **How spectral geometry arises in `Node2Vec` even without low-rank pressure (Observations 3 and 7 cycle graph. (a)** The Fiedler vectors of a cycle graph reflect the underlying cyclic structure, closely mirroring the `Node2Vec` embedding shown in Figure 18, where a similar geometry emerges. **(b)** Despite the different topology, the same spectral convergence dynamics emerge.

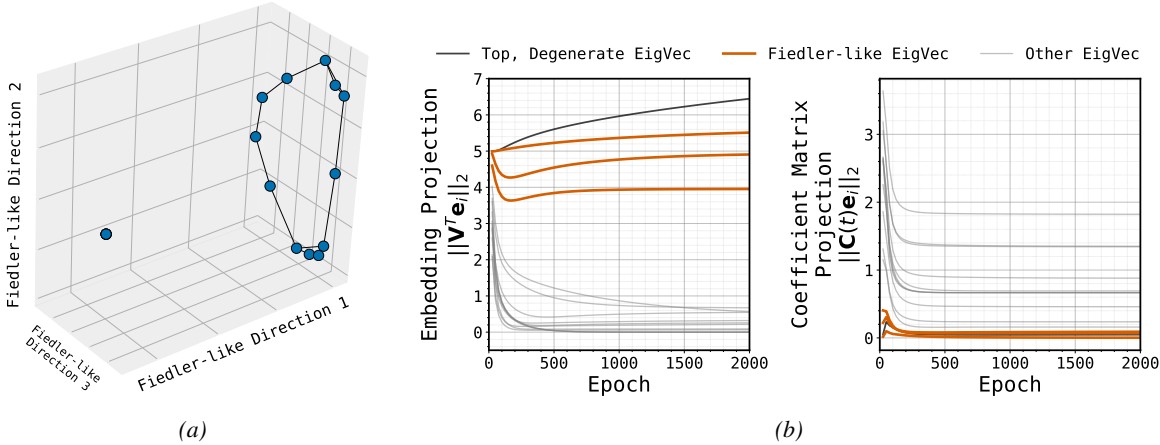

*(a)*                 *(b)*

*Figure 41.* **How spectral geometry arises in `Node2Vec` even without low-rank pressure (Observations 3 and 7) for random graphs.** **(a)** Even in random graphs without clear structural patterns, Fiedler vectors capture the most significant connectivity patterns. **(b)** The spectral convergence dynamics persist, demonstrating robustness across graph types.

