# OpenReview forum: "Deep sequence models tend to memorize geometrically; it is unclear why"
_ICML.cc/2026/Conference — ICML 2026 regular_

### Official Review · Reviewer_uriJ · 2026-03-11

**Soundness:** 4
**Presentation:** 4
**Significance:** 3
**Originality:** 4
**Overall Recommendation:** 5
**Confidence:** 4

**Summary:**

The paper revisits the common assumption that neural sequence models (such as transformers) store facts as associative lookups co-occurring tokens and demonstrate that the stored memories exhibit peculiar geometric structure. To isolate this phenomenon, they devise a path star graph navigation task (known to be difficult to satisfy) and uncover behaviour in embeddings (learnt by transformers, Mamba and node2vec encoders) that unravel an *l step sequential reasoning* task as a *1 step geometric lookup*.

Per my understanding, the core contribution of this work is to carefully devise a clean experimental setup showing that sequence models can internally solve reasoning tasks without explicit step-by-step inference, where : models trained on graph edges learn to answer path-finding queries by organising graph nodes geometrically rather than perform an explicit multi-step reasoning.

Through comprehensive experiments, they further illustrate how this phenomenon cannot be explained fully by architectural pressures, optimiser formulation, or design of training supervision, They posit that spectral bias similar to graph embedding methods like Node2Vec may be responsible for the emergence of geometry and. However, the exact mechanism driving geometric representations (beyond supervisory pressures) and its controllability (eg. spectral structure, graph connectivity) remains open to researchers in the machine learning community.

**Compliance With Llm Reviewing Policy:**

Affirmed.

**Final Justification:**

I had already voted to accept the paper and maintain my score as I think it could be a valuable contribution to the field.

**Key Questions For Authors:**

1. "Thus, we conjecture a similar spectral geometry in sequence models like the Transformer but—in hindsight—mildly adulterated by the competition with local associative memory. Despite this adulteration, the implications are profound: unlike the two-layer models, the Transformer comes with the power to reason over the global geometry, an ability lacking in the models."

While the experiments show that the spectral geometries similar to node2vec emerge during training other models, do you have any insights on the extent of the adulteration as a function of the design of the transformer (or SSM models)?

2. Do the authors reach a definitive conclusion on the role of reverse edges in the supervision from the set of experiments they conduct on why they are so important for this task? Similarly for the other architectural considerations such as pause tokens and weight tying. I would suggest that for the hypothesis driven experiments in the appendix, the authors additionally include a table laying out the setup and conclusions reached, or at least a bulleted list of key takeaways to make it easier to lookup.

3. For their experiments, how do the authors track "convergence"- is this when a stable geometry emerges (and doesn't change) or when training losses do not improve. Are there cases where "convergence" is reached, but the geometry emerges and then changes or gets diluted, especially with the larger capacity models.

**Limitations:**

The main limitation is the lack of straightforward takeaways to applications of deep sequence models, but I would not fault the paper too much for this since this is not its main contribution

**Strengths And Weaknesses:**

STRENGTHS:

- The presentation quality of the paper is excellent, and despite the analytical nature of the contribution, arguments are presented in a way that makes it easy for readers that are not theory experts.

- Arguments made in the paper are very thoroughly substantiated with rigorous experimentation, and results are presented in a systematic way that makes it easy to understand the motivation behind the experiments and interpret the results in connection to the mechanism being questioned.

- The paper's thesis, i.e. the formulation of "a memorisation puzzle" is fundamental to deep sequence models and emergent behaviour, and the posited connection to spectral graph theory is very interesting. The experimental rigour in conjunction with the careful task design makes it a strong contribution to the community.

WEAKNESSES:

- A sizeable amount of the analytical discussion on the phenomenon, especially the key connection to the spectra of node2vec, emergence of stable Eigen-directions across models etc is deferred to the appendix. While I understand that the main contribution of the paper is the analysis of emergence of geometry in the embedding space, I would suggest the authors re-work and improve the explanation in Section 4 to make the connection to the relevant appendix sections easier to follow rather than expect readers to comb through 10+ pages of theoretical exposition

- While the emergence of geometries in the case of the path-star and others graphs was explored, it is not very clear what class of problems this phenomenon generalises to (and consequently what the implications for applications are).

---

> ### Author Rebuttal · Authors · 2026-03-28
>
> We are grateful for your encouraging review of the paper!
>
> > What class of problems this phenomenon generalises to
>
> # **Other graphs and problems**
>
> Let us answer this at increasing levels of generality:
> - **Sec E.2** generalizes the phenomenon to **“branching” graphs** which are much more challenging to navigate as they have a fork at _every_ step.
> -  **There are multiple supporting graph/symbolic tasks that aren't path-star in literature** (see our intro: Khona et al., 2024; Wang et al., 2024a; Feng et al., 2024; Geerts et al., 2025; Ye et al., 2025; Huang et al., 2025) which stand as evidence for geometric memory (although it is not phrased that way). Khona et al., in particular is a distinct enough datapoint as they report path-finding on two types of graphs distinct from path-star.
> - Even **language data (such as years and months)** exhibit a global geometry as shown in subsequent work (https://arxiv.org/abs/2602.15029, Karkada et al.,) although they do not phrase it that way! After all, word co-occurrences are a form of graph data.
> - More broadly, we see the emergence of “**world models**” (Gurnee and Tegmark ‘24, https://arxiv.org/abs/2310.02207) as evidence of a “well-organized global geometry” emerging from (semi)-local data.
> - Even more broadly, we speculate that the “convergent/platonic representation hypothesis” must be a consequence of  the fact that there is only a single spectrum for any given graph connectivity, and this is what is learned by any multi-layered model.
>
> Having said that, the big unanswered question is whether geometries arise (or can be made to arise!) even when entities are more freeform, multi-token descriptions.
>
> ### Implications
> > the implications for applications
>
> In the future, we may/may not discover a geometry in other settings; regardless, there are implications that stems from the very dichotomy we've identified between associative and geometric memory. We can enumerate many questions:
>
> - As `yqP4` points out, research in mechanistic interpretability suggests that geometries/superposition etc., rely on bottleneck assumptions. Are they really necessary?
> - How should one edit knowledge, adapting to how associative or geometric the storage is?
> - When history accumulates in context, should we rather bake it into the model to get more "geometrically" available information?
> - In query-item retrieval, should one use a dual-encoder architecture or should we adopt the single-tower generative retrieval-type architecture (with poorer geometry)?
> - What would current theoretical analysis of scaling laws of memory and storage capacity look like if we adapted it to a geometric storage?
>
>
> Our work exposes these predicaments, and this is independent of whether a geometry itself arises or not!
>
> ---
>
> > Do you have any insights on the extent of the adulteration as a function of the design of the transformer (or SSM models)?
>
> - In follow-up analyses, we discovered that in multi-layer models, **higher weight decay/learning rate/dropout** significantly helps the geometry.
> - We also found that part of the “adulteration” is simply from the rise of certain highly negative “zig-zagging” eigendirections that mess with the visuals. These are directions multi-layer models can represent (a two weight-tied model cannot, since it is positive definite!). So this adulteration is still “geometric”  (it is still a low-rank factorization). See also our related discussion of “zig-zagging” in F.4.
>
> > Do the authors reach a definitive conclusion on the role of reverse edges in the supervision from the set of experiments they conduct on why they are so important for this task?
>
> Our current conclusion (discussed carefully in F.1) is that smaller graphs don’t need reverse edges (see our Fig 30 & also path-finding results in https://arxiv.org/abs/2402.07757 Khona et al.,); larger graphs need it for path-finding. Sadly, we don't  have an explanation; perhaps the well-known reversal curse is key.
>
> > a bulleted list of key takeaways to make it easier to lookup.
>
> We list this in Observation 5 on pg 38. Does it help?
>
> >  how do the authors track "convergence"- is this when a stable geometry emerges?
>
> We had typically trained long enough after top-k accuracy is maxed out; but we agree keeping track of the loss is a cleaner way to do this, and will clarify this.
>
> Thank you for your further presentation feedback which we completely agree with and will incorporate.
> We are happy to address any further questions!

---

> > ### Author Rebuttal · Reviewer_uriJ · 2026-04-03
> >
> > Thank you for the detailed responses to my comments! My review comments have sufficiently been addressed. I maintain my accept rating and enjoyed reading this paper.

---

### Official Review · Reviewer_yqP4 · 2026-03-12

**Soundness:** 3
**Presentation:** 4
**Significance:** 4
**Originality:** 3
**Overall Recommendation:** 5
**Confidence:** 4

**Summary:**

Sequence models (e.g., transformer-based architectures) learn to encode tasks geometrically, rather than using an associative lookup. This geometric encoding explains why these models succeed on NIAH-style path-finding tasks. The learning pressure that drives this representation geometry does not require supervisory signal from a NIAH task during pretraining; it does not require explicitly constrained model capacity; and it is not because associative memories take longer to learn (in fact they do not). Some evidence suggests that the geometry may be driven by the same mechanism that drives representation learning in node2vec.

**Compliance With Llm Reviewing Policy:**

Affirmed.

**Key Questions For Authors:**

Does this work connect with the linear representation hypothesis and the related literature on LLM representation geometry? Or are they conceptually distinct?

In word2vec, representation geometry is driven by factorizing the co-occurrence statistics. Could this be the underlying mechanism here as well? I.e., co-occurrence of 1-hop nodes = adjacency, so the representations are just the spectral factors of the adjacency.

**Limitations:**

yes

**Strengths And Weaknesses:**

I believe this is a very interesting and important paper. It begins to address the questions of whether internal representational geometry precedes the formation of circuits for task-specific computations, and whether superposition-like geometry necessarily requires bottlenecks. For the task considered, the answer appears to be yes, geometry occurs independently of circuits, and no, distinct representations can learn non-negligible overlaps even when the model is large enough to represent them orthogonally. This directly challenges some widely-held intuitions in mechanistic interpretability and the science of LLM explainability. (This angle is not explored in the paper, though I think these communities would find these points particularly interesting.)

It would be interesting to see plots of learning dynamics. Does the associative solution get learned first, followed by the geometric one? If so, what learning signal drives the transition from associative to geometric? Do the top Laplacian eigenvectors get learned first?

I am also curious about which of these observations might generalize to other graphs (not just star graphs) or even other tasks (learning transitions of finite state automata, see https://arxiv.org/pdf/2405.15943).

The experiments are clean, and competing hypotheses are tested independently before being ruled out. The scientific conclusions are clear, even if they are results are negative (i.e., the mystery remains). This paper provides an opportunity for future work to resolve the mystery; such a resolution would constitute an advance in the scientific understanding of how and why sequence models represent concepts in the way they do.

The plots are convincing and clear.

---

> ### Author Rebuttal · Authors · 2026-03-28
>
> Thank you for your thoughtful and positive feedback on the paper! **Thanks in particular, for bringing up a valuable connection to mechanistic interpretability literature. This is a great point, we will be sure to mention this in future versions of the paper.**
>
> ---
>
> There are a few questions best answered together:
>
> > Does the associative solution get learned first, followed by the geometric one?
> > Do the top Laplacian eigenvectors get learned first?
> > In word2vec, representation geometry is driven by factorizing the co-occurrence statistics. Could this be the underlying mechanism here as well?
> > What learning signal drives the transition from associative to geometric?
>
>
> Yes to the first three questions! Note that the models end up learning only the top eigenvectors (e.g., two directions for the cycle graph), and the rest simply get ignored/never arise — surprisingly, even when there’s no bottleneck limiting the model to two. This behavior is unique to CE loss; we explain why this "implicit-bottlenecking" happens in Sec 4 for two-layered models.
>
> The “associative first, geometric later” can be intuited through a simpler objective like $\sum\_{(u,v) \in E} f(u)[v]$ where you increase the logit of your neighbors; consider a 3-layer model $\phi^T W \phi$. The first gradient step would move the $W$ matrix in the direction of $A$ upto the rotation defined by $\phi$ — this is associative storage. There are also competing gradients where the model uses only a few directions in the $\phi^T ... \phi$ layers to approximate moving along $A$ (while the $W$ is forced to focus on these directions). This is indeed, as you say, a _factorized_ form of storage from which geometry arises; this factorization into the top eigenvectors requires many more gradient steps. Intuitively, learning the top eigenvectors corresponds to storing $A^L$ for some larger $L$, which takes more gradients to iteratively multiply the weights by $A$ (captured in `Eq 9, page 60`).
>
> ---
>
> > generalization to other graphs (not just star graphs) or even other tasks (learning transitions of finite state automata, see https://arxiv.org/pdf/2405.15943).
>
> Please see "Other graphs and problems" in response to `uriJ`. In short, **Sec E.2 generalizes to “branching” graphs** and there are supporting observations in literature (although not interpreted the way we do).
>
> The automata example is a fascinating observation and valuable setup; we suspect it may be complementary to the graph learning framework: the interestingness of the automata seems to come not from having to geometrically memorize a large graph, but from inferring a hidden state based off of a very tiny graph-based transition. The tasks however are similar in terms of their "in-weights" nature.
>
> > Connection to linear representation hypothesis
>
> Nice question! Our phenomenon is complementary to LRH. We discuss this in Line 1337.
>
> LRH studies can be summarized as: `_given that models learn some spectral embedding_, why does some algebraic structure arise in the embeddings? What structure in graphs/co-occurrences (e.g., friend of a friend is a friend) must result in that?`.
>
> We are interested in a complementary, precursor question: `why does the model even learn a spectral embedding in the first place, rather than storing co-occurrences as is?` This does not require ideating about specific structural properties about the graph (such as transitivity etc.,) that the LRH literature has studied. It's applicable to a more general class of graphs.
>
> ---
>
> Thank you once again for your time; we hope you found our answers helpful. We are happy to address any further questions from you!

---

> > ### Author Rebuttal · Reviewer_yqP4 · 2026-04-03
> >
> > Thanks! Cool paper, I hope it gets traction.

---

### Official Review · Reviewer_5avu · 2026-03-13

**Soundness:** 3
**Presentation:** 3
**Significance:** 4
**Originality:** 3
**Overall Recommendation:** 4
**Confidence:** 2

**Summary:**

The paper introduces a distinction between two modes of parametric memory in deep sequence models: associative memory, where co-occurring entities are stored as a lookup table, and geometric memory, where the model's embeddings encode global graph structure (e.g., multi-hop distances) not explicitly present in the local training signal. The authors demonstrate that next-token-trained Transformers and Mamba successfully learn an in-weights path-finding task on path-star graphs at scales up to $10^4$ nodes, despite the task being adversarially designed to defeat in-context learners. They show this success traces to a structured global geometry in the learned embeddings, then argue that standard explanations (global supervision, capacity pressure, optimizer bias) each fail to account for the geometry. Finally, they connect the phenomenon to spectral biases in 2-layer Node2Vec-style models, providing a partial theoretical account grounded in the eigenvectors of the graph's adjacency matrix.

**Compliance With Llm Reviewing Policy:**

Affirmed.

**Final Justification:**

My main concerns are mostly addressed.

The two bridges for the theory-to-Transformer gap, eigenvector projection barplots and the Saxe et al. analogy, do not provide a complete mechanistic proof, but they are reasonable within the paper's scope. The remaining gap is now clearly marked instead of overlooked. The frozen-embedding experiments) offer a valid associative solution by construction, which resolves my concern about the absence of associative baselines. The external evidence on generalizability is indirect but suitable.

I maintain my score of 4 and encourage the authors to include the eigenvector barplots and the hyperparameter-geometry findings in the final version.

**Key Questions For Authors:**

1. The path-finding training examples cover 75% of leaves (§2.1), and these examples expose the first token of each root-to-leaf path. Does the geometry arise even when training purely on edge-memorization examples, with no path-finding examples whatsoever?

2. The 2-layer spectral analysis shows that cross-entropy loss drives embeddings toward the top eigenvectors of the graph's adjacency matrix, without explicit pressures. But Transformers use multi-layer attention with residual connections, position encodings, and softmax routing. None of which are modeled in the 2-layer Node2Vec analysis. Can the authors identify even one structural property of Transformers (e.g., weight tying between embedding and unembedding layers, the self-attention inductive bias) that would make the spectral bias argument carry over?

3. The paper claims in §3.2 that associative and geometric memories are equally succinct for path-star graphs (Refutation 3d, Observation in §G.2). The informal argument is that both scale similarly in bits and L_2 norm. Does this equality hold for other graph families (e.g., social network graphs) where the spectrum is much flatter?

4. Section 4 argues that Node2Vec geometries are "much more well-organized" than those of Transformers. If Transformer memory is only partially geometric, are there architectural interventions, beyond the headroom analysis hinted at in §1, that would push a Transformer toward the cleaner Node2Vec geometry?

**Limitations:**

yes

**Strengths And Weaknesses:**

## Strength

1. The core experimental finding is cleanly demonstrated. The path-star task was adversarially designed to defeat in-context learning, so adapting it to an in-weights setting creates a sharp, interpretable probe. The fact that the model succeeds even when trained only on edge-memorization examples and the first token of paths, without any step-wise supervision, is genuinely hard to reconcile with the associative view, and the paper isolates this contradiction more clearly than prior work by removing in-context reasoning as a confounder.

2. The refutation structure is well-organized. The authors enumerate four plausible explanations, global supervision, capacity pressure, optimizational pressure, implicit capacity, and systematically refute each. The refutation of the "gradient descent as implicit capacity pressure" argument is particularly careful: they show that associative and geometric memories have equal complexity in the relevant graph families, so a succinct-fit explanation cannot differentiate them.

3. The paper's own self-awareness about scope is a point in its favor. By titling the paper "it is unclear why", the authors explicitly acknowledge they solve the memorization puzzle only partially (via spectral bias in 2-layer models) and leave the competition between associative and geometric memories in deeper models as an open question. This framing is more credible than overclaiming.


## Weaknesses

1. The gap between the 2-layer theory and the main Transformer/Mamba results is wider than the paper acknowledges. The spectral bias analysis (§4, §H) applies to a 2-layer Node2Vec-style model. The paper then argues that "a similar spectral geometry" must arise in Transformers, but this inference rests on visual similarity of embedding heatmaps (Figure 1) and the observation that Transformers are "mildly adulterated" versions of the simpler geometry. There is no theoretical or mechanistic bridge from the 2-layer analysis to multi-layer attention-based models. The 2-layer analysis shows how a low-rank spectral bias arises in a simplified setting, which does not explain why Transformers exhibit geometric memory at all, let alone why they would exhibit the same spectral structure.

2. The generalizability of the path-star findings to real-world sequence modeling is unclear. Path-star graphs are highly structured: fixed degree, uniform path length, disjoint paths, and a fixed global graph across all training examples. The memorization puzzle is posed and analyzed exclusively in this synthetic setting. The paper mentions in §6 that graph complexity (spectrum, connectivity, size) should determine whether geometric vs. associative storage is preferred, but does not empirically investigate any real-world graph or NLP setting.

3. The paper's treatment of "competition" between associative and geometric memory (raised in §2.4 and left open in §6) lacks a concrete experimental handle. The claim is that both are equally valid solutions to the training loss, and that which one prevails depends on the model's inductive bias. But the paper does not show a single case where associative memory provably wins, where all experiments show geometric outcomes. The refutations in §3 eliminate pressure-based explanations for why geometric memory arises, but do not explain what determines the degree of geometry, nor under what conditions a model would strongly prefer associative storage.

---

> ### Author Rebuttal · Authors · 2026-03-28
>
> Thank you for your time and effort in reading and reviewing our paper. We are happy to answer your questions below:
>
> > The gap between the 2-layer theory and the main Transformer/Mamba results... no theoretical or mechanistic bridge to multi-layer attention-based models...  inference rests on visual similarity
>
> For this, we offer two bridges (that we missed while writing the paper):
> - We projected the embedding layers of multi-layer models along the eigenvectors of the adjacency matrix $A$; **this produces  barplots similar to Node2Vec in Figure 7**; thus, the deeper model indeed relies only on the top eigendirections of $A$. We absolutely agree with you that the visual plots in themselves are not a trustworthy verification — so we’ll add these barplots.
> - As an intuitive theoretical bridge: the seminal analysis of Saxe et al., (https://arxiv.org/abs/1312.6120) shows that a 2-layer model  (under sq. error loss) learns an (SVD) factorization; they show how this extends to multi-layer models, under a special initialization where each layer's random singular directions are aligned with the previous layer's directions. **We can extend our argument similarly to _envision_ how (for CE loss + graph data setting) a purely geometric/spectral dynamic is _plausible_ and can propagate through our non-bottlenecked networks.** The gap that exists is analyzing the competition with associative memory; this competition appears when you get rid of Saxe et al’s assumption about the careful initialization. We openly acknowledge this gap (in line 420, col 1)  but are happy to emphasize it further.
>
> **Do these two bridges help sense the two$\to$multi-layer connection?** Our two layer analysis provides the necessary starting point to understand why a bottleneck is not needed, and we hope, a path forward.  We’ll make these bridges clearer in future versions of the paper. We thank you for helping us notice this.
>
> ---
>
> > The claim is that both are equally valid solutions to the training loss…. paper does not show a single case where associative memory provably wins, where all experiments show geometric outcomes.
>
> We point out otherwise:
> - **Multiple experiments show that associative memory is indeed a valid solution:** we freeze the embedding layer and have only one intermediate layer (e.g., Refutation 3b or Fig 6 left)  and show that the data can be fit. This solution can only be associative since only one trainable matrix exists (no factorization is possible!)
> - In follow-up analysis since this paper, we found that reducing the learning rate significantly (1e-4 or less) & setting the weight decay=0 and switching off dropout in the dataset, results in more associative solutions even in deeper models. We will make these clarifications.
>
> It's important that we make this clearer -- thank you!
>
> > The generalizability of the path-star findings to real-world sequence modeling is unclear.
>
> Please see our response to `uriJ` under *Other graphs and problems*. In short, there are observations about world models (https://arxiv.org/abs/2310.02207, Gurnee and Tegmark) and in language such as months/years (https://arxiv.org/abs/2602.15029, Karkada et al.,) where one can observe that a "well-organized global geometry" has risen from local data (although these papers do not phrase it that way).
>
>
> > Does geometry arise even… on edge-memorization examples, with no path-finding examples whatsoever?
>
> Yes, although weakly; please see Figure 5 for Transformer and Figure 13(c) Mamba (which shows a clearer geometry).  Also, all the tiny graphs experiments show a geometry without path-finding.
>
> > Transformers use multi-layer attention with residual connections, position encodings, and softmax routing.
>
> Note that even multi-layer MLPs memorize geometrically! So these extra connections, while potentially aiding the geometry, are not critical. The key question is why depth still allows geometries to arise, which we touch upon in our first point in the rebuttal.
>
> > Does this equality [of the bit/norm complexity] hold for other graph families (e.g., social network graphs) where the spectrum is much flatter?
>
> Nice question. The equality does not hold on generic graphs, but that may deceive us into thinking that norms/bits does explain the geometry. The graphs we present (where `num_edges = num_vertices`) are meant as  a **counterexample**.
>
>
> > are there architectural interventions, beyond the headroom analysis hinted at in §1…push towards geometry?
>
> As a corollary of one of our points above, we noticed that increasing `weight_decay` or `learning_rate` or `dropout` improves the geometry.
>
> ----
>
> We hope we were able to help address your concerns including how the dynamics in a 2-layer model does have a plausible way to arise in a deeper model; the main gap is a competition with associative memory which we acknowledge in the paper and can highlight further! If not, would anything else help address this concern? Thanks again for your positive review!

---

> > ### Author Rebuttal · Reviewer_5avu · 2026-04-04
> >
> > Thank you for the detailed response. My main concerns are mostly addressed.
> >
> > The two bridges for the theory-to-Transformer gap, eigenvector projection barplots and the Saxe et al. analogy, do not provide a complete mechanistic proof, but they are reasonable within the paper's scope. The remaining gap is now clearly marked instead of overlooked. The frozen-embedding experiments) offer a valid associative solution by construction, which resolves my concern about the absence of associative baselines. The external evidence on generalizability is indirect but suitable.
> >
> > I maintain my score of 4 and encourage the authors to include the eigenvector barplots and the hyperparameter-geometry findings in the final version.

---

### Official Review · Reviewer_uko9 · 2026-03-13

**Soundness:** 3
**Presentation:** 3
**Significance:** 2
**Originality:** 3
**Overall Recommendation:** 4
**Confidence:** 3

**Summary:**

The paper studies whether deep sequence models memorize facts primarily as local associative lookups or as a more global geometric representation. Using synthetic in-weights path-finding tasks on path-star graphs, the authors show that Transformers and Mamba can solve cases that are difficult to explain under a purely associative view. The paper then argues that this geometric form of memorization is surprising under standard intuitions about supervision, architecture, and optimization, and offers a partial explanation via spectral bias in a simplified 2-layer Node2Vec-style model.

**Compliance With Llm Reviewing Policy:**

Affirmed.

**Final Justification:**

Overall, the rebuttal provides helpful clarifications and increases my confidence in the robustness of the phenomenon. However, some questions regarding the underlying mechanism and the theory-to-model linkage remain open. I therefore maintain my current scores.

**Key Questions For Authors:**

1.How robust is the main phenomenon to removing the main training conveniences one at a time? In particular, I would like to understand the necessity of reverse-edge augmentation, pause tokens, and interleaving. A strong robustness result would substantially increase my confidence in the paper’s central claim.

2.Can the authors provide a more operational measure of associative vs. geometric storage within the same trained model? This would sharpen the mechanistic interpretation and reduce reliance on behavioral evidence alone.

3.How much does the result extend beyond path-star-like graph families? Positive evidence on a broader set of graph structures would materially strengthen the scope of the claim.

4.How predictive is the spectral-bias analysis for the actual deep sequence models studied here? For example, is there a quantitative relation between spectral structure in the learned embeddings and success on the hard path-finding cases?

**Limitations:**

Yes. The authors discuss key limitations, including the focus on synthetic tasks and the partial nature of the explanation. The work is primarily foundational, and no significant negative societal impact is apparent.

**Strengths And Weaknesses:**

**1.Strengths：**

1.Clear and interesting question. The paper poses a sharp conceptual distinction between associative and geometric memory, and turns it into a concrete empirical and theoretical investigation.

2.Well-chosen synthetic setup. The path-star task is a strong testbed for isolating the claimed phenomenon, and the in-weights setting makes the result more interesting than a standard in-context reasoning result.

3.Promising cross-architecture evidence. Similar trends in both Transformers and Mamba suggest the effect is not obviously tied to one specific architecture.

4.High originality. The main contribution is a new perspective on parametric memory rather than a new model, but this is a meaningful form of originality and could motivate follow-up work.

**2.Weaknesses：**

1.Evidence is narrower than the title-level claim. The main results are concentrated in synthetic graph settings, especially path-star-like constructions. This establishes an interesting phenomenon, but does not yet fully support the broader claim that deep sequence models generally tend to memorize geometrically.

2.The explanation is only partial. The spectral-bias account is insightful, but it is established in a simplified 2-layer setting rather than in the main deep sequence models of interest.

3.Robustness to training choices is not fully characterized. The observed behavior appears to depend on several recipe details such as reverse-edge augmentation, pause tokens, and example interleaving. A clearer robustness analysis would strengthen the claim that the effect emerges naturally.

4.Operational separation of associative vs. geometric memory remains unclear. The distinction is conceptually useful, but the paper does not yet provide a sharp internal metric or causal intervention that disentangles the two within a trained deep model.

5.From a presentation standpoint, several figures contain small text that is difficult to read at standard zoom, and enlarging labels would improve clarity. Minor language polishing could also improve readability, though the writing is generally clear.

Overall assessment

I found the paper original and thought-provoking. My main reservation is not that the phenomenon is uninteresting, but that the evidence currently supports a narrower claim than the paper sometimes suggests. I view the paper as stronger as a formulation of an important puzzle than as a definitive resolution of it.

---

> ### Author Rebuttal · Authors · 2026-03-28
>
> Thank you for your efforts in reviewing our paper, and for your positive review. We hear three main concerns, addressed below.
>
> ## Synthetic graphs
>
> > The main results are concentrated in synthetic graph settings, especially path-star-like constructions... **Positive evidence on a broader set of graph structures would materially strengthen the scope of the claim.**
>
> Let us clarify how our claims do generalize to varying extents:
> - Sec E.2 concretely generalizes the phenomenon to **“branching” graphs which are much more challenging** to navigate as they have a fork at every step.
> - There are **multiple supporting graph/symbolic tasks that aren't path-star in literature** (see our intro: Khona et al., 2024; Wang et al., 2024a; Feng et al., 2024; Geerts et al., 2025; Ye et al., 2025; Huang et al., 2025) which stand as evidence for geometric memory (although it is not phrased that way). Khona et al., (https://arxiv.org/abs/2402.07757) in particular is a distinct enough datapoint as they report path-finding on two types of graphs distinct from path-star.
>
> More broadly:
> - Even **language data (such as years and months)** exhibit a global geometry as shown in subsequent work (https://arxiv.org/abs/2602.15029, Karkada et al.,) although they do not phrase it that way! After all, word co-occurrences are a form of graph data.
> - More broadly, we see the emergence of “**world models**” (Gurnee and Tegmark ‘24, https://arxiv.org/abs/2310.02207) as evidence of a “well-organized global geometry” emerging from (semi)-local data.
> - Even more broadly, we speculate that the “convergent/platonic representation hypothesis” must be a consequence of the fact that there is only a single spectrum for any given graph connectivity, and this is what is learned by any multi-layered model.
>
> ## Robustness analysis
>
> > … reverse-edge augmentation, pause tokens, and example interleaving.  / How robust is the main phenomenon to removing the main training conveniences one at a time... **A strong robustness result would substantially increase my confidence in the paper’s central claim.**
>
> **Sec F** provides these analyses.
>  - Only the reverse edges are important (which aligns with the well-known "reversal curse").
>  - Pause tokens only speed up convergence (significantly) but are **not necessary**.
>  - Example interleaving is **not necessary**; two-stage finetuning works too, but is less enduring i.e., the highest accuracy is similar, but model begins to forget with further training.
>
> ## Partial explanation
>
> > The explanation is only partial… The spectral-bias account is insightful, but it is established in a simplified 2-layer setting rather than in the main deep sequence models of interest.
>
> Thank you for acknowledging the insight, which we admit has a gap (hence the title, `it's unclear why`, as `5avu` notes). But we missed mentioning **a key bridge in this gap**:
>
> The seminal analysis of Saxe et al., (https://arxiv.org/abs/1312.6120) shows that a (bottlenecked) 2-layer model (under sq. error) learns an (SVD) factorization; they show how this extends to multi-layer models, under a special initialization where each layer's random singular directions are aligned the previous layer's directions.
> **We can extend our argument similarly to _envision_ how (for CE loss + graph data setting) a purely geometric/spectral dynamic is _plausible_ and can propagate through our non-bottlenecked networks.**  The gap this leaves is analyzing the competition with associative memory; this competition appears when you get rid of Saxe et al’s assumption about the careful initialization. Thus, our two layer analysis provides the necessary starting point to understand why a bottleneck is not needed, and a path forward!
>
> > associative vs. geometric memory... distinction is conceptually useful, but the paper does not yet provide a sharp internal metric or causal intervention that disentangles the two within a trained deep model.
>
> We agree with this limitation. A causal intervention we may suggest is a separate stage of multi-hop path-finding with held-out test data; noting that this can be unstable due to forgetting.
>
>
> > ls there a quantitative relation between spectral structure in the learned embeddings and success on the hard path-finding cases?
>
> Interesting question! Spectral graph theory says that the top eigenvectors (called the Fiedler vectors) correspond to global connectivity patterns in the graph; we hope there may be insights in this theory that can quantify the relationship. For the specific path-star task, there is a straightforward relationship. In its spectrum one can compute the hardest first node as just `(embedding(goal_node) - embedding(start_node))/path_length`.
>
> ----
>
> Thanks for your other writing/plot feedback!
>
> Does (a) the varying graph examples (b) the existing robustness analyses and (c) the "vision" in the theory help view our claims more confidently? If not, happy to answer any further questions.

---

> > ### Author Rebuttal · Reviewer_uko9 · 2026-04-01
> >
> > Thank you for the detailed rebuttal.
> >
> > Two concerns remain. First, the rebuttal explicitly acknowledges the absence of an operational criterion to disentangle associative from geometric memory within trained models. Second, the discussion of the spectral account does not yet establish a quantitative predictive link between the two-layer analysis and performance on hard cases in the actual deep models.
> >
> > Overall, the rebuttal increases my confidence in the robustness of the phenomenon but does not fully close the mechanistic and theory-to-model gaps. I therefore maintain my current scores.

---

### Decision · Program_Chairs · 2026-04-30

**Decision:**

Accept (regular)

**Comment:**

All reviewers found the detailed well-organized experiments demonstrating the "in-weights learning" capability of sequence models such as transformers and Mamba extremely convincing, where the results clearly show that these models are able to learn to perform multi-hop graph traversal on path-star graphs while trained solely on edge memorization. Well designed experiments study and refute usual explanations of this behaviour, though large and deep sequence models have not been systematically considered. Thus, it is not clear whether scale has any role in this behaviour.

The authors discuss this behaviour in the context of associative and geometric memorization, where associative memorization rely on single-shot pattern matching while geometric memorization requires multi-step composition. The authors argue that the models are able to generalize because they are geometrically memorizing, and this learned compositional behaviour cannot be explained by usual explanations of capacity pressure, step-by-step supervision, and such. However, this distinction between the associative memorization and geometric one is not precise enough to be quantified (and thus used to probe the learned models), leaving much of this discussion open to interpretation.

The authors do provide a direction for explaining this behaviour in terms of a spectral bias, though this study is still empirical and preliminary.

Overall, this submission demonstrates this phenomenon clearly with well-crafted experiments and motivations. Thus, I recommend this submission be accepted.

-------

As a minor point, the latest version of this reference has a slightly different title and an additional author:
> Wang, X., Tan, S., Jin, M., Wang, W. Y., Panda, R., and Shen, Y. Do larger language models imply better generalization? a pretraining scaling law for implicit reasoning, 2025a. URL https://arxiv.org/abs/2504.03635